# Loss of the batten disease protein CLN3 leads to mis-trafficking of M6PR and defective autophagic-lysosomal reformation

Alessia Calcagni'[1,2] ✉, Leopoldo Staiano[3,4], Nicolina Zampelli[3], Nadia Minopoli[3,5], Niculin J. Herz[1,2], Giuseppe Di Tullio[3], Tuong Huynh[1,2], Jlenia Monfregola[3], Alessandra Esposito[3,6], Carmine Cirillo[3], Aleksandar Bajic[1,2], Mahla Zahabiyon[1,2], Rachel Curnock[7], Elena Polishchuk[3], Luke Parkitny[1,2], Diego Luis Medina[3,5], Nunzia Pastore[3,5], Peter J. Cullen[7], Giancarlo Parenti[3,5], Maria Antonietta De Matteis[3,8], Paolo Grumati[3,9] & Andrea Ballabio[1,2,3,5,6] ✉

Batten disease, one of the most devastating types of neurodegenerative lysosomal storage disorders, is caused by mutations in *CLN3*. Here, we show that CLN3 is a vesicular trafficking hub connecting the Golgi and lysosome compartments. Proteomic analysis reveals that CLN3 interacts with several endo-lysosomal trafficking proteins, including the cation-independent mannose 6 phosphate receptor (CI-M6PR), which coordinates the targeting of lysosomal enzymes to lysosomes. CLN3 depletion results in mis-trafficking of CI-M6PR, mis-sorting of lysosomal enzymes, and defective autophagic lysosomal reformation. Conversely, CLN3 overexpression promotes the formation of multiple lysosomal tubules, which are autophagy and CI-M6PR-dependent, generating newly formed proto-lysosomes. Together, our findings reveal that CLN3 functions as a link between the M6P-dependent trafficking of lysosomal enzymes and lysosomal reformation pathway, explaining the global impairment of lysosomal function in Batten disease.

Neuronal Ceroid Lipofuscinoses (NCLs) are neurodegenerative lysosomal storage disorders (LSDs) associated with the accumulation of auto-fluorescent ceroid material (lipofuscins) inside lysosomes[1]. NCLs are caused by mutations in genes encoding either transmembrane (e.g., CLN3, CLN7) or soluble (e.g., CLN1, CLN2, CLN5) lysosomal proteins, as well as non-lysosomal proteins (CLN6, CLN8)[2–4]. Unfortunately, the biochemical function and physiological role of most CLN proteins remains elusive. This also applies to CLN3, whose mutations underlie a severe form of NCL, known as Batten disease (NCL3)[5]. CLN3

is a predicted multi-pass transmembrane protein that was reported to be localized predominantly at the lysosome[6]. Similarly to other NCLs, NCL3 disease is associated with ceroid lipofuscin accumulation primarily in the brain[5], with devastating disease manifestations spanning from blindness, seizures, and dementia, to the progressive loss of cognitive and motor skills, ultimately leading to death. These symptoms typically arise at ~4–7 years of age, and cause death by age 30[2,7,8]. Most patients with NCL3 carry a homozygous 1.02 Kb deletion in the CLN3 gene, whereas some are compound heterozygotes, carrying this

[1]Department of Molecular and Human Genetics, Baylor College of Medicine, Houston, TX 77030, USA. [2]Jan and Dan Duncan Neurological Research Institute, Texas Children's Hospital, Houston, TX 77030, USA. [3]Telethon Institute of Genetics and Medicine (TIGEM), Naples, Italy. [4]Institute for Genetic and Biomedical Research, National Research Council (CNR), Milan, Italy. [5]Department of Translational Medical Sciences, Federico II University, 80131 Naples, Italy. [6]SSM School for Advanced Studies, Federico II University, Naples, Italy. [7]School of Biochemistry, Biomedical Sciences Building, University of Bristol, Bristol BS8 1TD, UK. [8]Department of Molecular Medicine and Medical Biotechnology, Federico II University, Naples, Italy. [9]Department of Clinical Medicine and Surgery, Federico II University, Naples, Italy. ✉e-mail: alessia.calcagni@bcm.edu; ballabio@tigem.it

deletion in combination with other mutations[9–12]. Currently approved therapies for NCL3 only manage symptoms, whereas effective therapeutic approaches are unavailable, mostly due to the lack of knowledge on the biological function of CLN3.

Lysosomal enzymes after being synthesized in the endoplasmic reticulum, they traffic through the Golgi where they acquire M6P residues, and then are targeted to the endo-lysosomes through the recognition of M6P residues by mannose-6-phosphate receptors (M6PRs)[13]. We discovered that CLN3 regulates the trafficking of CI-M6PR, thus playing a central role in the targeting of lysosomal enzymes to the lysosome. We also discovered that CLN3 is involved in Autophagic Lysosomal Reformation (ALR), an essential process to generate new functioning proto-lysosomes from mature lysosomes[14]. This process is promoted by prolonged starvation, a condition that activates autophagy. Upon autophagosome-lysosome fusion, cargo degradation results in the release of nutrients that are then required for mTOR reactivation. This triggers a membrane remodeling mechanism characterized by the generation of PI(4,5)P$_2$ subdomains, and the assembly of AP2-clathrin lattices, where the anchoring of kinesin-1 determines the generation of tubules and of novel proto-lysosomes[15,16]. Mutations in genes important for lysosomal function, have already been associated with ALR. For example, mutations in spinster, a critical regulator of ALR[17], result in alteration of lysosomal pH, defective lysosomal degradative function, and faulty ALR due to failure of mTOR reactivation[17]. Also, in flies, spinster mutations recapitulate several features of LSDs[18]. Notably, failure of ALR was also shown in some LSDs such as Scheie syndrome, Fabry disease, and Aspartylglucosaminuria[15].

Here we show that CLN3 localizes both at the Golgi apparatus and at the lysosome, where it interacts with multiple trafficking protein complexes to mediate recycling of CI-M6PR at the Golgi. Loss-of-function of CLN3 causes degradation of CI-M6PR in lysosomes, mis-targeting and secretion of lysosomal enzymes, and impairment of autophagic-lysosomal reformation.

## Results

### CLN3 traffics through the Golgi apparatus and lysosomes

The absence of reliable CLN3 antibodies has hindered studies on endogenous CLN3 protein localization and kinetics. We generated a novel anti-CLN3 polyclonal antibody to study the subcellular localization of the endogenous CLN3 protein. In ARPE19 cells, endogenous CLN3 was equally distributed between lysosomes and a perinuclear area overlapping with the trans-Golgi network marker TGN46 (Fig. 1a, b), and was undetectable on EEA1-positive early endosomes (Supplementary Fig. 1a, b) whereas no specific signal was observed in CLN3-KO cells (Fig. 1a). CLN3 Golgi signal in WT ARPE19 cells was also detected with Airyscan super-resolution microscopy (Fig. 1c). To further validate CLN3 subcellular distribution, we generated a doxycycline inducible CLN3 construct that was internally tagged (CLN3-innHA), as previous observations showed protein mis-localization upon N- or C-terminal tagging[9,19,20]. We generated ARPE19 pLVX-CLN3innHA stable cell lines and confirmed CLN3-innHA Golgi-lysosomal localization (Supplementary Fig. 1c). The same distribution pattern was also observed in HeLa cells (Supplementary Fig. 1d), no specific signal was detectable in CLN3 KO cells (Supplementary Fig. 1e). Overall, these results indicate that CLN3 localizes at both Golgi and lysosomal compartments. Immunoblot (IB) analysis performed in ARPE19 cells identified a 45 kDa and a 65–80 kDa form, which were both lost upon siRNA-mediated knock-down (siCLN3) and in CRISPR/Cas9 CLN3 KO cells (Supplementary Fig. 1f, g). The same result was confirmed in HeLa WT and CLN3-KO cells (Supplementary Fig. 1h, i). The antibody was further validated by performing CLN3-immunopurification on ARPE19 WT and CLN3-KO cells. This experiment clearly confirmed the presence of the 65–80 KDa band only in WT cells (Fig. 1d). Unfortunately, the lower band was not detectable because the signal was covered by the IgG heavy chains.

Tunicamycin treatment, which inhibits N-linked glycosylation, resulted in the appearance of a 42 kDa band, confirming that both CLN3 forms are glycosylated[21] (Supplementary Fig. 2a). Lysosome immuno-purification (Lyso-IP)[22] revealed that only the 65–80 kDa protein form localizes to lysosomes (Supplementary Fig. 2b), suggesting that the two CLN3 forms may represent different stages of protein maturation. To test this hypothesis, we used the doxycycline inducible CLN3 construct to temporally modulate CLN3 overexpression. Blocking CLN3-specific and total protein synthesis through concomitant removal of doxycycline and addition of cycloheximide (CHX) revealed that the 65–80 kDa band is long-lived, with a half-life of 12 h, whereas the low-molecular weight band is lost 2 h after inhibiting translation (Supplementary Fig. 2c). Short doxycycline treatments on CLN3-innHA cells resulted in the production of the 45 kDa form, which gradually disappeared after long doxycycline washouts, whereas the 65–80 kDa band became more evident (Fig. 1e, f). Bafilomycin A1, but not MG132, treatment resulted in CLN3 accumulation (Supplementary Fig. 2d, e), suggesting that CLN3 is mainly degraded through lysosome-mediated pathways. Notably, upon short doxycycline treatment, CLN3 exclusively localized to the Golgi apparatus (de novo synthesis), whereas the long doxycycline washout resulted in reduction of the protein at Golgi and appearance of the protein on lysosomes, indicating that the newly synthesized protein transits through the Golgi to reach the lysosomes (Supplementary Fig. 2f).

These data indicate that the 45 kDa and 65–80 kDa CLN3 protein forms represent the newly synthesized and the mature form respectively, with the latter undergoing heavy glycosylation and trafficking through the Golgi and lysosomal compartments.

### CLN3 interacts with protein complexes involved in membrane trafficking and recycling

To identify CLN3 protein interactors that mediate important aspects of CLN3 function, we performed CLN3 interactome analysis by co-IP and mass spectrometry (Co-IP-MS) in CLN3-innHA ARPE19 cells upon a 24 h doxycycline induction, in both fed and prolonged starvation conditions. This analysis revealed that CLN3 interacts with several proteins and protein complexes involved in endo-lysosomal trafficking pathways such as BORC (e.g., Lyspersin, Snapin), BLOC1 (e.g., Disbindin), ESCPE-1 (e.g., SNX5, SNX6), retromer (e.g., VPS26), SNARE (e.g., VAMP3, VAMP7), PI4K2A and other PI-related proteins, and the cation independent-mannose 6-phosphate receptor (CI-M6PR) (Fig. 2; Supplementary Data 1). Also, we noted that prolonged starvation increased the number of CLN3 interactors. Specifically, we identified 107 CLN3 interactors in fed cells and 158 interactors in starvation conditions (fold change ≥ 1.5, p value < 0.05), with 88 common interactors shared between the two sets (Supplementary Fig. 3a and Supplementary Data 1). Candidate interactors from these pathways were validated via Co-IP-immunoblot experiments (Supplementary Fig. 3b, c, d). These data revealed that CLN3 interacts with several protein complexes involved in lysosome motility, biogenesis of lysosome-related organelles, and transmembrane receptor recycling from the endosome, suggesting that CLN3 may play a role in lysosome biogenesis and vesicular trafficking.

### Loss of CLN3 leads to mis-trafficking of CI-M6PR

CLN3 interactome analysis revealed that CI-M6PR, which is involved in the sorting of lysosomal enzymes and is recycled to the TGN compartment by retrograde trafficking machineries from sorting endosomes[23–27], was among the most enriched CLN3 interactors. We observed that the levels of CI-M6PR were highly reduced in ARPE19 CLN3-depleted cells and recovered upon bafilomycin treatment (Fig. 3a, b, Supplementary Fig. 4a). Inhibition of lysosomal degradation through bafilomycin also revealed that the CI-M6PR accumulated into the lysosomal lumen, indicating mis-trafficking consequent degradation (Fig. 3c). To follow the trafficking of CI-M6PR, we performed a cell-

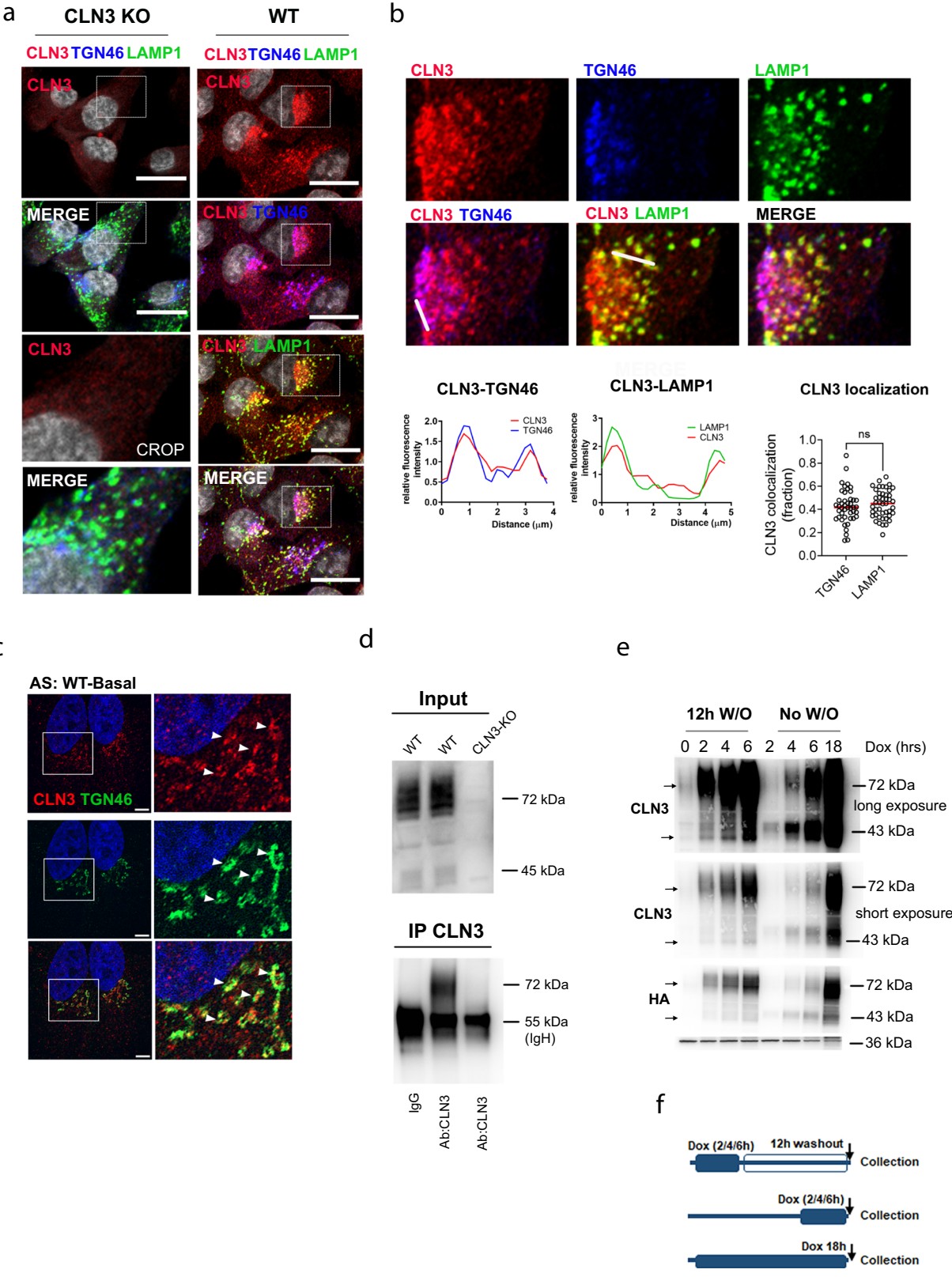

surface uptake assay, where cells are incubated with an antibody against the extracellular/luminal domain of CI-M6PR. After the uptake of the CI-M6PR, a portion of the protein is delivered from the plasma membrane (PM) to sorting endosomes and subsequently to the Golgi compartment, while a fraction is recycled back to the PM[28]. This assay showed that the amount of internalized receptor is also highly reduced in CLN3-KO cells (Fig. 3d), confirming the previous data. Flow cytometry (FC) also showed a reduction of CI-M6PR at the PM (Fig. 3e). During endocytosis, the internalized receptor accumulated in enlarged EEA1-positive early endosomes in CLN3 KO cells (Fig. 4a), whereas its Golgi delivery was impaired (Fig. 4b). No defects in the recycling of the receptor to the PM were detectable (Supplementary Fig. 4b, c).

**Fig. 1 | CLN3 protein localizes to lysosomes and Golgi. a** ARPE19 WT and CLN3-KO cells stained with antibodies against CLN3 (red), TGN46 (blue) and LAMP1 (green), analyzed by confocal microscopy. Scale bar 20 µm. Insets show image enlargements. **b** Enlargement of WT cells shown in (**a**), and fluorescent line intensity plot showing CLN3-LAMP1 and CLN3-TGN46 co-localization. White lines show analyzed areas. CLN3-TGN46 and CLN3-LAMP1 Manders' co-localization coefficients and mean+single values are also shown. N = 45 cells, unpaired *t*-test (two-tailed), *P* value 0.4977 (NS). **c** ARPE19 cells stained with antibodies against CLN3 (red) and TGN46 (green) and imaged with Airyscan super-resolution microscopy. Scale bar 5 µm

**d** CLN3- immunoprecipitates were prepared from WT and CLN3-KO ARPE19 cells and analyzed by immunoblotting for the indicated proteins. Repeated two times. IgH, IgG heavy-chains. **e** Immunoblot of ARPE19 cells infected with pLVX-CLN3innHA and induced for 0, 4, 6, or 18 h with doxycycline followed by a 12 h doxycycline washout, or collected immediately after induction. Arrows indicate 45 kDa and 65–80 kDa bands. W/O washout, Dox doxycycline, hrs hours. Repeated two times. **f** Scheme of doxycycline protocol used to study CLN3 protein maturation. Source data are provided as a Source Data file.

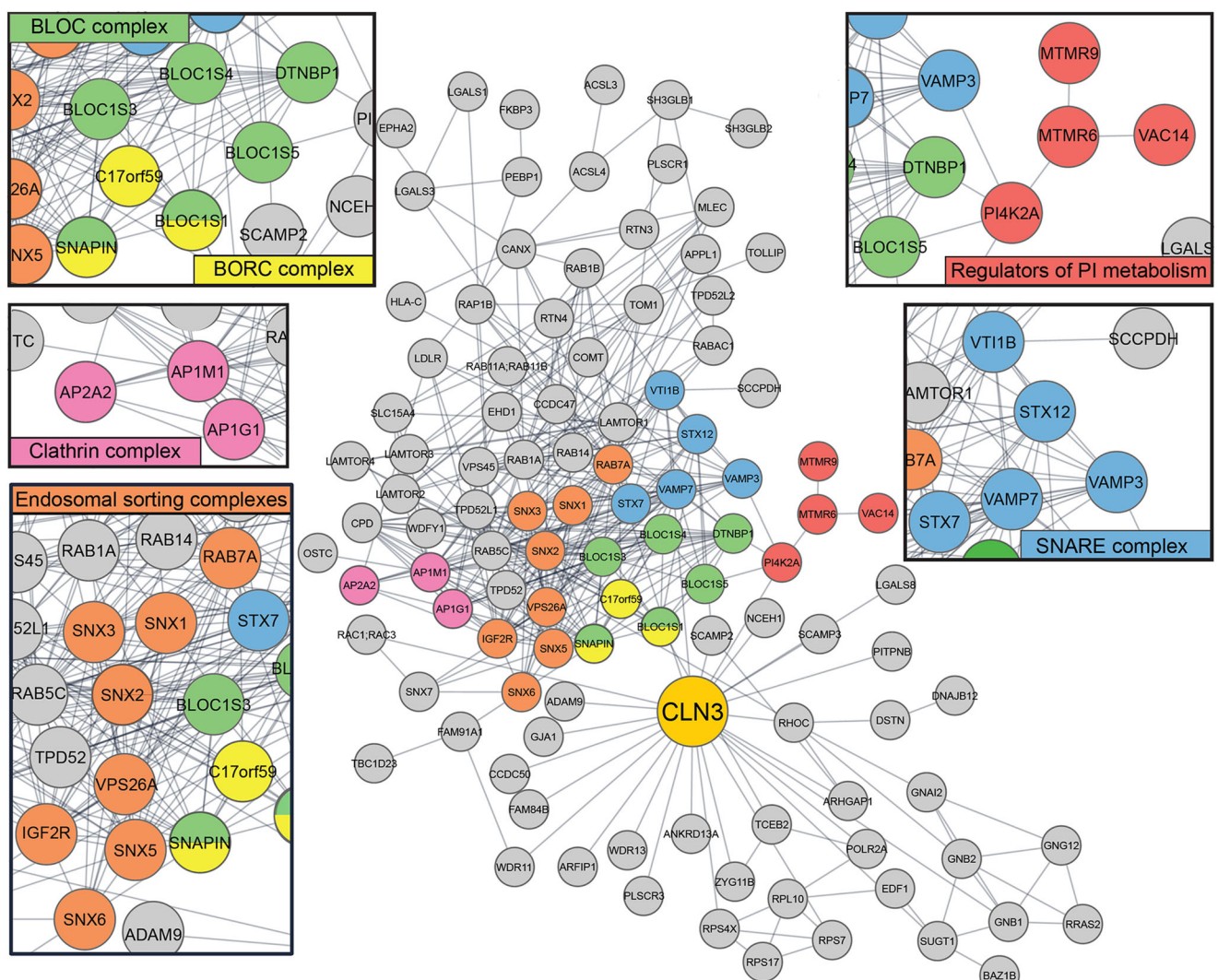

**Fig. 2 | CLN3 interacts with endo-lysosomal trafficking and recycling complexes.** Interactome analysis in ARPE19-pLVX-CLN3innHA cells induced for 24 h with 1 µg/ml doxycycline, and immunoprecipitated for the HA-tag, showing enriched hits (fold change ≥ 1.5, *p* value < 0.05) compared to cells not expressing the HA

tag. Relevant protein complexes identified with Corum and displayed with Cytoscape, are color-coded and reported in separate enlargements. N = three independent experiments. Data are provided in Supplementary Data 1. *P* values are calculated using two-tailed unpaired *t*-tests.

Conversely, CLN3 overexpression enhanced trafficking of the CI-M6PR to the Golgi compartment (Fig. 4a–c). Delivery of the CI-M6PR to the Golgi compartment is mainly modulated by the retromer and ESCPE-1 complex[29,30]. Upon CLN3-overexpression, the enhanced Golgi delivery of the receptor was partially rescued by silencing of the ESCPE-1 complex, but was mostly unaffected by retromer depletion (Fig. 4d). Of note, several components of the ESCPE-1 complex were also identified among CLN3 interactors, indicating that it may cooperate with CLN3 in the recycling of the receptor at Golgi. Impaired Golgi recycling of CI-M6PR was rescued by CLN3 reintroduction (Supplementary Fig. 4d) and was also detected in HeLa CLN3 KO cells (Supplementary

Fig. 4e). These results indicate that CLN3 is required for the correct trafficking of CI-M6PR.

## Loss of CLN3 causes mis-sorting of lysosomal enzymes
Due to the role of CLN3 in the trafficking of CI-M6PR, its loss may result in mis-targeting of lysosomal enzymes. To test this hypothesis, we analyzed the lysosomal proteome composition of wild type and CLN3 KO cells. MS analysis of Lyso-IP[22]-purified lysosomes from CLN3 KO cells, revealed significant alterations of the lysosomal proteome. Specifically, we detected an intra-lysosomal reduction of 14 different lysosomal hydrolases and as well as of CI-M6PR (Fig. 5a,

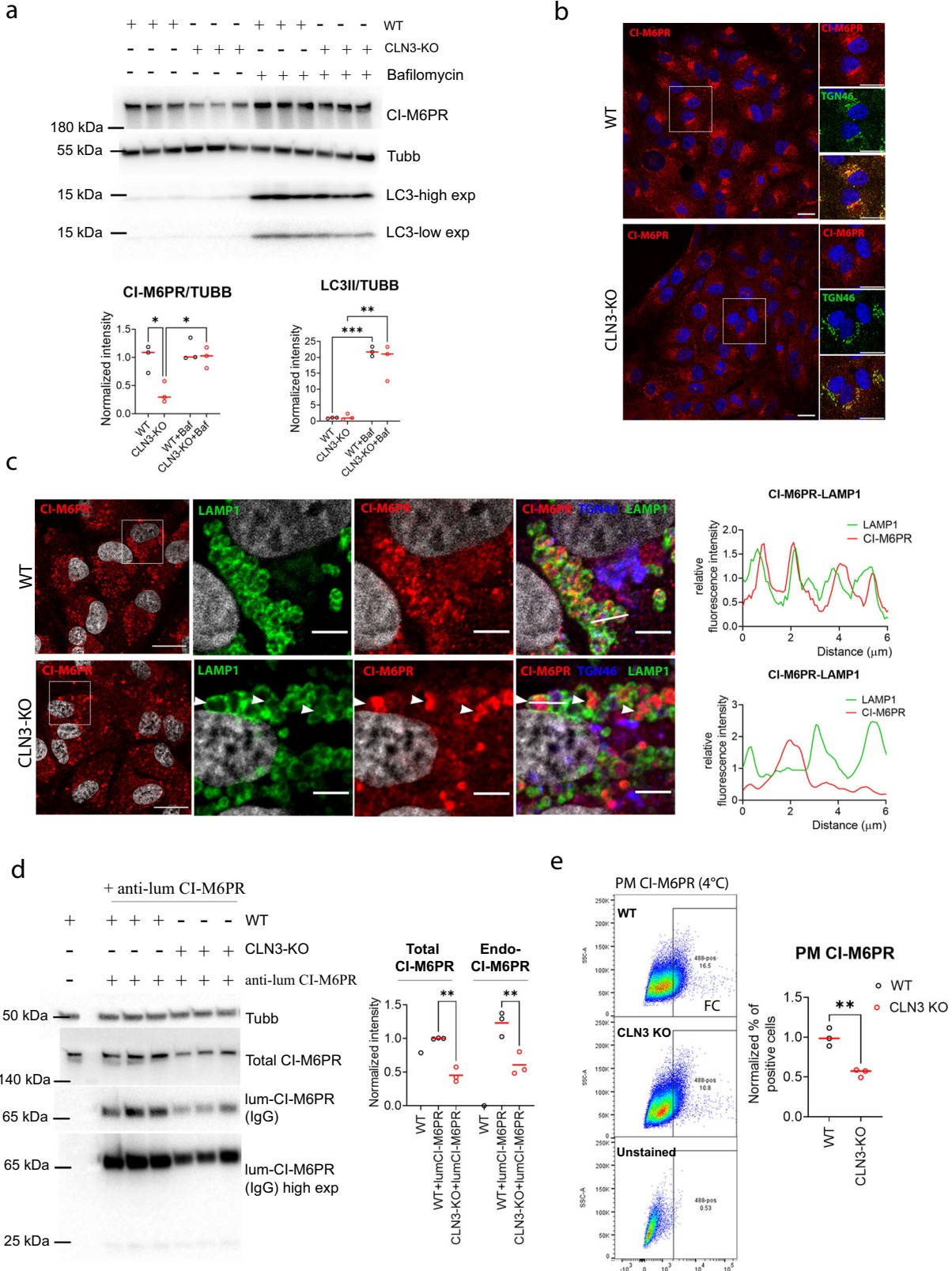

Supplementary Fig. 5a, b, c, Supplementary Data 2, 3). These results were validated by both immunoblot analysis (Supplementary Fig. 5d) and enzymatic assays (Supplementary Fig. 5e), which confirmed a reduction of GAA, HexA and GusB enzyme activities within the lysosomes of CLN3 KO cells. More specifically, we observed a substantial reduction of the intermediate and mature forms of the lysosomal enzyme alpha-glucosidase (GAA) in CLN3 KO cells, suggesting reduced intra-lysosomal maturation of the enzyme (Fig. 5b). GAA precursor levels were highly increased in the media of CLN3 KO cells, which is in line with a secretion of the neo-synthesized enzyme into the media (Fig. 5c). Similar results were obtained for the beta-glucuronidase (GusB) enzyme, whose mature form was reduced in the pellet

**Fig. 3 | CI-M6PR is degraded in CLN3-KO cells. a** ARPE19 WT and CLN3-KO cells were treated with 20 nM bafilomycin for 12 h and analyzed by immunoblotting with the indicated antibodies. Each lane is an independent replicate. The relative quantifications are shown in graph, values are normalized against b-tubulin (Tubb) and WT cells. N = three independent experiments. Unpaired *t*-test (Tukey's multiple comparisons test) (**p* < 0.05, ***p* < 0.01, ****p* < 0.001). **b** ARPE19 WT and CLN3-KO cells stained with antibodies against CI-M6PR and TGN46, analyzed by confocal microscopy. Scale bar 20 μm (left), and 5 μm (right, insets). **c** ARPE19 WT and CLN3-KO cells were treated with 20 nM bafilomycin for 12 h and stained with antibodies against CI-M6PR, TGN46 and LAMP1. Overlapped CI-M6PR-LAMP1 fluorescent line intensity plots are shown. Scale bar 20 μm (left), and 5 μm (right, insets). **d** ARPE19 WT and CLN3-KO cells were loaded with saturating concentrations (5 ug/ml) of a

monoclonal anti-CI-M6PR antibody recognizing the extracellular/luminal domain (lum-CI-M6PR) at 37 °C for 60 min and were analyzed by immunoblot with antibodies against the indicated proteins. To detect lum-CI-M6PR, only the relative secondary antibody was blotted. Each lane is an independent replicate. Relative quantifications are shown in graph, values are normalized against b-tubulin (Tubb) and WT+lum-CI-M6PR condition. N = three independent experiments. One-way Anova (***p* < 0.01). **e** Representative flow cytometry dot plots from ARPE19 WT and CLN3 KO cells, loaded with 5ug/ml of anti-lum-CI-M6PR at 4 °C for 45 min, and stained for PM CI-M6PR, or unstained (lower plot). Selected positive populations and their relative quantification are shown (N = three independent experiments, unpaired *t*-test (two-tailed), mean ± SEM, ***p* < 0.01). The gating strategy is provided in Supplementary Fig. 9. Source data are provided as a Source Data file.

(Supplementary Fig. 6a) and the precursor form increased in the medium of CLN3 KO cells (Supplementary Fig. 6b). Consistent with these results, enzymatic assays performed on the supernatant and cell lysates of CLN3 KO cells revealed increased secretion of GAA, GusB and HexA enzymes in the media, and a reduction of their cellular content (Supplementary Fig. 6c).

We also found that rhGAA PM-to-lysosome trafficking was impaired in CLN3 KO cells, in which GAA accumulated in endosomes (Fig. 5d). Furthermore, lysosomal degradation of rhGAA was severely impaired in CLN3 KO cells, whereas it was enhanced after a 30 min-chase in CLN3-overexpressing cells (Fig. 5e). Finally, live-imaging of cells loaded with rhGAA-546 after prolonged starvation revealed that most of the enzyme failed to reach lysosomes after a 60 min-loading in CLN3 KO cells (Supplementary Fig. 6d), confirming our previous results.

Together these results are consistent with the above-described trafficking defect of CI-M6PR, resulting in mis-sorting of lysosomal proteins and leading to global impairment of lysosomal degradative capacity.

## CLN3 modulates lysosomal tubulation and reformation

To investigate the effects of CLN3 depletion on lysosomes, we performed ultrastructural analysis on CLN3 KO cells. This analysis revealed the presence of enlarged and aggregated lysosomes and accumulation of autolysosomes (Fig. 6a, Supplementary Fig. 7a), a phenotype that was exacerbated when the cell cycle was blocked (Fig. 6b), suggesting an impairment of a lysosomal biogenesis process known as ALR[14–16]. ALR is induced upon prolonged starvation (i.e., serum + glutamine deprivation)[15], during which lysosomal degradation of autophagic cargos results in reactivation of the mTOR pathway, a critical step for ALR[15]. Notably, we observed that mTOR signalling was insensitive to prolonged starvation in CLN3-KO cells, and its reactivation was defective, possibly because of the impaired lysosomal degradation capacity (Fig. 6c). Moreover, after prolonged starvation CLN3-depleted cells showed accumulation of enlarged LC3-GFP⁺ autolysosomes (Fig. 6d), supporting an impairment of ALR. We also observed that CLN3 reintroduction in CLN3-KO cells, not only rescued lysosomal storage (Supplementary Fig. 7b), but also induced extensive lysosome tubulation, an important step in the autophagic-lysosomal reformation (ALR) process[14–16] (Fig. 6e). Moreover, in CLN3-rescued cells lysosomes were increased in number, and reduced in size, in line with the activation of ALR (Fig. 6e, Supplementary Fig. 7c).

Induction of ALR by prolonged starvation in CLN3-*inn*HA overexpressing cells revealed remodeling of the lysosomal compartment, with the appearance of multiple tubules on lysosomes which bud, elongate and are then released (Fig. 7a, b; Supplementary Movie 1). These tubules were positive for the markers dextran, LAMP1 and CLN3 (Fig. 7a, b; Supplementary Fig. 8a–c) and closely resembled tubules observed during ALR[14–16,31]. Induction of CLN3-*inn*HA expression in both basal and prolonged starvation conditions also resulted in a striking increase in the total number of lysosomes per cell compared with control cells, especially in the prolonged starvation conditions

(Fig. 7b, Supplementary Fig. 8d). These data indicate that ALR is enhanced upon CLN3 overexpression. In line with these results, lysosomal tubules were nearly absent in CLN3 KO cells and no significant difference in the number of lysosomes was detected between starved and fed CLN3 KO cells (Fig. 7b, Supplementary Fig. 8d). Furthermore, prolonged starvation failed to restore normal lysosomal size in CLN3 KO cells (Fig. 7b; Supplementary Movie 2, 3).

To test whether the tubulation phenotype was dependent upon autophagy induction, we first we co-expressed LAMP1-mCherry and LC3-GFP and observed LAMP1⁺ tubules extending from autolysosomes (LAMP1⁺, LC3⁺) upon 16 h of starvation (Supplementary Fig. 8e). Next, we silenced the ATG7 protein, an essential autophagy effector enzyme, and then assessed lysosomal tubulation and reformation rate in CLN3 overexpressing cells upon prolonged starvation. Silencing of ATG7 completely abolished lysosomal tubules, reduced lysosomal number and increased lysosomal size, confirming that the observed lysosomal reformation process is autophagy-dependent (Fig. 7c, Supplementary Fig. 8f). Together, these data indicate that CLN3 drives lysosomal tubule formation, which is required for the generation of new lysosomes through ALR.

We then reasoned that CLN3-mediated lysosomal tubulation and reformation function may be connected to its role on CI-M6PR and on the delivery of lysosomal enzymes. We performed lysosomal immunopurification analysis, and confirmed that in CLN3-overexpressing cells, not only lysosome numbers were increased, but also the levels of lysosomal enzymes were higher, indicating that newly-formed lysosomes contained lysosomal enzymes (Fig. 8a). In addition, new lysosomes and tubules were positive for rhGAA and pepstatin A (PepA), confirming that CLN3 was able to promote delivery of lysosomal enzymes in the new lysosomes (Fig. 8b). Finally, we found that silencing of CI-M6PR completely abolished CLN3-mediated lysosomal tubulation and reformation phenotypes (Fig. 8c, Supplementary Fig. 8g, h). These data indicate that CLN3-mediated hypertubulation requires the presence of CI-M6PR, and of key proteins in ALR such as the autophagy regulator ATG7.

## Discussion

Batten disease, caused by mutations in *CLN3*, is one of the most devastating neurodegenerative childhood diseases. Its progressive nature and fatal outcome represent a heavy burden for the families of affected children and for society. CLN3 has been linked to a variety of different processes, such as regulation of lysosomal pH[32,33], autophagy[34,35], calcium homeostasis[36,37], cell proliferation[38,39] and synaptic activity[12], and it was recently shown to be essential for the clearance of glycerophosphodiesters from lysosomes[40]. More recently, attention has shifted to the ability of CLN3 to modulate endocytic pathways and membrane dynamics[41–43], suggesting that this protein may exert a global role on lysosomal function. Interestingly, Schmidtke et al.[43] recently described a general loss of lysosomal enzymes and defective trafficking of Tf and TfR in a cerebellar murine NCL3 cell line, while Yasa et al.[42] reported a role for CLN3 in regulating retromer-RAB7A function. These results suggest a role of CLN3 in the

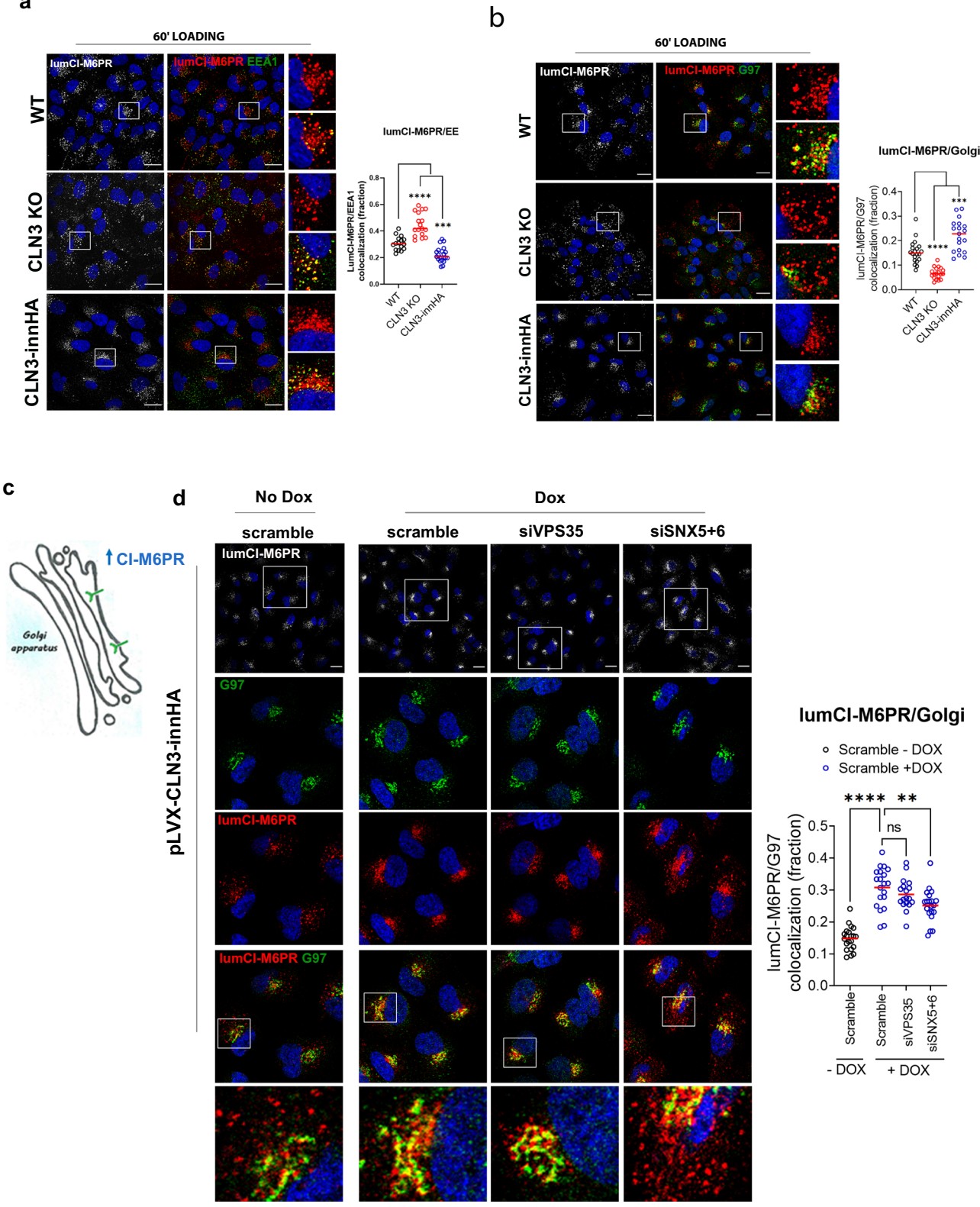

regulation of endo-lysosomal trafficking and sorting of lysosomal enzymes.

Thanks to the generation of a novel CLN3 antibody we were able to observe that CLN3 localization is not limited to lysosomes, as previously indicated, but a significant fraction of CLN3 is also found at the Golgi apparatus. More specifically, we observed a Golgi-specific 45 kDa form, which represents the newly synthesized protein, and a mature, highly glycosylated, 65–80 kDa form located both at the Golgi and

lysosomes. We also observed that CLN3 Golgi localization, which was visible at early times of induction, was significantly reduced upon inhibition of protein biosynthesis (CHX treatment). These results strongly suggest that CLN3 is not a Golgi resident protein, but that the newly synthesized protein transits through the Golgi to reach the lysosomes.

Moreover, proteomic analyses showed that CLN3 interacts with several endo-lysosomal trafficking complexes, such as BORC, BLOC1,

**Fig. 4 | CLN3 regulates the recycling of CI-M6PR. a** Representative confocal images of ARPE19 cells loaded with 5ug/ml anti-lumCI-M6PR antibody, at 37 °C for 60 min (lumCI-M6PR, red), and co-stained with EEA1 (green) (scale bar 20 μm). Insets show enlargement areas. Graph shows lumCI-M6PR-EEA1 Manders' colocalization coefficient ($N = 16–21$ areas, mean + single values, three independent replicates). One-way Anova (Tukey's multiple comparisons test), (***$p < 0.001$, ****$p < 0.0001$). **b** Representative confocal images of ARPE19 cells loaded with lumCI-M6PR (red), as described in (**a**), and co-stained with G97 (green) (scale bar 20 μm). Insets show enlargement areas. Quantification shows lumCI-M6PR-G97 Manders' colocalization coefficient ($N = 16–21$ areas, mean+single values, three

independent replicates) one-way Anova (***$p < 0.001$, ****$p < 0.0001$). **c** Schematic representation of lumCI-M6PR localization upon CLN3-overexpression. **d** ARPE CLN3innHA cells were transfected with SNX5 + 6, VPS35 or scramble siRNA for 72 h, and the day after induced with 1 μg/ml doxycycline for the remaining 40 h, loaded with 5ug/ml anti-lumCI-M6PR antibody (red), at 37 °C for 45 min, and then chased for 30 min. Cells were co-stained with G97 (green) and analyzed by confocal microscopy (scale bar 20 μm). Manders' colocalization coefficients are shown. Results are means+single values, $N = 18–21$ areas, four independent experiments, one-way Anova. Scale bar 10 μm. (**$p < 0.01$, ****$p < 0.0001$). Source data are provided as a Source Data file.

retromer and ESCPE-1, as well as with SNAREs (e.g., VAMP3, VAMP7) proteins, clathrin adaptors, PI4K2A, a regulator of phosphoinositide metabolism, and CI-M6PR, an important sorting receptor of lysosomal hydrolases[13,44]. Notably, CI-M6PR was among the most enriched CLN3 interactors. Most of lysosomal enzymes acquire mannose-6-phosphate residues, which are then recognized by specific sorting receptors (mannose-6-phosphate receptors, MPRs) and sorted into the endo-lysosomal compartment[13,44,45]. We found that loss of CLN3 resulted in mis-sorting of CI-M6PR to the lysosomes, with its consequent degradation.

Considering the role CI-M6PR, we postulated that CLN3 depletion would result in a global depletion of lysosomal hydrolases in the lysosomal lumen. In line with this hypothesis, proteomic analysis performed on purified lysosomes revealed a significant reduction of intra-lysosomal levels of several hydrolases. Schmidtke et al.[43], and by Metcalf et al.[46] also reported defects in the trafficking of lysosomal enzymes in CLN3 KO cells. These data are consistent with an impaired sorting of lysosomal enzymes due to defective recycling of CI-M6PR. We also observed that GAA protein levels and enzymatic activity were reduced in CLN3 KO cells, and increased in the media, together with other lysosomal enzymes such as HexA. Notably, a similar mechanism was reported in two other neurodegenerative LSDs caused by mutations affecting the MPR-sorting pathway: mucolipidosis II or Inclusion-cell (I-cell) disease, and mucolipidosis III (pseudo-Hurler polydystrophy Inclusion-cell)[47]. In these conditions, lysosomal enzymes cannot enter the lysosomal delivery route, instead they are secreted extracellularly, similarly to what we observed in CLN3 KO cells. Similar results were also reported by Pechincha et al.[48], who recently showed that depletion of LYSET, a core component of the M6P pathway, caused depletion of several lysosomal enzymes from the lysosomes, together with their consequent secretion in the media. Notably, all 14 enzymes that we found to be reduced in CLN3-KO lysosomes, were also reduced in lysosomes isolated from LYSET-depleted cells, confirming that our results are in line with a defect in the M6P pathway (Table 1).

Finally, we found that GAA maturation, and rhGAA endosome-to-lysosome delivery and lysosomal degradation was impaired in CLN3 KO cells, while CLN3-overexpressing cells displayed enhanced lysosomal degradation of the rhGAA-internalized enzyme, which is in line with the enhanced recycling of CI-M6PR observed in CLN3-overexpressing cells.

We reasoned that depletion of CI-M6PR and of lysosomal enzymes, may affect lysosomal function. In the absence of CLN3, electron microscopy revealed that cells accumulated enlarged and aggregated lysosomes, which is a feature of impaired ALR. Also, cells were unable to reactivate mTOR activity upon prolonged starvation and showed accumulation of enlarged LC3-GFP+ autolysosomes 16 h post serum+glutamine starvation. Both of these features resemble defects observed during failure of ALR[15–17]. Consistently, we found that tubules were almost completely absent in CLN3 KO cells, in both fed and prolonged starvation conditions. Rong et al.[17] previously showed that during prolonged starvation a portion of lysosomes is initially consumed until they are restored by ALR upon starvation-induced autophagy. We quantified the number of lysosomes during starvation

and observed that in CLN3-KO cells it remained stable over time, unlike in WT cells. These data suggest that loss of CLN3 impairs both lysosomal consumption during the initial phase of starvation and lysosomal reformation during prolonged starvation.

Oppositely from CLN3 depletion, we observed that CLN3 overexpression positively modulates ALR. CLN3 overexpression resulted in tubule formation and in a substantial increase in the number of lysosomes, especially upon prolonged starvation, and a reduction of lysosomal size. Specifically, the number of lysosomes was significantly decreased after 8 h of starvation and recovered after 16 h to levels that are significantly higher compared to WT cells. We also determined that inhibition of autophagy through ATG7 silencing completely abolished CLN3-dependent tubules, confirming that this process is autophagy-dependent. Interestingly, newly formed proto-lysosomes contained lysosomal enzymes, as shown by lysoIP data, and tubules were positive for rhGAA, or pepstatinA, supporting the hypothesis that CLN3 may modulate ALR by promoting the delivery of lysosomal enzymes. Consistently, we observed that CI-M6PR is essential for CLN3-mediated ALR, as its silencing completely abolished lysosomal tubulation and reformation.

Other interactors identified in our proteomic analysis may also play a role in the lysosomal reformation process. Both Lyspersin, a BORC subunit essential for lysosomal positioning and motility[49], and PI4K2A, a member of the family of phosphatidylinositol (PtdIns) 4-kinases responsible for phosphatidylinositol 4-monophosphate (PI4P) production[50,51], were also highly enriched among CLN3 interactors. The BORC subunit BLOC1S1 was recently shown to be important in the initiation of the lysosomal tubulation process[52]. Also, mutations of Borcs7, a subunit of the BORC complex, result in a neurodegenerative phenotype associated with progressive axonal dystrophy and perinatal death in mice[53]. Furthermore, PI4K2A was shown to play a role in ALR in hereditary spastic paraplegia (HSP), a group of neurodegenerative diseases associated with severe motor decline[54,55]. The specific role of the interaction of CLN3 with Lyspersin and PI4K2A will be the subject of future studies.

Notably, a recent study showed that loss of function of CLN7, another lysosomal transmembrane protein involved in NCL, resulted in both depletion of multiple soluble lysosomal proteins and impaired lysosome reformation[56]. Also in this case, the impaired degradative capacity of lysosomes abolished mTOR reactivation and ALR. These findings indicate that ALR impairment and global alteration of the lysosomal degradative capacity may be a pathogenic mechanism common to other NCLs and suggest that CLN3 and CLN7 have similar roles in intracellular trafficking. In summary, we discovered that CLN3 plays a crucial role in the sorting of CI-M6PR and links this process with ALR, thus explaining the global lysosomal dysfunction observed in Batten disease (Fig. 8d). Hopefully, future studies based on these data will point to new therapeutic strategies for this devastating disease.

## Methods

### Cell cultures

Cells were cultured in the following media: HeLa and HEK 293 T in DMEM (Cat# 16777-200, VWR), ARPE19 in DMEM-F12 (Cat# 11320082,

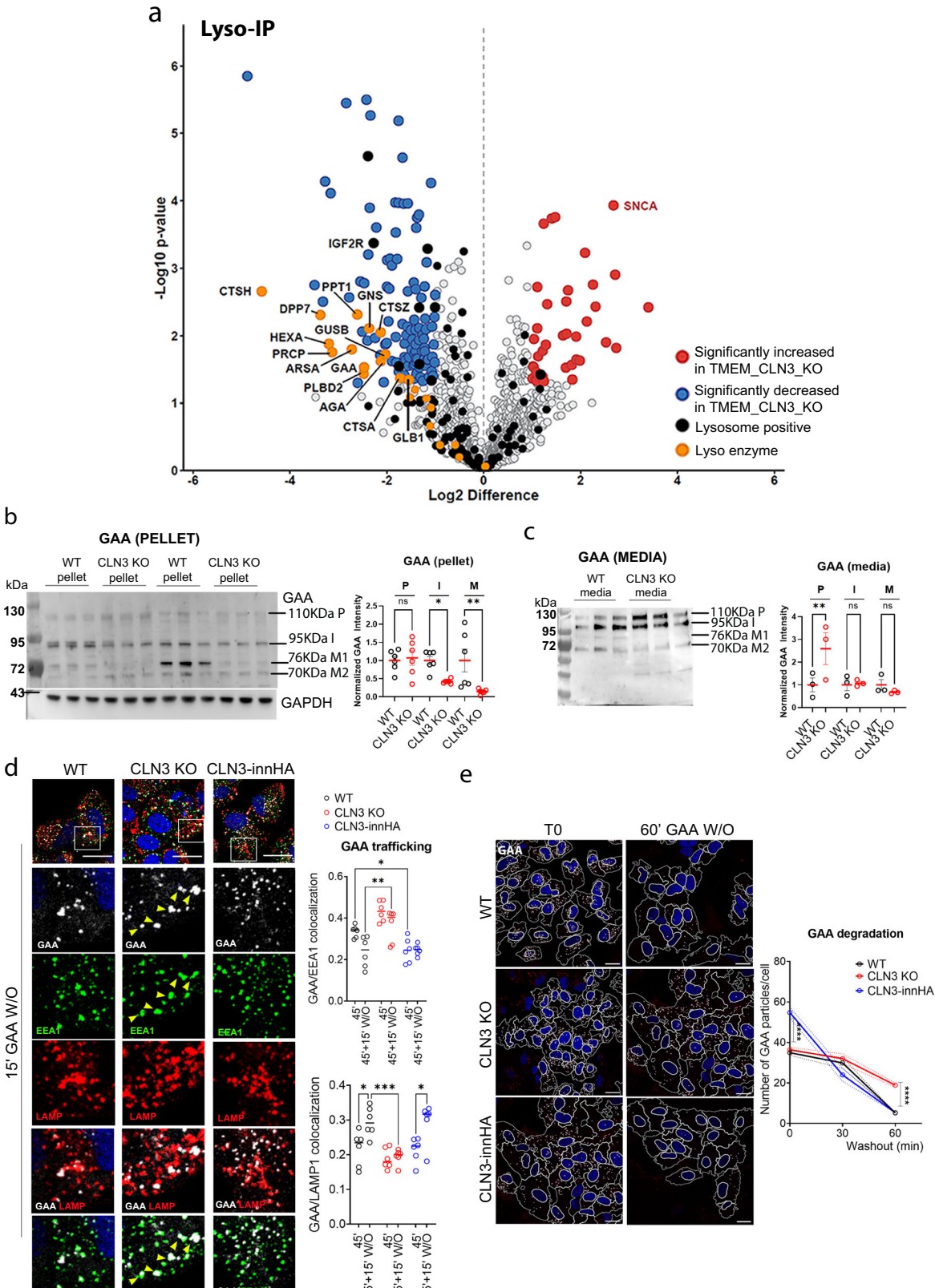

Thermo Fisher Scientific) and DMEM/F-12, no glutamine (Cat# 21331-020, Gibco). All media were supplemented with 10% inactivated FBS (tetracycline-free) (Cat# 631106, Clontech/Takara), 2 mM glutamine (Cat# 25030081, ThermoFisher), penicillin (100 IU/mL) and streptomycin (100 µg/mL) (Cat# 10378016, ThermoFisher) and maintained at 37 °C and 5% CO2. Stable Tet-On ARPE19 CLN3-innHA, CLN3 and HeLa CLN3-innHA cell lines were generated by transducing cells with the lenti-pLVX Tet-One Inducible Expression System (Clontech). All cell lines were validated by morphological analysis and routinely tested for absence of mycoplasma.

**Fig. 5 | CLN3 depletion causes mis-sorting of lysosomal enzymes. a** Volcano plots of the lysosomal proteome from ARPE19 WT and CLN3 KO cells. Lysosomal immuno-purification was performed as reported in Supplementary Fig. 2b. Volcano plots highlight proteins significantly depleted (blue dots, fold change ≤−1, $p$ value < 0.05) and enriched (red dots, fold change ≥1, $p$ value > 0.05) compared to WT cells. Cells without HA expression were used as negative control. Lysosomal enzymes are depicted in yellow. $N = 4$ independent experiments. Data are provided in Supplementary Data 2. $P$ values are calculated using two-tailed unpaired $t$-tests. **b**, **c** Immunoblot analysis of GAA protein maturation in WT and CLN3 KO ARPE19 cells (**b**, pellet, $N = 6$ independent replicates; **c** media $N = 3$ independent replicates); mean ± SEM, one-way Anova (Fisher's LSD test), with relative quantifications (*$p < 0.05$, **$p < 0.01$). P precursor, I intermediate, M1 mature 1, M2 mature 2. Graphs report the levels of precursor, intermediate, and mature proteins (expressed as sum of M1 + M2), respectively normalized against GAPDH and WT samples.

**d** Representative confocal images of ARPE WT, CLN3 KO and CLN3innHA cells (induced with 1 µg/ml doxycycline for 40 h) loaded for 45 min at 37 °C with 40 µg/ml rhGAA-546 and then chased for 15 min. RhGAA particles are shown in white. Cells were then co-stained for EEA1 (green) and LAMP1 (red) and analyzed by confocal microscopy. Manders' colocalization coefficients of internalized rhGAA with EEA1 and LAMP1 are shown. Results are means + single values, three independent experiments, 45′ loading $N = 14$ areas, 45′ + 15′ W/O $N = 12$ areas, two-way Anova (*$p < 0.05$, **$p < 0.01$, ***$p < 0.001$). Scale bar 20 µm. **e** Representative confocal images of GAA degradation, in cells treated as described in (**a**), and chased for 30 and 60 min. Images were generated with the Imaris software, white dots represent rhGAA particles. Graphs are relative to the number of GAA particles per cell, over time (T0 $N = 13$–14 areas, T30 $N = 12$–15 areas, T60 = 10–14 areas, mean ± SEM, three independent replicates, two-way Anova, ****$p < 0.0001$). Scale bar 10 µm. Source data are provided as a Source Data file.

## Generation of CLN3-KO CRISPr/Cas9 cell lines

ARPE19 cells full KO for CLN3 gene was generated using the CRISPr/Cas9 system. A gRNA sequence with a low off-target score was selected by using the http://crispor.tefor.net/crispor.pv online tool. The following guide was selected: GGCGCGCATTGGAAGAACG. The "ALL in One" vector containing each gRNA was obtained from SIGMA (CAS9GFPP). Cells were electroporated using the Amaxa system with the nucleofection kit (Cat.# VCA-1003, Lonza). GFP-positive cells were FACS sorted into 96-well plates to obtain single-cell derived colonies carrying the INDEL mutations. Upon genomic DNA extraction, the genomic sequence containing the targeted region was amplified by PCR reaction with the specific primers: CLN3KOup tgaggggaatgagagctgac, CLN3KOlow cggtcacttccctcttctca. PCR products were analyzed by DNA Sanger sequencing and cell clones carrying homozygous mutations introducing a premature stop codon (c. 109delA) were selected and expanded. Clones were grown to confluency and screened for knockout deletion by western blotting and immunostaining. Clone 26 and clone 21 were selected, and clone 26 was used for the reported experiments. HeLa cells were seeded 1 day prior to transfection, then transiently transfected with CRISPR plasmids encoding the specific gRNA guide against CLN3 (GGCGCGCATTGGAAGAACG), the Cas9 enzyme and a puromycin-resistance marker using FuGENE® 6 (Promega). Transfected cells were selected by incubation with 1 µg/ml puromycin for 24 h. Following puromycin selection, cells were seeded into a 96-well plate at a density of 1 cell per well. Clones were grown to confluency and screened for knockout deletion by western blotting and immunostaining.

## Plasmids and generation of cells stably expressing cDNAs

pLVX-TeT-ON CLN3, CLN3-innHA, LAMP1-mCherry inducible lentiviral vectors were generated by standard cloning using the In-fusion HD cloning kit (#638920, Takara). pLJC5-TMEM192-3xHA plasmid was obtained on Addgene (Cat# 104434). Lentiviruses were produced by co-transfecting HEK293T cells with the plasmids indicated above in combination with VSV-G and ΔVPR packaging plasmids. Lipofectamine LTX (Invitrogen) was used as transfection reagent. Twelve hours post transfection, medium was changed to DMEM supplemented with 20% FBS. Forty-eight hours later, virus-containing supernatants were collected, passed through a 0.45 µm filter to eliminate cell debris and used for infection in the presence of 8 µg/ml polybrene (Cat# TR-1003-G, EMD Millipore). Twenty-four hours later, cells were selected with puromycin or blasticidin for selection. The pEGFP-N1-PI4K2A was previously published[57]. PeT28A and pGEX-4T1-CLN3 plasmids were generated by standard cloning using the In-fusion HD cloning kit (#638920, Takara).

## Reagents and antibodies

The homemade polyclonal anti-CLN3 antibody was generated by immunizing rabbits with a purified *Escherichia Coli* expressed His-

tagged CLN3 polypeptide, containing both luminal and cytoplasmic protein domains.

Reagents used in this study were obtained from the following sources: HA (Biolegend, Cat# 902301; Cat# 901501, 1:400 dilution for IF, 1:1000 for WB), LAMP1 (Santa Cruz, Cat# sc-20011, 1:200 dilution for IF, 1:1000 dilution for WB), GM130 (abcam, ab52649, 1:400 dilution for IF), TGN46 (Bio-Rad, AHP500GT, 1:400 dilution for IF), LAMP1 (Hybridoma Bank, Cat# H4A3-a, 1:500 dilution for immunoEM), LC3 (Novus, Cat#NB100-2220 1:1000 dilution for WB), p62 (Novus 2C11, Cat# H00008878-M01, 1:1000 dilution for WB), NBR1 (Abnova, cat# H00004077-M01, 1:1000 dilution for WB), ATP synthase C (Abcam, Cat# ab181243, 1:1000 dilution for WB), PI4K2A (Santa Cruz, Cat# sc-390026, 1:1000 dilution for WB), Pallidin (Cat# 10891-2-AP, Proteintech, 1:500 dilution for WB), Dysbindin (Novus, Cat# NBP2-16245, 1:1000 dilution for WB), Rab7 (CellSignaling, Cat# 9367 S, 1:1000 dilution for WB), VPS35 (Abcam, ab10099, 1:1000 dilution for WB), Lamtor1 (CellSignaling, Cat# 8975, 1:1000 dilution for WB), Lamtor2 (CellSignaling, Cat# 8145, 1:1000 dilution for WB), VAMP3 (Novus, NB300-510, 1:1000 dilution for WB), TfR (Thermo, Cat# 13-6890 1:1000 dilution for WB), Rab11 (Prointech, Cat# 15903-1-AP, 1:1000 dilution for WB), CI-M6PR (Abcam, Cat# ab32815, 1:1000 dilution for WB and Novus, 2G11 Cat# NB300-514SS, 1:200 dilution for IF), Cathepsin H (Santa Cruz, Cat# sc-398527, 1:1000 dilution for WB), Cathepsin X/Z/P (R&D, Cat# AF934, 1:1000 dilution for WB), Cathepsin D (abcam, Cat# ab75852, 1:1000 dilution for WB), DPP7 (Novus Biologicals, Cat# NBP132875, 1:1000 dilution for WB), HexA (Abcam, ab189865, 1:1000 dilution for WB), GAPDH (Santa Cruz, sc-32233 1:5000 dilution for WB). HRP-conjugated secondary antibodies to Mouse (Cat# 401215 − 1:6000 dilution) and Rabbit (Cat# 401315 − 1:6000 dilution). HA–Agarose (Sigma, Cat# A2095), HA-magnetic beads (Thermo, Cat# 88836), GFP (Cromotek, Cat# GTA-20), anti-rabbit IgG (Bethyl, Cat.# P120-101), ProteinA-Sepharose 4B Conjugate (Invitrogen, Cat#101041); Donkey anti-Rabbit IgG (H + L) Alexa Fluor 488 (Cat# A-21206 − 1:500 dilution), Alexa Fluor 568 (Cat# A-10042 −1:500 dilution), Donkey anti-mouse IgG (H + L) Alexa Fluor 568 (Cat# A- 10037 − 1:500 dilution), Alexa Fluor 647 (Cat# A-31571 − 1:500 dilution), Alexa Fluor 594 (Cat# A-21203 − 1:500 dilution), Donkey anti-goat IgG (H + L) Alexa Fluor 647 (Cat# A-21447 − 1:500 dilution), Donkey anti-sheep (H + L) Alexa Fluor 488 (Cat# A-11015 − 1:500 dilution) were from Thermo Fisher Scientific.

Chemicals: Protease Inhibitor Cocktail (Cat# P8340, Sigma), Puromycin (Cat# P9620, Sigma), PhosSTOP phosphatase inhibitor cocktail tablets (Cat# 04906837001, ThermoFisher), MG132 (Cat# M7449, Sigma), Bafilomycin (Cat# sml1661, Sigma), Cycloheximide (Cat# C1988-1G, Sigma), Doxycycline (Cat# D9891, Sigma), DSP (Cat# 22585, ThermoFisher), Dextran (Cat# D22910, ThermoFisher), Lysotraker (Cat# L7528, ThermoFisher), Blasticydin (Cat# A1113903, Fisher), Polybrene (Cat# TR-1003-G, EMD Millipore, Hydroxyurea (Cat #H8627, Sigma).

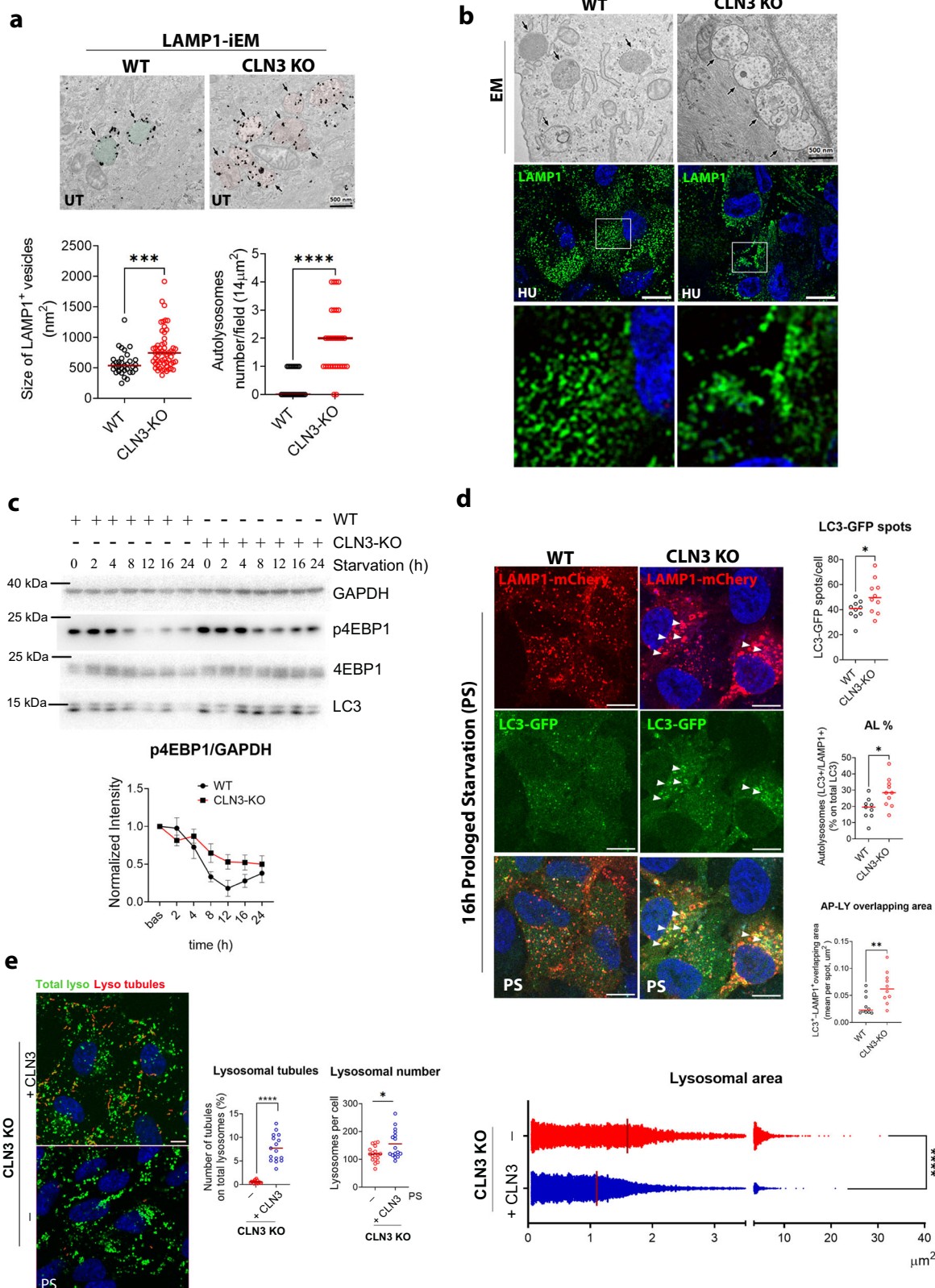

## Cell treatments and protein knockdown

Prolonged starvation was always performed by incubating cells in serum+glutamine starvation media (prolonged starvation media: DMEM/F-12, no glutamine, Cat# 21331-020 Gibco, supplemented with 1% pen/strept, Cat# 10378016, ThermoFisher).

All doxycycline experiments were performed treating cells with 1 μg/ml doxycycline (Sigma, Cat#D9891) for 40 h (unless differently stated).

To perform CLN3-HA IP-followed mass spec analyses, cells were induced with doxycycline for 24 h in complete media, and 16 h prior to

**Fig. 6 | CLN3 depletion causes accumulation of enlarged autolysosomes.**
**a** Ultrastructural analysis of lysosomal compartment. LAMP1 immuno-EM (iEM) images of ARPE19 cells showing normal lysosomes in WT cells (arrows, green), and aggregated lysosomes with undigested material in CLN3 KO cells (arrows, pink). Scale bar 500 nm. Relative quantifications are shown in graph. Size of LAMP1 vesicles, $N = 33$–57 areas, Autolysosomes number/field, $N = 26$–30 areas. Unpaired $t$-test (two-tailed) (***$p < 0.001$, ****$p < 0.0001$). **b** Transmission electron microscopy (EM, scale bar 500 nm) and confocal microscopy (Scale bar 20 μm, magnification of the outlined areas are shown at the bottom) images of ARPE19 cells upon block of cell cycle for 1 week, with 1 mM hydroxyurea (HU) treatment. Aggregated lysosomes in KO cells are indicated by arrows. Repeated two times. **c** ARPE19 WT and CLN3-KO cells were progressively starved (serum + glutamine) for the indicated time point and analyzed by immunoblot with the indicated antibodies. Mean ± SEM, four independent experiments. Quantifications are shown in graph. **d** pLVX-LAMP1-mCherry cells were induced with doxycycline for 40 h and transfected with LC3-GFP plasmid for 24 h. Cells were then starved for 16 h and analyzed by confocal microscopy. Quantifications shown in graph indicate the number of LC3-GFP spots per cell (top), the percentage of autolysosomes normalized on total lysosomes (middle), and the LC3 + LAMP1 overlapping area (bottom) ($N = 9$–10 areas, unpaired $t$-test, two-tailed, *$p < 0.05$, **$p < 0.01$). Scale bar 10 μm. **e** Representative live-image confocal snapshots of CLN3-KO+pLVX-CLN3-innHA with and without 40 h of 1 μg/ml doxycycline induction, in prolonged starvation (16 h) and loaded with 1 μg/μl dextran for 1 h the night prior imaging. Lysosomes (dextran, green) and lysosomal tubules (red). Scale bar 7 μm. Graphs show number of lysosomal tubules, normalized against the total number of lysosomes per field, lysosome number per cell, normalized against the total number of cells, and lysosomes area quantification, each spot is a single lysosome. $N = 90$–141 cells, three independent experiments, mean + single values are reported, unpaired $t$-test (two-tailed) (*$p < 0.05$, **$p < 0.01$, ****$p < 0.0001$). Source data are provided as a Source Data file.

collection, the starvation condition was switched to prolonged starvation media completed with doxycycline.

To block cell cycle, cells were plated in order to reach a 60–70% confluency after 24 h, and then were treated with 1 mM HU (Cat #H8627, Sigma) for 1 week, replacing the treatment every 48 h. Cells stop dividing after 24 h from treatment addition.

To perform ALR live-imaging experiments, cells were plated on Glass Bottom 35 mm Culture Dishes (VWR, Cat# 10810-054). CLN3-overexpressing cells were induced with doxycycline for 40 h. The day before imaging, cells were loaded in complete medium with 1 μg/μl dextran-488 (10.000 MW, Cat# D22910, ThermoFisher,) for 1 h at 37 °C, allowing internalization of the dye. Cells were then extensively washed with PBS, and then incubated with for 16 h in serum+glutamine starvation, which enables both dextran labelling of lysosomes, and activation of lysosomal reformation. Alternatively, ARPE19 stable cell lines expressing the TeT-ON LAMP1-mCherry construct were induced with 1μg/ml doxycycline (Sigma, Cat#D9891) for 40 h, and shifted to prolonged starvation media (containing doxycycline) 16 h prior to imaging. Before acquisition, nuclei were stained with the Hoechst 33342 (Cat# R37605, ThermoFisher) dye and media supplemented with 10 mM Hepes.

For siRNA-based experiments, cells were transfected using Lipofectamine® RNAiMAX Transfection Reagent (Cat#13778075, Invitrogen) with the indicated siRNAs and analyzed after 72 h unless stated otherwise. The following siRNAs were from Dharmacon: non-targeting siRNA Pool (D-001810-10-05). Other siRNA were synthesized form Sigma-Aldrich: siATG7-1 GAAGCUCCCAAGGACAUUA, siATG7-2 CGCUUAACAUUGGAGUUCA, siATG7-3 GGAACACUGUAUAACACCA, siVPS35-1 GUUGUUAUGUGCUUAGUA[30], siVPS35-2 AAAUACCACUUGACACUUA[30], siSNX5 CUACGAAGCCCGACUUUGA[30], siSNX6 UAAAUCAGCAGAUGGAGUA[30], siCI-M6PR-1 CUACCUGUAUGAGAUCCAA, siCI-M6PR-2 GGACGGCUGCAAUCAAUGA and siCI-M6PR-3 GCUAAACAGUUCGCAAGGA.

## CI-M6PR trafficking assays
CI-M6PR recycling assay was performed as previously reported[58]. To track CI-M6PRs after their internalization at the plasma membrane, cells were plated on coverslips, and after 48 h were incubated with saturating concentrations (5 μg/ml) of the monoclonal anti-CI-M6PR (Cat# NB300-514, Clone 2G11, Novus) antibody (recognizing the extracellular/luminal domain of the protein) at 37 °C for the indicated time points. Cells were then either washed and fixed with 4% PFA, or chased for the indicated time points (as reported in figure legends). Cells were quickly switched in ice-cold methanol, for 5 min at −20 °C, washed and blocked with 3% BSA for 15 min. CI-MPR-antibody complexes were imaged in permeabilized cells upon addition of the specific secondary antibody for 45 min at RT.

For CI-M6PR plasma-membrane assay, cells were quickly washed with media+BSA1%, and then incubated at 4 °C for 45 min with a pre-cooled mix containing saturating concentrations (5 μg/ml) of the monoclonal anti-CI-M6PR (Cat# NB300-514, Clone 2G11, Novus) antibody in complete media+10 mM HEPES, and then processed for FC or immunofluorescence. For recycled PM CI-M6PR, cells were incubated with the antibody at 37 °C for 15, 30 or 60 min in complete media, and then processed for FC or immunofluorescence. For immunofluorescence, cells were then washed two times with cold PBS, and then fixed with methanol-free 4%PFA for 10 min on ice. Plasma membrane CI-M6PR was imaged in non-permeabilized cells upon addition of the specific secondary antibody for 45 min at RT, and nuclei and cell membranes were respectively counterstained with Hoechst 33342 (Cat.# R37605, ThermoFisher) and CellMask™ Plasma Membrane Stain (Cat# C10046, ThermoFisher). For FC, cells were washed once with 2 ml of ice-cold FC buffer (PBS, 1% FBS, 2 mM EDTA), centrifuged at 300 g for 5 min, and then incubated with a pre-cooled mix containing the specific secondary antibody (1:500), for 45 min at 4 °C. Cells were then washed once, and incubated with the Helix NP™ NIR (Biolegend, cat# 425301) dye, a far-red emitting nucleic acid stain used for the discrimination of live and dead cells. The FACSDiva software was used to collect the data, while FlowJo (v10.8.1) was used for data analysis.

## Trafficking of rhGAA and PepstatinA
RhGAA was kindly provided by Prof. Giancarlo Parenti. rhGAA (2 mg/ml) was conjugated with the Alexa Fluor™–546 Protein Labeling Kit (ThermoFisher, cat# A10237) following manufacturer instruction. Cells were incubated with saturating concentrations of rhGAA (1:50 dilution, 40 μg/ml) in complete media at 37 °C for 45 min, washed and chased for the indicated time points, and then fixed and processed for immunofluorescence. For live-image experiments, cells were loaded with 1 μg/μl dextran-488 for 1 h at 37 °C, and then extensively washed for 16 h in serum+glutamine starvation medium (prolonged starvation media). The next day, cells were loaded with 40 μg/ml rhGAA for 60 min at 37 °C and then washed prior to imaging.

Pepstatin A labels active cathepsin D inside lysosomes. For PepstatinA loading experiments, cells were loaded with 1 μM Pepstatin A-BODIPY® Conjugate (Life technologies, cat# P12271), for 30 min at 37 °C, and then fixed at 37 °C for 15 min via the addition of an equal volume of prewarmed, freshly made 8% PFA in 2× microtubule stabilization buffer (MTSB; 160 mM PIPES pH 6.8, 10 mM EGTA, 2 mM MgCl2) to minimize tubules rupture, as previously reported[59].

## Mass spectrometry
All the experiments were performed in a labeling free setting and samples prepared using the in StageTip (iST) method[60]. Instruments for LC MS/MS analysis consisted of a NanoLC 1200 coupled via a nano-electrospray ionization source to the quadrupole-based Q Exactive HF benchtop mass spectrometer[61]. Peptide separation was carried out according to their hydrophobicity on a home-made chromatographic column, 75 μm ID, 8 Um tip, 250 mm bed packed

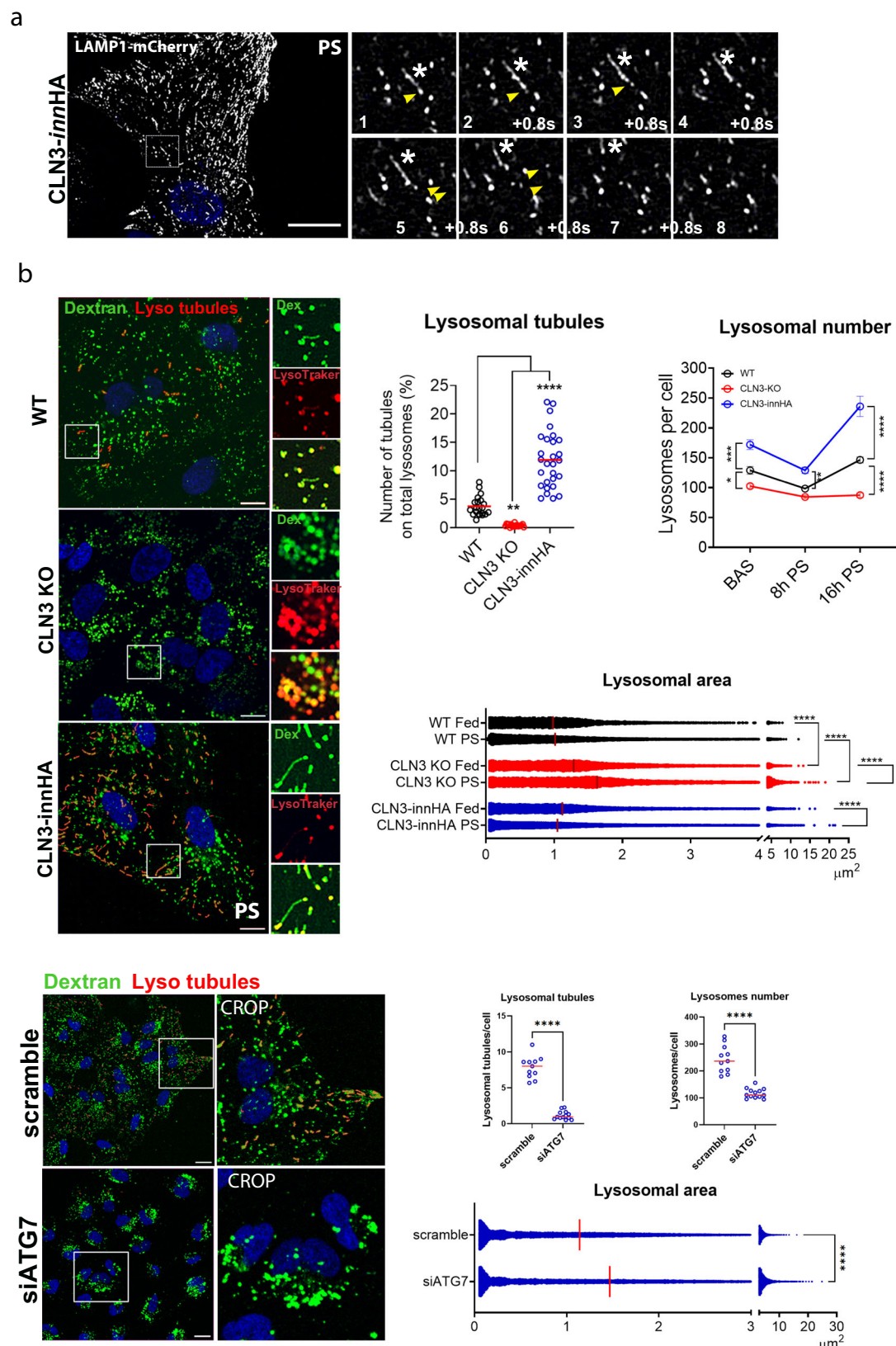

with Reprosil-PUR, C18-AQ, 1.9 μm particle size, 120 Angstrom pore size (New Objective, Inc., cat. PF7508-250H363), using a binary buffer system consisting of solution A: 0.1% formic acid and B: 80% acetonitrile, 0.1% formic acid. Runs of 240 min after loading were used for proteome, while runs of 75 min were used for Interactome and Lyso-IP. In both cases, a constant flow rate of 300 nl/min was used. MS data

were acquired using a data-dependent top-20, for the proteome, or top-15, for interactome, method with maximum injection time of 20 ms, a scan range of 300–1650Th, an AGC target of 3e6 and a resolution of 120,000. Resolution, for MS/MS spectra, was set to 15,000 at 200 m/z, for the proteome, and 45,000 at 200 m/z, for interactome. AGC target was set 1E5, max injection time to 20 ms and

**Fig. 7 | CLN3 drives lysosomal tubulation and reformation. a** Representative time-lapse images of ARPE19-CLN3innHA + LAMP1-mCherry cells, induced with 1 µg/ml doxycycline for 40 h, and highlighting tubules (asterisks), budding and being released from lysosomes (arrows). Cells were imaged after 16 h of serum +glutamine starvation to induce ALR. Repeated six times. Scale bar 20 µm. **b** Representative live-image confocal snapshots of ARPE19 WT, CLN3 KO and CLN3-overexpressing cells treated with 1 µg/ml doxycycline for 40 h, loaded with 1 µg/µl Dextran for 1 h the night before imaging, and then treated with PS media for 16 h to label lysosomes (green), with lysosomal tubules marked in red. Images were generated with the Imaris software. Enlargement panels show lysosomes and tubules stained with dextran (green) and lysosotracker (red, 100 nM). Lysotracker was added right before imaging. Scale bar 10 µm. Top graphs depict the number of lysosomal tubules and lysosomes showed in (**b**). Number of lysosomal tubules (left graph) is normalized against the total number of lysosomes per field (N = 102–134 cells, three independent experiments). Lysosome number per cell (right graph), is normalized against the total number of cells, and quantified in fed (FED) and starvation conditions (8 h and 16 h PS, prolonged starvation) (basal N = 274–375 cells, 8 h PS N = 344–500 cells, 16 h PS N = 225–598 cells). N = 3 independent experiments. Mean ± SEM, two-way Anova (*$p < 0.05$, **$p < 0.01$, ***$p < 0.001$, ****$p < 0.0001$). Bottom graph shows lysosome area quantification from images shown in (**b**). Each spot represents a single lysosome, quantified in fed and starvation conditions (N = 102–134 cells, N = 3 independent experiments). Mean + single values are reported, one-way Anova. PS, prolonged starvation (*$p < 0.05$, **$p < 0.01$, ***$p < 0.001$, ****$p < 0.0001$). **c** Representative live-image confocal snapshots of ARPE19-CLN3innHA transfected with ATG7 or scramble siRNA for 72 h, induced with doxycycline for 40 h and loaded with 1 µg/µl dextran as described in (**b**). Scale bar 20 µm. Graphs show number of lysosomal tubules, normalized against the total number of lysosomes per field, lysosome number per cell, normalized against the total number of cells, and lysosomes area quantification, each spot is a single lysosome. N = 197–303 cells, three independent experiments, mean + single values are reported, unpaired *t*-test (two-tailed) (****$p < 0.0001$). Source data are provided as a Source Data file.

the isolation window to 1.4Th. The intensity threshold was set at 2.0 E4 and Dynamic exclusion at 30 s.

## MS data processing and analysis

Raw mass spectrometry data were processed with MaxQuant (1.6.2.10)[62] using default settings (FDR = 0.01, oxidized methionine (M) and acetylation (protein N-term) as variable modifications, and car-bamidomethyl (C) as fixed modification). For protein assignment, spectra were correlated with the Uniprot *Homo Sapiens* (v.2019), including list of common contaminants. Label-free quantitation (LFQ) and "Match between runs" were enabled. Bioinformatics analysis was performed with Perseus 1.6.2.3[63]. The LFQ intensities were logarithmized, grouped and filtered for min.valid number (min.3 in at least one group). Missing values have been replaced by random numbers that are drawn from a normal distribution. Proteins with Log2 ratios ≥1 and a *p* value ≤ 0.05 were considered significantly enriched. To identify significant enriched GO terms in Lyso-IP, we utilized the 1D enrichment tool in Perseus[64]. The protein-protein interaction network was built in the Cytoscape environment[65]. Proteins belonging to the selected cluster were loaded into the STRING plugin and the network was subsequently generated[66].

The mass spectrometry proteomics data have been deposited to the ProteomeXchange Consortium via the PRIDE[67] partner repository with the dataset identifier PXD031582.

## Cell lysis, western blotting and immunoprecipitation

Cells were rinsed once with PBS and lysed in ice-cold RIPA buffer (150 mM NaCl, 1% NP-40, 0.5% sodium deoxycholate, 0.1% SDS, 1 mM EDTA, 50 mM Tris HCl pH 8) supplemented with protease and phosphatase inhibitors. Total lysates were passed ten times through a 25-gauge needle with syringe, kept at 4 °C for 30 min, processed with three freeze-thaw cycles, and then cleared by centrifugation in a microcentrifuge (18,800 g at 4 °C for 20 min). Protein concentration was measured by BCA assay and samples were prepared in Laemmli buffer and boiled for 25 min at 37 °C. Quantification of western blotting was performed by calculating the intensity of protein bands by densitometry analysis using the Fiji software.

For HA immunoprecipitations, cells grown in 10 cm culture dishes were washed twice with cold PBS and then incubated with 1 mg/mL DSP (dithiobis(succinimidyl propionate)) (Cat#22585, Thermo Fischer Scientific) crosslinker for 7 min at room temperature. The cross-linking reaction was quenched by adding Tris-HCl (pH 8.5) to a final concentration of 100 mM. Cells were rinsed twice with ice cold PBS and lysed with RIPA (150 mM NaCl, 1% NP-40, 0.5% sodium deoxycholate, 0.1% SDS, 1 mM EDTA, 50 mM Tris HCl pH 8) lysis buffer supplemented with protease and phosphatase inhibitors. Cell lysates were then incubated with anti-HA (Cat# A2095, Sigma) or anti-GFP trap agarose beads (Cat# GTA-20, Chromotek) at 4 °C, washed six times, resolved by SDS-polyacrylamide gel electrophoresis on 4–12% Bis-Tris gradient gels (Cat# NP0336BOX NuPage, Thermo Fischer Scientific) and analyzed by immunoblotting with the indicated primary antibodies.

For endogenous CLN3 immunuoprecipitation, cells grown in 10 cm culture dishes were rinsed twice with PBS and lysed in ice-cold RIPA lysis buffer (150 mM NaCl, 1% NP-40, 0.5% sodium deoxycholate, 0.1% SDS, 1 mM EDTA, 50 mM Tris HCl pH 8) supplemented with protease and phosphatase inhibitors. Cell lysates were then incubated overnight with anti-CLN3 antibody (1 µg per mg of immunoprecipitated protein) or with anti-rabbit IgG (Bethyl, Cat.# P120-101) at 4 °C, then incubated for 45 min at 4 °C with ProteinA-Sepharose 4B Conjugate (Invitrogen, Cat#101041), washed four times and resolved by SDS-polyacrylamide gel electrophoresis on 4–12% Bis-Tris gradient gels (Cat# NP0323PK2 NuPage, Thermo Fischer Scientific) and analyzed by immunoblotting with the indicated primary antibodies.

## Lysosomal immunopurification (Lyso-IP)

Lysosomes from cells expressing TMEM192-3xHA were purified as previously described[22,68]. Briefly, cells were seeded in a 15 cm at a density appropriate for them to reach confluency after 24 h. Samples were processed separately to ensure rapid isolation of lysosomes using buffers that were pre-chilled on ice. Cells were quickly rinsed twice with ice-cold PBS buffer and then scraped in KPBS (136 mM KCl, 10 m M KH2PO4, pH 7.25), supplemented with Protease Inhibitors, and collected by centrifugation at 1000 × *g* for 2 min at 4 °C. Pelleted cells were resuspended in a total volume of 1 ml of fractionation buffer (140 mM KCl, 50 mM Sucrose,1 mM DTT, 2mMEGTA, 2.5 mM MgCl2, 25 mM Hepes, pH adjusted to 7.25 in KOH) supplemented with Protease Inhibitors, and 50 µl of the suspension was stored for processing of the whole-cell fraction (input). The remaining 950 µl were gently homogenized with 20 strokes of a 2 ml homogenizer, followed by centrifugation at 1000 g for 5 min. Post-nuclear supernatant was harvested and incubated at 4 °C with anti-HA magnetic beads (Cat#88836, ThermoFisher) pre-equilibrated in fractionation buffer. Lysosome-bound beads were gently washed four times with KPBS on a DynaMag Spin Magnet and then eluted in fractionation buffer+NP40 0.5% for 30 min at 4 °C. Beads were removed and the resulting eluate was prepped for subsequent proteomics and biochemical analyses.

## Confocal microscopy

For immunofluorescence experiments, cells were grown on 8-well Lab-Tek II—Chamber Slides or coverslips and treated as indicated. Cells were fixed with 4% PFA for 10 min, washed three times with PBS, blocked and permeabilized for 30 min with blocking solution (0.05% saponin, 0.5% BSA, and 50 mM NH4Cl in PBS). For endogenous CLN3 staining, samples were permeabilized for 5 min by the addition of 20 µM digitonin under continuous agitation, washed three times and blocked for 30 min with 5% goat serum and 50 mM NH4Cl. Primary

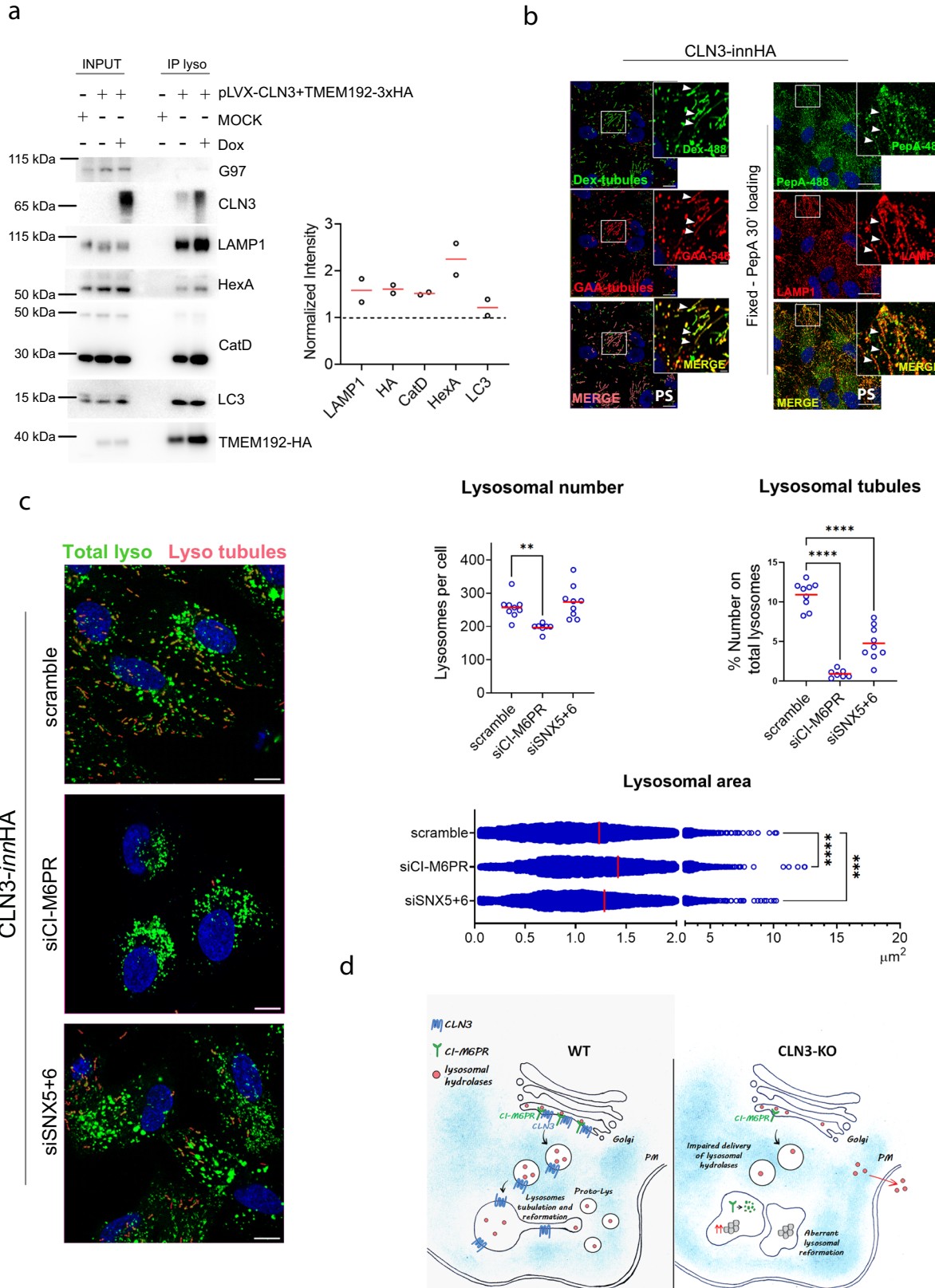

antibody incubations were performed overnight at 4 °C. Samples were washed with PBS and incubated with fluorochrome-conjugated secondary antibodies for 50 min at RT (Alexa Fluor 488, Alexa Fluor 568, and Alexa Fluor 633). Fixed cells were mounted in VECTASHIELD® mounting medium with DAPI and analyzed using LSM 710 or LSM 880 + Airyscan systems (Carl Zeiss). Optical sections were collected under a ×63 oil immersion objective at a definition of 1024 × 1024 (average of eight or sixteen scans), with the pinhole adjusted to 1 Airy unit for each channel emission to have all intensity values between 1 and 254 (linear range).

For high content images, cells were seeded in 96-well plates and incubated for 24 h. After incubation, cells were treated as incubated

**Fig. 8 | CLN3 modulate ALR by promoting the delivery of lysosomal enzymes.**
**a** Lysosomal immuno-purification analysis of pLVX-CLN3-innHA cells with and
without 40 h doxycycline induction (repeated twice). Cells were analyzed by
immuno-blot analysis with the indicated antibodies. Quantification is shown in
graph. **b** Left, representative live-image confocal snapshots of ARPE19 CLN3-innHA
cells upon prolonged starvation and 40 h doxycycline induction, loaded with 1 µg/
µl dextran as described in **7b**, and then loaded for 60 min with 40 µg/ml rhGAA-546
before imaging. Lysosomal tubules that are positive to both Dextran and GAA, are
reported in pink and are indicated by white arrows. Enlargements areas are
reported. Scale bar 7µm. Right, ARPE19 CLN3-innHA cells upon prolonged starva-
tion and 40 h doxycycline induction, loaded with 1 µM PepA (green) and stained for
LAMP1 (red). PepA-488 and LAMP1 positive tubules are indicated by white arrows.
Repeated two times. Scale bar 20 µm. **c** Representative live-image confocal snap-
shots of ARPE19-CLN3innHA transfected with CI-M6PR, SNX5 + 6 or scramble siRNA
for 72 h, induced with 1 µg/ml doxycycline for 40 h and loaded with 1 µg/µl dextran
as described in **7b**. Scale bar 20 µm. Graphs show number of lysosomal tubules,
normalized against the total number of lysosomes per field, lysosome number per

cell, normalized against the total number of cells, and lysosomes area quantifica-
tion. $N = 5–9$ fields per replicate, three independent experiments. For lysosomal
tubules quantification and lysosome number per cell, mean + single values (each
spot is an averaged measurement of the replicates) are shown. For lysosomes area,
each spot is a single lysosome. $N = 5–9$ fields per replicate, three independent
experiments, one-way Anova (Dunnett's multiple comparisons test) (**$p < 0.01$,
***$p < 0.001$, ****$p < 0.0001$). **d** A model for the role of CLN3 in the regulation of
lysosomal biogenesis and reformation. In the presence of CLN3, lysosomal
enzymes are recognized by CI-M6PR, and then they are sorted into lysosomes,
resulting in lysosomal biogenesis and reformation. On lysosomes, CLN3 modulates
the formation of tubules, which then gives rise to new proto-lysosomes. The
absence of CLN3 results in mis-trafficking of the CI-M6PR, that is then degraded
inside lysosomes. This leads to mis-sorting of neo-synthesized lysosomal enzymes
into the secretion route, causing reduction of multiple lysosomal enzymes within
the lysosome and consequent accumulation of undigested material (lipofuscins),
impaired mTOR reactivation and failure of ALR. Source data are provided as a
Source Data file.

above, then fixed with 4% PFA for 10 min, washed three times with PBS,
blocked and permeabilized for 30 min with blocking solution (0.05%
saponin, 0.5% BSA, and 50 mM NH4Cl in PBS). Primary and secondary
antibody incubations were performed as described above. At least ten
images per field were acquired in each well using the confocal auto-
mated imager OPERA, from PerkinElmer. For image analysis, we used
Columbus 2.6.0.127073 released by PerkinElmer. This online platform
is based on Harmony High-Content Imaging and Analysis Software
which provides an easy quantification of complex cellular phenotypes.

### Live cell imaging
For live imaging of lysosomal tubules, cells were plated in glass-
bottomed dishes (Cat# 10810-054, VWR), and cells were maintained at
37 °C with 5% CO. Right before imaging, nuclei were stained with the
Hoechst 33342 (Cat# R37605, ThermoFisher) dye and media supple-
mented with 10 mM Hepes. Imaging was performed on a LSM 880
confocal microscope (Zeiss) equipped with a full incubation chamber,
and optical sections were obtained under a ×40 water immersion
objective at a definition of 1024 × 1024, with the pinhole adjusted to 1
Airy unit for each channel emission to have all intensity values between 1

and 254 (linear range). Images were collected with a Zeiss Zen Blue
software, and movies were recorded at 3 s intervals for at least 50 cycles.

### Image analysis
For live-imaging experiments, morphometric analyses of lysosomes
and lysosomal tubules were performed with the Imaris software. Total
lysosomes (Dextran channel) were identified with the Imaris 'surface'
module: for thresholding, constant image smoothing and background
subtraction were applied. Surfaces detail (0.2 µm) and diameter of
largest sphere (0.150 µm) were determined (threshold values were
kept constant along images). Lysosomal structures smaller than
0.05 µm were filtered out. Lysosomal tubules were also identified with
the Imaris "surface" module, applying a sphericity and ellipsoid filter to
lysosomal masks, which allowed to filter out non-tubular lysosomal
structures based on their shape.

For the quantification of the number of GAA and CI-M6PR parti-
cles, image were processed with the Imaris software. The 'cell' module
was applied to identify nuclei, cell borders and spots: briefly, constant
image smoothing and background subtraction were applied to identify
nuclei, cells and particles, and kept constant along images. Moreover,
nuclei were identified according to their threshold and diameter, and
split by seed points. Cells were split considering either cell channel
intensity or distance from the nucleus, and finally particles were
identified according to their diameter and thresholding parameters.

The level of colocalization (CLN3-LAMP, CLN3-TGN49, lumCI-
M6PR-EEA1, lumCI-M6PR-G97, rhGAA-EEA1, rhGAA-LAMP1) was calcu-
lated acquiring confocal sections at the same laser power and photo-
multiplier gain. Images were then processed using the Image J
software. Single channels from each image were converted into 8-bit
grayscale images. The ImageJ JACoP Plugin was then used to create
thresholded images (dark background threshold was applied to sub-
tract background) and to calculate Mander's Colocalization
Coefficients (MCC).

To calculate the percentage of LC3-LAMP1+ autolysosomes, the
number of LC3-GFP spots and LC3-LAMP1 overlapping area, confocal
sections were acquired at the same laser power and photomultiplier
gain. Images were then processed using the ImageJ software. Single
channels from each image were converted into 8-bit grayscale images,
and then thresholded in order to subtract background. The Image J
"Analyze Particles" plugin was then used to identify, count, and mea-
sure the area of structures (with an area above 0.05 µm²) in channel 1
(LAMP1) and in channel 2 (LC3). The structures in channel 1 (LAMP)
were used to build a LC3ΔLAMP1 mask, that was generated by over-
lapping LAMP1 structures to LC3 channel, and then by subtracting the
structures containing both markers. The remaining structures, positive
only for LC3 (LC3ΔLAMP1), were counted and their area was measured
with the "analyze particles" tool. By difference, the number of

**Table 1 | List of lysosomal enzymes significantly reduced in
both lysosomes of CLN3- and LYSET-KO cells[48]**

| Luminal lysosomal enzymes | | |
|---|---|---|
| **Reduced in CLN3-KO & LYSET-KO** | **Log2 Difference (CLN3-KO/CTRL)** | **Log2 Difference (LYSET-KO/CTRL)** |
| CTSH | −4.58066 | −2.88505 |
| DPP7 | −3.37081 | −4.88403 |
| HEXA | −3.19612 | −4.70883 |
| PRCP | −3.12789 | −2.64847 |
| ARSA | −2.72279 | −4.38194 |
| PPT1 | −2.60860 | −2.49936 |
| PLBD2 | −2.47825 | −5.46654 |
| GAA | −2.46810 | −0.64867 |
| GNS | −2.36429 | −3.49467 |
| CTSZ | −2.12500 | −4.80665 |
| AGA | −2.12223 | −2.57942 |
| GUSB | −2.02897 | −3.42793 |
| CTSA | −1.72551 | −3.76310 |
| GLB1 | −1.54273 | −4.99637 |

List of lysosomal enzymes significantly reduced in both lysosomes from CLN3- and LYSET-KO
cells. Data from LYSET-KO cells were obtained from Pechincha et al.[48]

structures containing both LAMP1 and LC3 (autolysosomes), as well as the identification of the LC3-LAMP1 overlapping area was calculated by subtracting LC3 numbers and area values to LC3ΔLAMP1 values (note that this quantitative analysis procedure does not use merged images and is not affected by the fluorescence intensity).

## Enzymatic assays

GAA assay: Acid α-Glucosidase (GAA) activity was assayed by using the fluorogenic substrate 4-methylumbelliferyl-α-D-glucopyranoside (4MU) (Sigma-Aldrich) according to a published procedure. Briefly, 3 μg of immuno-purified lysosomes, 10 μg of sample lysates, or 10 μl of concentrated cell media were incubated with the fluorogenic substrate (2 mM) in 20 μl of 0.2 M acetate buffer (pH 4.0), for 60 min at 37 °C. The reaction was stopped by adding 200 uL of 0.5 M glycine-carbonate buffer (pH 10.7). Fluorescence was measured at 365 nm (excitation) and 450 nm (emission) on a Promega GloMax Multidetection system fluorimeter. Protein concentration was measured by the Pierce BCA protein assay kit.

HexA/B assay: HexA/B activity was assayed by using the substrate 4-Methylumbellifery-β-D- acetamide-2-deossi- β-glucopiranoside. 3 μg of immuno-purified lysosomes, 10 μg of sample lysates, or 10 μl of concentrated cell media were incubated at 50 °C × 2 h (inactivation of HexA) or maintained at 4 °C (HexA+B). Then all the samples were incubated with the florigenic substrate (5 mM) in a buffer containing 0.2 M Na-phosphate, 0.1 M Citrate Buffer (pH 4.4) and 0.2% Albumin (Pi/Ci Buffer-BSA) for 30 min at 37 °C in incubation mixtures of 20 μl. The reaction was stopped by adding 200 uL of 0.5 M glycine-carbonate buffer (pH 10.7). Fluorescence was read at 365 nm (excitation) and 450 nm (emission) on a Promega GloMax Multidetection system fluorimeter. Protein concentration was measured by the Pierce BCA protein assay kit.

GUSB assay: B-Glucoronidase (GUSB) activity was assayed by using the florigenic substrate 4-methylumbelliferyl-β-glucuronide (4MU). 3 μg of immuno-purified lysosomes, 10 μg of sample lysates, or 10 μl of concentrated cell media were incubated with the florigenic substrate in 0.15 M Acetate buffer (pH 3.5) for 60 min at 37 °C in incubation mixtures of 50 μl. The reaction was stopped by adding 200uL of 0.5 M carbonate buffer (pH 10.7) supplemented with 0.025% Triton X100. Fluorescence was read at 365 nm (excitation) and 450 nm (emission) on a Promega GloMax Multidetection system fluorimeter. Protein concentration was measured by the Pierce BCA protein assay kit.

## Electron microscopy

For immuno-EM analysis, the cells were fixed with the mixture of 4% paraformaldehyde (PFA) and 0.05% glutaraldehyde (GA) for 10 min at RT, then 'washed' with 4% PFA once to remove the residual GA and fixed again with 4% PFA for 30 min at RT. Next the cells were incubated with the blocking/permeabilizing mixture (0.5% BSA, 0.1% saponin, 50 mM NH4Cl) for 30 min and subsequently with the primary antibody against LAMP-1 (Developmental Studies Hybridoma Bank, Cat N° H4A3-a) or primary purified antibody against HA (BioLegend, Cat N° 16B12-Previously Covance catalog# MMS-101P), diluted 1:500 (LAMP-1) or 1:100 (HA) in blocking/ permeabilizing solution. The following day, the cells were washed and incubated with the secondary antibody, anti-mouse Fab' fragment coupled to 1.4-nm gold particles (Nanoprobes, Cat N° 2002, anti-mouse nanogold) diluted 1:50 in blocking/permeabilizing solution, for 2 h RT. The GoldEnhance™ EM kit (from Nanoprobes) was used to enhance ultrasmall gold particles.

For conventional EM the cells (or tissue) were fixed with 1% GA prepared in 0.2 M HEPES buffer for 30 min (RT). Samples prepared for IEM or conventional EM were post-fixed in OsO4 and uranyl acetate, dehydrated, embedded in Epon and polymerized at 60 °C for 72 h. For each sample, thin sections were cut using a Leica EM UC7 ultramicrotome (Leica Microsystems, Vienna, Austria). EM images were acquired from thin sections using a FEI Tecnai-12 electron microscope

(FEI, Eindhoven, Netherlands) equipped with a VELETTA CCD digital camera (Soft Imaging Systems GmbH, Munster, Germany).

## Statistics and reproducibility

The experiments were repeated at least three times, unless stated otherwise. As indicated in the figure legends, all quantitative data are presented as the mean of biologically independent experiments or samples. For each experiment we described specific statistic test used and the relative significance in the figure legend. Statistical analyses were performed using GraphPad Prism 8.0.

## Reporting summary

Further information on research design is available in the Nature Portfolio Reporting Summary linked to this article.

## Data availability

Full scans for all western blots as well as source data for all the graphs are provided with this paper. For graphs, the exact $p$ value for all the experiments is present in the Source data file. All other data are available from the corresponding author on reasonable request. The MS proteomics data were deposited to the ProteomeXchange Consortium through the PRIDE partner repository with the dataset identifier PXD031582. To identify enriched GO terms in the Lyso-IP dataset, we utilized the 1D enrichment tool in Perseus. The protein-protein interaction network was built in the Cytoscape environment. Source data are provided with this paper.

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

## Acknowledgements

We thank Juan S Bonifacino and Michael Marks, Carmine Settembre, Graciana Diez-Roux and Dustin C. Bagley for for helpful suggestions, and critical reading of the manuscript. We thank Angela Nacca for technical help. This work was supported by the US National Institutes of Health (RO1-NS078072 and RO1CA260205), the Huffington Foundation (A.B.); the NCL-Stiftung foundation (A.C.), the Italian Telethon Foundation (A.B.); MIUR PRIN E5L5P3_OO2 (A.B.), European Research Council H2020 AdG 'LYSOSOMICS 694282' (A.B.); European Regional Development Fund—POR Campania FESR 2014/2020 (A.B) A.I.R.C. (Italian Association for Cancer Research) 'IG-22103' and '5×1000-21051' (A.B.), the Wellcome Trust (104568/Z/14/Z and 220260/Z/20/Z) (P.J.C.); the Medical Research Council (MR/L007363/1 and MR/P018807/1) (P.J.C.); the Lister Institute of Preventive Medicine and the Royal Society Noreen Murray Research Professorship (RSRP/R1/211004) (PJC). Research reported in this publication was supported by the Eunice Kennedy Shriver National Institute of Child Health & Human Development of the National Institutes of Health under Award Number P50HD103555, for the use of the Human Neuronal Differentiation Core (HNDC) and of the Neurovisualization core. The content is solely the responsibility of the authors and does not necessarily represent the official views of the National Institutes of Health. This work was supported in part by the Cytometry and Cell Sorting Core at Baylor College of Medicine with funding from the CPRIT Core Facility Support Award (CPRIT-RP180672), the NIH (CA125123 and RR024574) and the assistance of Joel M. Sederstrom.

## Author contributions

A.C. and An.B. conceived the study. A.C. designed and performed most of the experiments. L.S. performed most of the experiments involving CLN3 staining, interpreted the results and suggested experiments. N.M. performed enzymatic assays and GAA maturation experiments. G.D.T. and N.Z. generated the CLN3 antibody. N.Z., N.J.H. performed experiments involved in CLN3- and Lyso-IP, and provided technical support to A.C. in the execution of the experiments. C.C. and P.G. generated and analyzed proteomics data. T.H. generated constructs and transcriptional data, and supported A.C. in the execution of the experiments. J.M. generated Crispr/Cas9 gene-edited ARPE19 cells and performed some microscopy experiments. A.E. performed endogenous CLN3 immunoprecipitation, designed the model, and supported A.C. in the execution of the experiments. Al.B and M.Z. were involved in experiments related to the characterization of trafficking defects in CLN3-depleted cells. L.P. performed flow cytometry experiments. E.P. performed electron microscopy analysis. R.C. and P.J.C. generated HeLa CLN3-KO cells, performed some experiments for their characterization, revised the paper and suggested experiments. D.L.M. provided resources and supervision for high content experiments. N.P. provided technical support to A.C., M.A.D.M., and G.P. revised the paper and suggested experiments. A.C. and An.B. wrote the paper. An.B. supervised the study.

## Competing interests

A.B. is co-founder of CASMA Therapeutics and advisory board member of Next Generation Diagnostics and Avilar and Coave Therapeutics. The remaining authors declare no competing interests.
