## [Peer Review File · Nature Communications]

LOSS OF THE BATTEN DISEASE PROTEIN CLN3 LEADS TO
MIS-TRAFFICKING OF M6PR AND DEFECTIVE
AUTOPHAGIC-LYSOSOMAL REFORMATIONReviewers' comments:

Reviewer #1 (Remarks to the Author):

Summary:

This paper reports two roles for CLN3 that are linked to maintaining proper lysosome function. First, they report that CLN3 plays a role in promoting lysosome biogenesis through ALR. Their CLN3 overexpression cell culture system displays a significant increase in lysosome tubulation in conditions that induce ALR (prolonged serum/glutamine starvation). In contrast, CLN3 KO cells exhibit markedly reduced lysosome tubulation compared to that of wildtype and CLN3-overexpressing cells. The second role they assign CLN3 is its role in PM-to-Golgi retrograde trafficking. They identify CI-M6PR as an interactor of CLN3 and show that CLN3 mediates retrograde transport of CI-M6PR to the Golgi. However, in CLN3-deficient cells, this retrograde transport is impaired, resulting in mislocalization of CI-M6PR on early endosomes. Consequently, this trafficking defect impairs proper sorting of lysosomal enzymes, contributing to reduced lysosomal degradative capacity. The authors propose that lysosome dysfunction, as a result of the trafficking defect caused by CLN3-deficiency, may be a disease mechanism of Batten disease and potentially other neuronal ceroid lipofuscinoses. MFSD8/CLN7 mutations (also a NCL disease) cause ALR defects (PMID: 29514215) and like CLN3 a lysosomal membrane protein. As such, this isn't the first example of a link between CLN genes and ALR.

Major Comments:

- The authors describe a lysosome biogenesis pathway via CLN3-mediated ALR. TFEB-dependent lysosome biogenesis is not mentioned anywhere in the manuscript. Presumably, the starvation condition (serum/glutamine starvation) used to induce ALR also activates TFEB nuclear translocation. Where does TFEB fit in this picture? How is TFEB affected by CLN3 KO?
- Autophagy control – in cells deficient for autophagy (e.g. ATG5 knockout), is CLN3-mediated ALR absent? Presumably, ALR requires autophagy – as the name implies – so the prediction here would be that autophagy-deficient cells do not display ALR. This control may help distinguish whether lysosomal tubulation is autophagy-dependent (ALR) or whether it is lysosomes tubulating independent of autophagy.
- Line 85: Authors claim that N/C terminal tagging of CLN3 causes mis-localization. Please add citation to papers that prove this point or show this mis-trafficking in the figures.

- A previous study (PMID: 20526321) reported that ALR requires reactivation of mTOR. How is mTOR reactivation affected in the context of CLN3 KO? Is mTOR reactivation impaired in CLN3 KO?
- Based on still images – I am not convinced that their lysosomal tubulation measurements are accurate. Subcellular structures quantified as tubules on still images seem arbitrary. Based on what criteria were structures quantified as ALR tubules? Maybe the movies are more convincing?
- Line 123 ff: Authors claim that overexpression of CLN3^{innHA} constructs causes excessive ALR. Could the phenotype be due to overexpression, since cells expressing CLN3^{innHA} appear to contain more dextran in lysosomes /& tubules (Suppl. Fig 4a). Given that the authors previously claim that N/C terminal fusion of CLN3 are mis-trafficked, please add these as controls to the ALR experiments.
- Previous studies have shown that adback of 10% FBS to starved cells for as little as 15 min can robustly induce ALR (PMID: 33119550). Given the problems with the tagging of CLN3 and the potential of overexpression artifacts, the authors should test if this method would be an alternative to measure ALR deficiency in CLN3 knockouts.
- Authors claim that CLN3 is integral to lysosomal reformation and show that KO has no lysosomal tubules (Fig 3c). However, CLN3 KO shows extensive tubulation in representative images depicted in Supplementary Figure 4h, which is either an error in the figure annotation or would contradict the authors previously stated conclusions. Please elaborate on this discrepancy.
- Results related to CI-M6PR recycling is confusing. Figure 4C indicate that WT cells have more CI-M6PR on PM compared to that of CLN3 KO. This appears to contradict the results in Figure 4D where more CI-M6PR is seen on PM fraction in CLN3 KO cells.
 - o Also, could the authors elaborate on the reduction in total CI-M6PR levels in CLN3 KO cells (Supp Figure 6D)? I would have imagined CI-M6PR being more abundant in CLN3 KO cells because more of the protein appear to be localized on early endosomal structure where presumably there is less chance of turnover, as opposed to being in golgi/lysosome.
 - o Furthermore, model drawn in Figure 7 shows more CI-M6PR on the PM and endosomes in CLN3 KO. Please elaborate on this discrepancy.
- Figure 4E – I am not sure what the data is trying to say. Is there a better way to depict this?

- It is unclear whether CLN3-mediated ALR and retrograde transport are linked or not. Does ALR affect retrograde trafficking? If so, how? Maybe this can be tested via depletion of Lyspersin or PI4K2A and perform colocalization studies of CI-M6PR and EEA1/Golgi markers.

- In general, materials and methods are very general and not detailed enough to understand all the experimental outcomes.

- Please add information about cell culture treatments (concentration of media, drugs, dyes & duration).

- Antibody validation is missing.

- For image analysis matters, more detail about pre-processing, filtering and thresholding for co-localization and quantification is necessary. Which thresholding kernel did the authors use for the segmentation?

- High-content imaging is not mentioned in the materials and method. On which system was it performed?

-

Minor comments

- The authors claim to have generated a new CLN3-specific antibody. However, the authors do not show any of the validation and testing of the antibody. Please incorporate into the supplement

- Knockouts genomic sequencing information about be included in the supplemental figure and not in the materials and methods. Authors state that cells containing the homozygous mutation (c.1055 delA) were expanded, but two clones are show in the supplementary figure 1. Which clone was used for the later figures? Please indicate in text or materials. Interestingly, RNA levels are ~50% reduced in KOs (Suppl. Fig. 1g). Is there an explanation for this?

- What were the CLN3^{innHA} expression levels used in the experiments. Authors fail to disclose dox-concentration, duration of induction or WB on protein levels. What happens if you rescue CLN3 closer to WT levels?

- Style of boxplots/scatterplots is not consistent throughout the paper. Please change all boxplots to either scatterplots or boxplots + scatterplots, to show the underlying data-distribution.

- Line 448: CI-M6PR assay: Why did authors change the cited fixation method for their experiments? Cited paper states the fixation with MeOH, but authors choose to fix with PFA and then MeOH. Could they elaborate on their reasoning behind this?

- Size of scale bars for image insets are not mentioned or depicted. Please add.

- Fig 4e: Y-axis of FACS histogram are not matching between WT and KO. Please adjust.

- Fig 5a: Authors state that “several mitochondrial proteins” accumulate in CLN3 lysosomes, but these are not annotated in the volcano plots. Please add labels for proteins of interest.
- Fig 6b: Cell masks indicated are based on what? Some cells in the example image of the CLN3 KO were apparently discarded (no highlight of cell and nucleus) for the analysis, based on what metrics?
- Fig 6b: Image panel overlaps into evaluation plot
- Suppl. Fig 1d: CLN3 signal is clearly visible in the HeLa KO and overlaps with Lamp1 signal. Why is Lamp1 all peri-nuclear, but in later figures distributed over the whole cell body?
- Suppl. Fig 2c/d: Authors show one quantification of WB band intensities, but in Suppl. Fig. 4g they show the same analysis with more than n=1. Please add more repeats to analysis of Suppl. Fig 2c/d.
- Suppl. Fig 4c: Lamp1 spots are normalised per well, but should be normalised to number per cell. Otherwise, this could skew the data if one replicate has more cells / well than the other. Please change.
- Suppl. Fig 6a: inserts do not match the ROI indicated in the overview images. Some the insets are rotated compared to the overview areas. Please correct.

Overall, the paper contains several interesting observations related to CLN3 but there is a lack of coherency in the story, and some aspects of the paper appear to be of limited robustness. The paper would be strengthened if a link between ALR and retrograde transport could be found. As it stands, the two processes are disjointed and there isn't a molecular explanation that paints a clear pathway for CLN3.

Reviewer #2 (Remarks to the Author):

This manuscript seeks to elucidate the cellular roles of Cln3. Mutations in Cln3 are associated with Batten's disease, a form of lysosomal storage disorder. Thus, understanding how Cln3 acts has significant disease implications. This study has two main findings. First, Cln3 acts in autophagic lysosome reformation (ALR). Second, Cln3 regulates the endosomal transport of mannose-6-phosphate receptor (MPR).

To gain insight into how Cln3 acts in ALR, a proteomic approach was undertaken, which identified many Cln3 interactors. Some of these are transport factors acting in endosomal pathways. Functional studies identified two being relevant for Cln3 acting in ALR, lyspersin (a component of the BORC complex) and PI4K2A. For MPR transport, this study shows that Cln3 regulates the retrograde transport of MPR from the early endosome (EE) to the Golgi. Perturbing this regulation reduces the delivery of lysosomal proteins to the lysosome. The study also generated an antibody, which appears for the first time to be

able to detect endogenous Cln3. Using this antibody, the study detects Cln3 to reside predominantly at the lysosome and the Golgi.

Although these findings provide new information on Cln3, they represent limited insights.

Major comments:

1. The main novelty of this study is the finding that Cln3 acts in ALR. Although lyspersin and PI4K2A were identified to be needed for this role of Cln3, how these factors act in this context needs to be clarified. This is particularly true for PI4K2A, which is predicted to generate PI4P. How does this phosphoinositide promote the role of Cln3 in ALR?

2. The other major finding, that Cln3 regulates MPR transport, is even less developed. Tracking a pool of MPR from the cell surface, the study shows that Cln3 regulates MPR transport from the EE to the Golgi. However, little insight is provided on how Cln3 regulates this transport. A recent study has found that Cln3 regulates the endosomal transport of MPR to the TGN through the involvements of Rab7, retromer, and sortilin (JCS, 2020; doi:10.1242/jcs.234047). Given these findings, the current study needs to provide better mechanistic insight into how Cln3 acts in transport from the EE to the Golgi. Identifying new factors and elucidating they act in mediating Cln3 regulation of MPR transport would represent a significant advancement.

Reviewer #3 (Remarks to the Author):

Calcagni et al.,

THE BATTEN DISEASE PROTEIN CLN3 IS AN INTEGRAL COMPONENT OF LYSOSOMAL 2 BIOGENESIS AND REFORMATION MACHINERY.

Mutations in CLN3 cause a severe childhood onset, neurodegenerative lysosomal storage disorder called Batten disease, where the disease mechanism remains enigmatic. In this study Calcagni et al report that CLN3 is a multifunctional lysosomal protein that modulates critical functions including autophagic lysosome reformation to generate proto-lysosomes and maintain lysosome homeostasis.

Mechanistically CLN3 regulation of ALR requires interaction with BORC-complex subunit Lyspersin and the phosphatidylinositol 4-kinase PI4K2A. CLN3 also interacts with the mannose 6 phosphate receptor, which coordinates the targeting of lysosomal enzymes to lysosomes. CLN3 depletion impairs M6PR retrograde PM-to-Golgi sorting leading to mistargeting of lysosomal enzymes. Altogether these findings work towards explaining the impaired lysosomal function in Batten disease.

A major strength of this study is that the link between Batten disease and inhibition of autophagic lysosome reformation (ALR) is highly interesting and very topical, given the rapid emergence of ALR defects in human disease that is generating substantial recent interest. However, substantial further

experiments are needed to substantiate a role for CLN3 in ALR (refer comments below). Additionally, the introduction is very brief and would be improved with a definition/description of autophagic lysosome reformation and its relationship to autophagy functions, given it is a major focus of this work. The introduction also does not acknowledge progress on ALR research for other lysosome storage disorders including Scheie syndrome (Mucopolysaccharidosis type I) Fabry disease, Aspartylglucosaminuria and Gaucher disease (PMIDs: 20526321, 27378698). Further to this, the authors show that CLN3 regulation of ALR requires its binding partners BORC and the PI4K2A, both of which have published roles in ALR which is not acknowledged in this manuscript (PMID: 33618608, 33629936). It would be appropriate to include discussion of these findings within the context of this established work.

Major Points:

There are several critical issues that question whether CLN3 is required for autophagic lysosome reformation.

First, examination of tubules emanating from LAMP1-positive structures, is not definitive to conclude effects of CL3 on ALR (Figures 3A-C). This is because during ALR, reformation tubules extend from autolysosomes (LC3+/LAMP1+), however, lysosomes (LC3-/LAMP1+) can also form tubules that function during lysosome reformation. To verify ALR effects, LC3 and LAMP1 co-staining experiments are needed in the CLN3 gain- and loss- of function cell models. Additionally, if these membrane tubules are related to ALR, and hence autophagy dependent, then autophagy suppression in CL3 overexpression cells should block the increased formation of reformation tubules.

Second, key definitive features of ALR inhibition are lacking from CLN3 KO cells. It is known that ALR is not active under fed conditions, but is required for lysosome homeostasis during prolonged starvation-induced autophagy. As a consequence, the disruption of ALR regulatory proteins does not affect lysosome numbers under fed conditions, but lysosomes become progressively depleted following starvation. CLN3 KO cells do not exhibit this highly characteristic effect, as lysosome numbers do not differ between Fed and fasted states. Further to this, the images in Figure 3C do not support data on lysosome numbers shown in the graph – in the images presented, lysosomes appear increased in CLN3 KO cells compared to WT and CL3-gain of function cells.

Third, a second major definitive feature of ALR inhibition is the pervasive accumulation of highly enlarged LAMP1-positive structures (enlarged autolysosomes). While the graph in Figure 3 measuring lysosome area suggests a minimal size increase in CL3 KO cells under fasting conditions; when ALR is inhibited these should be very prominent and significantly larger. The images of LAMP1 staining in CL3 KO cells also do not appear to show any highly enlarged LAMP1 structures that would be consistent with ALR inhibition.

Issues with the specificity of the new CLN3 antibody to examine endogenous CL3 localization and expression is also questioned. Immunoblot analysis in three independent CLN3 KO HeLa cells still

consistently identifies one/multiple bands – whether this represents CLN3 (ie. KO is unsuccessful) or non-specific detection of additional proteins is uncertain as molecular weight markers are not shown (Supplementary Figures 1e and 2a). Either way, this is problematic in terms of antibody specificity and/or validation of the CL3 KO cells. Furthermore, immunostaining with the CLN3 antibody in CLN3 KO HeLa cells also still detects punctate structures with some overlap with LAMP1 at lysosomes (Supplementary Figure 1d). Given this, what assurances are there that the punctate CLN3 localization in Figure 1a-b and Supplementary Figure 1a-c does not represent detection of this non-specific protein? These CL3 antibody issues are also possibly impact Supplementary Fig. 2B-E.

Additional points:

1. The localization of CL3 at the Golgi is convincing, the presence of CL3 at lysosomes less so and should be conclusively verified by immuno-EM. Additional localization studies are needed to independently verify that recombinant-tagged CLN3 localizes the Golgi and lysosomes, therefore corroborating the localization pattern observed for the endogenous protein. The conclusion that “CLN3 traffics through the Golgi and lysosomal compartments” is not supported by the data shown, which identifies the presence of CLN3 at these sites and no evidence of trafficking effects per se.
2. Supplementary Figure 2e: CLN3 mRNA levels need to be examined via quantitative RT-PCR to verify effects are due to post-translational degradation. Is CLN3 degradation lysosome- or autophagy-dependent or both? Bafilomycin studies do not differentiate between these two possibilities and is a critical question give the identified role for CLN3 in ALR. The authors should examine CLN3 expression in autophagy-deficient cells such as Atg5 or Beclin KO cells. Are there labels missing from the top of supplementary Figure 2e, as only three of the four rows of +++ and --- are labelled (ie. WT, MG132 and Bafilomycin)?
3. Supplementary Figure 3b: IP HA, the HA immunoblot is poor quality and HA-CLN3 is not clearly discernable. Molecular weight markers are needed on all immunoblots in Supplementary Figure 3 to verify precipitated proteins are running at their expected molecular weights.
4. Figure 3C shows that gain of CLN3 results in increased formation of lysosome reformation tubules during ALR, while their formation is reduced in CLN3 KO cells. One interpretation of this data that has not been considered is that, CLN3 controls ALR initiation via effects on mTOR reactivation during autophagy (PMID: 20526321). This is an important consideration given mTOR activation is reduced in CLN3 KO mice (PMID: 16714284). To strengthen their subsequent mechanistic studies, the authors should exclude that CLN3 affects mTOR reactivation during prolonged starvation-induced autophagy, and thereby does not impact initiation of the ALR process per se. This would include phospho-mTOR immunoblot analysis to examine mTOR reactivation in cell lysates and immunostaining for locally active mTOR (phosphorylated) on LC3/LAMP1 immunostained autolysosomes.

5. If ALR is inhibited then autophagosomes accumulate during starvation-induced autophagy, because there are not enough lysosomes to fuse with. In contrast, increased ALR and therefore enhanced lysosome production, should increase the maturation of autophagosomes. The authors should examine effects on autophagosomes in CLN3 loss- or gain- of function cells, to ascertain if this is consistent with functional effects on ALR.

6. Are the images in Supplementary Figure 4h labelled correctly, as there appear to be more reformation tubules in CLN3 KO cells without reconstitution of CLN3, compared to those re-expressing CLN3? Furthermore, despite the graph showing lysosome numbers are increased in CLN3 KO cells when CLN3 protein is re-expressed, this is inconsistent with the cell images in Supplementary Figure 4e and 4h which clearly show no differences in lysosome numbers in CLN3 KO cells +/- CLN3 re-expression. Data is internally inconsistent here.

7. All experiments use CLN3 KO cells. As Batten disease is frequently caused by CL3 missense mutations, authors should consider examining effects of CL3 mutants of ALR, to improve the disease relevance of their novel pathway.

Reviewer #4 (Remarks to the Author):

This interesting manuscript reports functions for a lysosomal protein, CLN3, in lysosome tubulation and autophagic lysosomal reformation (ALR) through interaction with the BORC complex and PI4K2A. This part of the study is novel, strong and interesting. The authors go on to describe a complementary role for CLN3 in CI-M6PR-mediated delivery of lysosomal enzymes. The manuscript is well-written, thorough and well presented and the study provides insight into the role of CLN3-mediated trafficking pathways and lysosomal function in the pathogenesis of Batten disease.

However, I didn't see evidence of a role for CLN3 in "de novo lysosomal biogenesis" as claimed. The defective trafficking of lysosomal enzymes in CLN3 knockout cells is consistent with previously reported findings that maturation and delivery of lysosomal protease are reduced in cells lacking CLN3 due to defective CI-M6PR traffic (PMID 18817525). This paper is of clear relevance and should have been cited. The effect shown here of CLN3 overexpression on pepstatin, which binds cathepsin-D builds on this established role of CLN3 in delivery/maturation of lysosomal enzymes, but I'm not convinced it reflects de novo lysosomal biogenesis.

Specific comments:

- It would be nice to see markers of ALR at tubulating lysosomes in CLN3 overexpressing cells.

- The interaction between CLN3 and PI4K2A is interesting. This may be beyond the scope of this study but is PI4K2A mis-localised in CLN3 KO cells and is endo-lysosomal PI4P affected? This could provide insight into the mechanism of CLN3-mediated tubulation and fission. The potential importance of PI4P in tubulation is discussed in the context of PI4P being a precursor for PI(4,5)P₂, but PI4P production also drives PS enrichment at the endosome for the recruitment of fission machinery; similar mechanisms may occur at tubulating LE/Lys.
- The CI-M6PR assay only looks at a subpopulation of protein internalized from the PM - would be nice to see global distribution. It is suggested that CI-M6PR is diverted away from the Golgi to the PM but is reduced at the PM - is that because it is so rapidly internalized? Any why is CI-M6PR total protein so reduced? Couldn't this alone explain the reduction in efficiency of lysosomal enzyme delivery?
- Lysosomal enzymes appear reduced in the lysosome, but increased in the medium. Is this due to loss of M6PR retrograde traffic to the Golgi resulting in increased secretory traffic from the TGN? Or increased lysosome exocytosis? I assume the former but it would be nice to show if possible.
- Fig. 6a) Assuming that GAA/LAMP colocalization is shown in white, by eye it appears increased in the KO image shown and LAMP1/EEA1 appear to both be staining several GAA +ve compartments. Do you have a different image to better reflect the quantitation?
- Fig. 6b) The legend says 30 and 60 min but the fig says T0 and 60min. Please clarify what T0 means. In the KO. The Y axis label isn't visible.
- Fig. 6c) This is very clear data but please add details of the dextran incubation to the legend (it's in the methods but easier to see in the legend).
- Fig. 6d) The PepA experiments need repeating in KO and WT cells to be able to draw any conclusions.
- In the discussion, line 285 reads "CLN3 localizes at the Golgi where it regulates the sorting of CI-M6PR". Your data seems to be more suggesting that CLN3 is predominantly functioning at the endosome to sort CI-M6PR to the Golgi.

Reviewers' comments:

Reviewer #1 (Remarks to the Author):

Summary:

This paper reports two roles for CLN3 that are linked to maintaining proper lysosome function. First, they report that CLN3 plays a role in promoting lysosome biogenesis through ALR. Their CLN3 overexpression cell culture system displays a significant increase in lysosome tubulation in conditions that induce ALR (prolonged serum/glutamine starvation). In contrast, CLN3 KO cells exhibit markedly reduced lysosome tubulation compared to that of wildtype and CLN3-overexpressing cells. The second role they assign CLN3 is its role in PM-to-Golgi retrograde trafficking. They identify CI-M6PR as an interactor of CLN3 and show that CLN3 mediates retrograde transport of CI-M6PR to the Golgi. However, in CLN3-deficient cells, this retrograde transport is impaired, resulting in mislocalization of CI-M6PR on early endosomes. Consequently, this trafficking defect impairs proper sorting of lysosomal enzymes, contributing to reduced lysosomal degradative capacity. The authors propose that lysosome dysfunction, as a result of the trafficking defect caused by CLN3-deficiency, may be a disease mechanism of Batten disease and potentially other neuronal ceroid lipofuscinoses. MFSD8/CLN7 mutations (also a NCL disease) cause ALR defects (PMID: 29514215) and like CLN3 a lysosomal membrane protein. As such, this isn't the first example of a link between CLN genes and ALR.

We agree with the reviewer, this is not the first example of a link between CLN genes and ALR, but we consider this to be an important observation. NCL diseases share common pathogenic traits (e.g. cognitive and motor degeneration, vision loss) and show accumulation of lipofuscins in the brain, but there is no clear functional connection among many of the NCL genes. Our findings reveal a novel link between the M6P-dependent trafficking of lysosomal enzymes and the lysosomal reformation pathway, explaining the global impairment of lysosomal function observed in Batten disease. Finally, we discovered that CLN3 plays an important role in the regulation of both lysosome tubulation and reformation, which was never reported for other CLN genes.

Major Comments:

- The authors describe a lysosome biogenesis pathway via CLN3-mediated ALR. TFEB-dependent lysosome biogenesis is not mentioned anywhere in the manuscript. Presumably, the starvation condition (serum/glutamine starvation) used to induce ALR also activates TFEB nuclear translocation. Where does TFEB fit in this picture? How is TFEB affected by CLN3 KO?

Prolonged starvation initially promotes mTOR inhibition, and subsequently its activation through the release of nutrients from lysosomes, and this process is crucial for activation of ALR (PMID 20526321). mTORC1-mediated phosphorylation inhibits TFEB by promoting its cytoplasmic retention (PMID 22343943, 22692423, 22576015). In line with this, we observed that prolonged (i.e. 16h) starvation resulted in a partial TFEB re-phosphorylation, which is shown by a shift in TFEB molecular weight (Rebuttal Fig.1). This is consistent with the kinetics of mTOR reactivation observed in WT cells (see Fig.6c). Notably, in CLN3-KO cells we did not detect any changes in TFEB phosphorylation over time (rebuttal Fig.1) or mTORC1 re-activation (Fig.6C), in line with a failure of the ALR process.

Rebuttal Fig.1

ARPE19 cells were starved for serum and glutamine for the indicated time, and analyzed by western blot using the indicated antibodies.

- Autophagy control – in cells deficient for autophagy (e.g. ATG5 knockout), is CLN3-mediated ALR absent? Presumably, ALR requires autophagy – as the name implies – so the prediction here would be that autophagy-deficient cells do not display ALR. This control may help distinguish whether lysosomal tubulation is autophagy-dependent (ALR) or whether it is lysosomes tubulating independent of autophagy.

We thank the reviewer for this important suggestion. We performed the experiment as suggested, and silenced ATG7 in CLN3-overexpressing cells. We observed that blocking autophagy completely abolished lysosomal tubulation and reformation (Fig.7c). This result confirmed that CLN3-mediated lysosomal tubulation is dependent on autophagy.

- Line 85: Authors claim that N/C terminal tagging of CLN3 causes mis-localization. Please add citation to papers that prove this point or show this mis-trafficking in the figures.

CLN3 mis-localization upon N/C-ter tagging was reported in literature, and referenced were included in the revised version of the text (PMID: 10191111, 10749980). We repeated CLN3 co-staining with Golgi and lysosomal markers and confirmed that the C-ter CLN3 tagging is highly mis-localizing the protein in the ER. The N-ter tagging approach resulted in a partial CLN3 delocalization, as many cells showed CLN3 dispersion in the cytoplasm, but in some cells a portion of the protein still displays a more reliable Golgi or lysosomal distribution (see Rebuttal Fig. 2).

Rebuttal Fig.2

ARPE19 cells were transfected with either CLN3-Myc or Myc-CLN3 and stained for the indicated antibodies. All scale bars are 20µm.

- A previous study (PMID: 20526321) reported that ALR requires reactivation of mTOR. How is mTOR reactivation affected in the context of CLN3 KO? Is mTOR reactivation impaired in CLN3 KO?

We checked mTOR phosphorylation status over time upon progressive removal of serum and glutamine. As reported above (see rebuttal Fig.1, Fig.6c), we observed that WT cells reactivated mTOR after 16h of starvation, whereas CLN3-KO cells showed defective mTOR reactivation after prolonged starvation.

- Based on still images – I am not convinced that their lysosomal tubulation measurements are accurate. Subcellular structures quantified as tubules on still images seem arbitrary. Based on what criteria were structures quantified as ALR tubules? Maybe the movies are more convincing? Morphometric analyses of lysosomes and lysosomal tubules were performed with the Imaris software. Total lysosomes (Dextran channel) were identified with the Imaris ‘surface’ module: for thresholding, image smoothing, and background subtraction were applied. Surfaces detail (0.2µm) and diameter of largest sphere (0.150µm) were determined (threshold values were kept constant along images). Lysosomal structures smaller than 0.05µm were filtered out. Lysosomal tubules were also identified with the Imaris “surface” module, applying a sphericity and ellipsoid filter to lysosomal masks, which allowed to filter out non-tubular lysosomal structures based on their shape. Rebuttal Fig. 3 shows reference images before and after the lysosomal and tubule masks were applied.

Rebuttal Fig.3

Representative pictures indicating the different image processing steps for tubules identification. All scale bars are 10 μ m. LM, lysosomal mask; TM, tubules mask.

Movies are also included in the supplementary information file. Tubules are clearly visible there, as in the case reported here (Rebuttal Fig. 4, Supplementary Movie 1)

Rebuttal Fig. 4

Snapshot from a movie showing tubular lysosomes (LAMP1-mCherry). Scale bar is 20 μ m.

- Line 123 ff: Authors claim that overexpression of CLN3^{innHA} constructs causes excessive ALR. Could the phenotype be due to overexpression, since cells expressing CLN3^{innHA} appear to contain more dextran in lysosomes /& tubules (Suppl. Fig 4a). Given that the authors previously claim that N/C terminal fusion of CLN3 are mis-trafficked, please add these as controls to the ALR experiments.

We think that the ideal control for this experiment, should be a CLN3 plasmid that is still able to localize at the lysosome, but is missing the ability of inducing CLN3-mediated functions. Instead of using the N/C-ter tagged CLN3 constructs, we mutagenized the CLN3 plasmid with a missense pathogenetic mutation (nt. G1001A, p.Arg334His, exon 13, PMID: 9311735), and cloned it in the same inducible lentiviral backbone used for the tubulation experiments. Cells transduced with the mutant vector revealed that CLN3^{G1001A} is still able to reach lysosomes, but it is also quite dispersed in the cytoplasm. In these cells, even if mutant CLN3 localizes on lysosomes, no tubulation was observed. Also, no tubules were observed from CLN3-KO cells transduced with CLN3^{G1001A}. These results indicate that the hyper-tubulation and lysosome reformation phenotype is the result of a specific CLN3 function (Rebuttal Fig. 5).

Rebuttal Fig. 5

Cells transduced with the Tet-on CLN3^{G1001A} lentiviral vector, induced with doxycycline for 40h and starved (serum+glutamine) for 16h. Scale bar is 20µm.

- Previous studies have shown that addback of 10% FBS to starved cells for as little as 15 min can robustly induce ALR (PMID: 33119550). Given the problems with the tagging of CLN3 and the potential of overexpression artifacts, the authors should test if this method would be an alternative to measure ALR deficiency in CLN3 knockouts.

We thank the reviewer for the interesting suggestion. We performed the experiment by starving cells for 8h with Ebss, but unfortunately, we were unable to detect a strong induction of tubulation after 15' or 30' of serum addition, even in CLN3-overexpressing cells. Perhaps this protocol needs further optimization in this cell line.

- Authors claim that CLN3 is integral to lysosomal reformation and show that KO has no lysosomal tubules (Fig 3c). However, CLN3 KO shows extensive tubulation in representative images depicted in Supplementary Figure 4h, which is either an error in the figure annotation or would contradict the authors previously stated conclusions. Please elaborate on this discrepancy.

We apologize for the error. The images were inverted. The new figure has been adjusted and results are now reported in Figure 6e.

- Results related to CI-M6PR recycling is confusing. Figure 4C indicate that WT cells have more CI-M6PR on PM compared to that of CLN3 KO. This appears to contradict the results in Figure 4D where more CI-M6PR is seen on PM fraction in CLN3 KO cells.

As stated by the reviewer, in Figure 3e (previously reported as 4c) we observed that CLN3-depleted cells have less PM CI-M6PR at the steady-state, as this measurement was performed by incubating cells with the lumCI-M6PR antibody at 4 °C. To rule out a defect in the recycling of the

receptor to the PM, we loaded cells again with the lumCI-M6PR antibody, but this time the incubation was performed at 37°C, allowing its internalization, and then we measured only the PM fraction by FC (Supplementary Figure 4c). Normally, most of the internalized receptor is delivered to the Golgi compartment, and only a small fraction recycles back to the PM. In KO cells we observed that the amount of PM receptor was normal, ruling out recycling defects (Supplementary Fig. 4c). Supplementary Figure 4b also shows that in KO cells the PM levels of CI-M6PR recover at 60 minutes. Thanks to the point raised by the reviewer, we noticed that total levels of CI-M6PR were significantly reduced in CLN3-KO cells, and their levels went back to normal upon inhibition of lysosomal degradation with bafilomycin (Figure 3a, b). Also, bafilomycin treatment revealed that a portion of CI-M6PR, which in WT cells is detectable at the lysosomal membrane, accumulated within the lysosomal lumen in CLN3-depleted cells, suggesting mistrafficking of the receptor inside the lysosomes in KO cells (Figure 3c).

o Also, could the authors elaborate on the reduction in total CI-M6PR levels in CLN3 KO cells (Supp Figure 6D)? I would have imagined CI-M6PR being more abundant in CLN3 KO cells because more of the protein appear to be localized on early endosomal structure where presumably there is less chance of turnover, as opposed to being in golgi/lysosome.

This point was partially addressed above. We observed that most of the receptor is degraded inside lysosomes (see above). To better dissect the trafficking of the receptor, we followed its retrograde trafficking from the PM to Golgi, and found that this trafficking is impaired and that less receptor reaches the Golgi apparatus, remaining in peripheral structures which are endosomes (Figure 4a, b, Supplementary Figure 4d, e). This confirms that the trafficking defect happens at the level of the endosome-to-golgi delivery, and that this may cause the receptor redirection to lysosomes. Moreover, we also observed that Cln3 induction promotes accumulation of the receptor at Golgi in an ESCPE-1 dependent manner, indicating that CLN3 may mediate the ESCPE1-dependent Golgi delivery of the receptor (Figure 4d).

o Furthermore, model drawn in Figure 7 shows more CI-M6PR on the PM and endosomes in CLN3 KO. Please elaborate on this discrepancy.

The model has been now modified based on the latest results (Figure 8d).

• Figure 4E – I am not sure what the data is trying to say. Is there a better way to depict this?

This point was addressed above.

• It is unclear whether CLN3-mediated ALR and retrograde transport are linked or not. Does ALR affect retrograde trafficking? If so, how? Maybe this can be tested via depletion of Lyspersin or PI4K2A and perform colocalization studies of CI-M6PR and EEA1/Golgi markers.

We think that the two phenotypes are correlated. CI-M6PR defective trafficking to the Golgi, determines its degradation into the lysosomes and consequent missorting of lysosomal enzymes. The impaired lysosomal degradative capacity, in turn determines failure of mTOR reactivation (Figure 6c), and consequently of ALR (relevant data are reported in Figure 6 and 7), as this is the case of Spinster, whose mutations cause defects in sugar transport and consequently in

lysosomal function, resulting in the failure of mTOR reactivation and ALR (PMID: 21518918). Also, we demonstrated that CI-M6PR is essential to ALR functioning, as CI-M6PR depletion completely abolishes the CLN3-mediated lysosomal tubulation and reformation phenotype (Figure 8c). We therefore concluded that trafficking of CI-M6PR is important for ALR. We cannot exclude that altering ALR function through manipulation of lyspersin and PI4K2A may have consequences on the trafficking of the CI-M6PR, but we think that exploring this point is beyond the scope of this study.

- In general, materials and methods are very general and not detailed enough to understand all the experimental outcomes.

We added experimental details to the methods section and in the figure legends.

- Please add information about cell culture treatments (concentration of media, drugs, dyes & duration).

These experimental details were added in the methods section and/or figure legends.

- Antibody validation is missing.

For the antibody validation, we performed endogenous CLN3 immunopurification, where we can clearly detect the glycosylated band being precipitated only in WT cells (Figure 1f). We could not detect the lower molecular-weight band, as it is detectable at very low levels in the total lysate, and in the IP it overlaps with the band of the IgG heavy chains (Figure 1f). Additionally, in Supplementary Figure 1e and f, we showed that the antibody is able to recognize endogenous CLN3 in WT, but not in KO cells. Finally, in supplementary Figure 2b, we used the antibody to detect CLN3 from purified lysosomes.

- For image analysis matters, more detail about pre-processing, filtering and thresholding for co-localization and quantification is necessary. Which thresholding kernel did the authors use for the segmentation?

All the information has been added to the 'Image analysis' section in the methods section.

- High-content imaging is not mentioned in the materials and method. On which system was it performed?

High-content imaging was performed on a confocal automated imager OPERA, from PerkinElmer, and the experimental details were added in the "Confocal microscopy" section.

Minor comments

- The authors claim to have generated a new CLN3-specific antibody. However, the authors do not show any of the validation and testing of the antibody. Please incorporate into the supplement

This point was addressed above. In the new version of the manuscript, we validated the antibody by performing endogenous CLN3 IP. Validation of the antibody in WT and CLN3-KO cells by immunoblot is shown in Supplementary Figure 1 e and f. Also, in supplementary Figure 2b, we used the antibody to detect CLN3 from purified lysosomes.

- Knockouts genomic sequencing information about be included in the supplemental figure and not in the materials and methods. Authors state that cells containing the homozygous mutation (c.1055 delA) were expanded, but two clones are show in the supplementary figure 1. Which clone was used for the later figures? Please indicate in text or materials.

Information about the mutation was added in the figure legend (Supplementary Figure 1f) and in the methods. Also, we noticed that the mutation was not annotated correctly. We apologize for the error. The position of the premature stop codon is c. 109delA. We corrected the information in the methods and in the legend.

Interestingly, RNA levels are ~50% reduced in KOs (Suppl. Fig. 1g). Is there an explanation for this?

CLN3-KO cells carry a deletion that causes a premature stop codon in the gene sequence, but the gene can still be transcribed. Reduction of the transcript may be due to nonsense mediated decay.

- What were the CLN3innHA expression levels used in the experiments. Authors fail to disclose dox-concentration, duration of induction or WB on protein levels. What happens if you rescue CLN3 closer to WT levels?

Details about Dox-concentration and duration of induction have now been added to the methods. For our experiments we induced cells with 1µg/ml doxycycline for 40h. Induction of CLN3 transcript with the lenti-TetON system is of about 50 fold, we tried to obtain lower transduction levels by using scalar dilutions of the virus or of doxycycline, but we could never go below this level of induction.

- Style of boxplots/scatterplots is not consistent throughout the paper. Please change all boxplots to either scatterplots or boxplots + scatterplots, to show the underlying data-distribution.

We changed all plots to scatterplots throughout the paper.

- Line 448: CI-M6PR assay: Why did authors change the cited fixation method for their experiments? Cited paper states the fixation with MeOH, but authors choose to fix with PFA and then MeOH. Could they elaborate on their reasoning behind this?

In the CI-M6PR recycling assay, the PFA fixation step is not required for the recognition of CI-M6PR antibody by secondary antibodies, but it is critical for the colocalization analysis. We tried fixing cells just with methanol, but that caused failure of some co-stainings with endosomal or TGN markers, as these antibodies work better after PFA fixation. The addition of a fixation step with ultra-pure PFA, allowed a better preservation of these epitopes, without affecting the signal of CI-M6PR antibody.

- Size of scale bars for image insets are not mentioned or depicted. Please add.

All the scale bar sizes are now reported in the figure legends.

- Fig 4e: Y-axis of FACS histogram are not matching between WT and KO. Please adjust.

The images have been adjusted, and results are now shown in Supplementary Figure 4c.

- Fig 5a: Authors state that “several mitochondrial proteins” accumulate in CLN3 lysosomes, but these are not annotated in the volcano plots. Please add labels for proteins of interest.

This information is now reported in Supplementary Figure 5a.

- Fig 6b: Cell masks indicated are based on what? Some cells in the example image of the CLN3 KO were apparently discarded (no highlight of cell and nucleus) for the analysis, based on what metrics?

Imaris software was used to create cell masks based on the CellMask™ Plasma Membrane Stain. Whenever it was not possible for the software to discriminate between two adjacent cells (when cells were merged, or contained part of the cytoplasm of a different cell), we discarded them to avoid errors in the final count.

- Fig 6b: Image panel overlaps into evaluation plot

The figure (now Figure 5e) was adjusted.

- Suppl. Fig 1d: CLN3 signal is clearly visible in the HeLa KO and overlaps with Lamp1 signal. Why is Lamp1 all peri-nuclear, but in later figures distributed over the whole cell body?

This is a processing error, we apologize for the mistake, the same channel intensities are now shown in the two figures (Supplementary Figure 1d). Regarding the point relative to LAMP1, the image on the right is a crop of the one on the left.

- Suppl. Fig 2c/d: Authors show one quantification of WB band intensities, but in Suppl. Fig. 4g they show the same analysis with more than n=1. Please add more repeats to analysis of Suppl. Fig 2c/d.

We decided to keep only Supplementary Figure 2c and added more replicates to it.

- Suppl. Fig 4c: Lamp1 spots are normalized per well, but should be normalized to number per cell. Otherwise, this could skew the data if one replicate has more cells / well than the other. Please change.

The number of spots in this experiment were normalized per cell, and these values were then normalized per well. This information was added in the figure legend.

- Suppl. Fig 6a: inserts do not match the ROI indicated in the overview images. Some the insets are rotated compared to the overview areas. Please correct.

We apologize for the error, we adjusted all the inserts and results are now shown in Figure 4d

Overall, the paper contains several interesting observations related to CLN3 but there is a lack of coherency in the story, and some aspects of the paper appear to be of limited robustness. The paper would be strengthened if a link between ALR and retrograde transport could be found. As it stands, the two processes are disjointed and there isn't a molecular explanation that paints a clear pathway for CLN3.

As discussed above, we demonstrated that CI-M6PR is essential to ALR functioning, as CI-M6PR depletion completely abolish the CLN3-mediated lysosomal tubulation and reformation

phenotype (Figure 8c). This represents a functional link between CLN3 role in CI-M6PR trafficking (and targeting of lysosomal enzymes) and ALR.

Reviewer #2 (Remarks to the Author):

This manuscript seeks to elucidate the cellular roles of Cln3. Mutations in Cln3 are associated with Batten's disease, a form of lysosomal storage disorder. Thus, understanding how Cln3 acts has significant disease implications. This study has two main findings. First, Cln3 acts in autophagic lysosome reformation (ALR). Second, Cln3 regulates the endosomal transport of mannose-6-phosphate receptor (MPR).

To gain insight into how Cln3 acts in ALR, a proteomic approach was undertaken, which identified many Cln3 interactors. Some of these are transport factors acting in endosomal pathways. Functional studies identified two being relevant for Cln3 acting in ALR, lyspersin (a component of the BORC complex) and PI4K2A. For MPR transport, this study shows that Cln3 regulates the retrograde transport of MPR from the early endosome (EE) to the Golgi. Perturbing this regulation reduces the delivery of lysosomal proteins to the lysosome. The study also generated an antibody, which appears for the first time to be able to detect endogenous Cln3. Using this antibody, the study detects Cln3 to reside predominantly at the lysosome and the Golgi. Although these findings provide new information on Cln3, they represent limited insights.

Major comments:

1. The main novelty of this study is the finding that Cln3 acts in ALR. Although lyspersin and PI4K2A were identified to be needed for this role of Cln3, how these factors act in this context needs to be clarified. This is particularly true for PI4K2A, which is predicted to generate PI4P. How does this phosphoinositide promote the role of Cln3 in ALR?

In this revised version of the manuscript, we show that CI-M6PR is of crucial importance in CLN3-mediated ALR process. Specifically, we report that CLN3 is degraded inside lysosomes in CLN3-KO cells (Figure 3), affecting lysosomal content and degradative capacity (Figure 5), and resulting in failure of mTOR reactivation (Figure 6c), and consequently ALR (Figures 6, 7). This is also the case of Spinster, whose mutations cause defects in sugar transport and consequently in lysosomal function, resulting in the failure of mTOR reactivation and ALR (PMID: 21518918). Finally, we show that CI-M6PR is required for ALR to happen, as its depletion completely abolished CLN3-mediated lysosomal tubulation and reformation phenotype (see Figure 8c). Concerning the potential role of PI4K2A, we agree with the reviewer that our data are too preliminary to indicate that it participates in mediating the effect of Cln3 in ALR. Exploring this would require an extensive set of experiments, which are beyond the scope of this manuscript.

2. The other major finding, that Cln3 regulates MPR transport, is even less developed. Tracking a pool of MPR from the cell surface, the study shows that Cln3 regulates MPR transport from the EE to the Golgi. However, little insight is provided on how Cln3 regulates this transport. A recent study has found that Cln3 regulates the endosomal transport of MPR to the TGN through the

involvements of Rab7, retromer, and sortilin (JCS, 2020; doi:10.1242/jcs.234047). Given these findings, the current study needs to provide better mechanistic insight into how Cln3 acts in transport from the EE to the Golgi. Identifying new factors and elucidating they act in mediating Cln3 regulation of MPR transport would represent a significant advancement.

We thank the reviewer for this important observation. The revised version of the paper contains several additional data relevant to the trafficking of the receptor. We observed that total levels of CI-M6PR were significantly reduced in CLN3-KO cells, and their levels went back to normal upon inhibition of lysosomal degradation with bafilomycin (Figure 3a, b). Also, bafilomycin treatment revealed that a portion of CI-M6PR, which in WT cells is detectable at the lysosomal membrane, accumulated within the lysosomal lumen in CLN3-depleted cells, suggesting mistrafficking of the receptor inside the lysosomes in KO cells (Figure 3c). We also detected an impairment in the sorting from the PM to Golgi, with decreased amount of receptor reaching the Golgi apparatus (Figure 4a, b, Supplementary Figure 4d, e). This confirms that the trafficking defect happens at the level of endosome-to-Golgi delivery. Furthermore, CLN3 induction was able to enhance the Golgi delivery of the receptor (Figure 4), strengthening our observations in CLN3 KO cells. Considering that the delivery of the CI-M6PR to the Golgi compartment is modulated by the retromer and ESCPE-1 complexes (PMID 17606993, 28935633), we then checked whether CLN3-mediated Golgi delivery may be modulated by their depletion. We found that the enhanced CLN3-dependent Golgi delivery of the receptor was partially rescued by silencing of the ESCPE-1 complex, but was mostly unaffected by retromer depletion (Figure 4d). Of note, several components of the ESCPE-1 complex were also identified among CLN3 interactors, indicating that it may cooperate with CLN3 in the recycling of the receptor at Golgi. These results corroborate the relevance of CLN3 in the endosome-to-Golgi sorting of the receptor and that CLN3 and the ESCPE-1 complex are both coordinating this process.

Reviewer #3 (Remarks to the Author):
Calcagni et al.,

THE BATTEN DISEASE PROTEIN CLN3 IS AN INTEGRAL COMPONENT OF LYSOSOMAL 2 BIOGENESIS AND REFORMATION MACHINERY.

Mutations in CLN3 cause a severe childhood onset, neurodegenerative lysosomal storage disorder called Batten disease, where the disease mechanism remains enigmatic. In this study Calcagni et al report that CLN3 is a multifunctional lysosomal protein that modulates critical functions including autophagic lysosome reformation to generate proto-lysosomes and maintain lysosome homeostasis. Mechanistically CLN3 regulation of ALR requires interaction with BORC-complex subunit Lyspersin and the phosphatidylinositol 4-kinase PI4K2A. CLN3 also interacts with the mannose 6 phosphate receptor, which coordinates the targeting of lysosomal enzymes to lysosomes. CLN3 depletion impairs M6PR retrograde PM-to-Golgi sorting leading to mistargeting of lysosomal enzymes. Altogether these findings work towards explaining the impaired lysosomal function in Batten disease.

A major strength of this study is that the link between Batten disease and inhibition of autophagic lysosome reformation (ALR) is highly interesting and very topical, given the rapid emergence of ALR defects in human disease that is generating substantial recent interest. However, substantial further experiments are needed to substantiate a role for CLN3 in ALR (refer comments below). Additionally, the introduction is very brief and would be improved with a definition/description of autophagic lysosome reformation and its relationship to autophagy functions, given it is a major focus of this work. The introduction also does not acknowledge progress on ALR research for other lysosome storage disorders including Scheie syndrome (Mucopolysaccharidosis type I) Fabry disease, Aspartylglucosaminuria and Gaucher disease (PMIDs: 20526321, 27378698). Further to this, the authors show that CLN3 regulation of ALR requires its binding partners BIRC and the PI4K2A, both of which have published roles in ALR which is not acknowledged in this manuscript (PMID: 33618608, 33629936). It would be appropriate to include discussion of these findings within the context of this established work.

We thank the reviewer for raising important points, as they provided important insights to the story. In the revised version of the paper we have integrated all the suggestions of the reviewer in both the introduction and discussion sections.

Major Points:

There are several critical issues that question whether CLN3 is required for autophagic lysosome reformation.

First, examination of tubules emanating from LAMP1-positive structures, is not definitive to conclude effects of CL3 on ALR (Figures 3A-C). This is because during ALR, reformation tubules extend from autolysosomes (LC3+/LAMP1+), however, lysosomes (LC3-/LAMP1+) can also form tubules that function during lysosome reformation. To verify ALR effects, LC3 and LAMP1 co-staining experiments are needed in the CLN3 gain- and loss- of function cell models. Additionally, if these membrane tubules are related to ALR, and hence autophagy dependent, then autophagy suppression in CL3 overexpression cells should block the increased formation of reformation tubules.

As the reviewer suggested, we performed live confocal imaging of cells transduced with LAMP1-mCherry and transfected with LC3-GFP and we could detect the presence of tubules in CLN3-overexpressing cells budding from LC3-LAMP1+ lysosomes (Supplementary Figure 8d). Because this process is very dynamic, and LC3 is constantly degraded within lysosomes after autophagosome fusion, we also detected free tubules. Therefore, to determine whether CLN3-mediated lysosomes tubulation is relying on autophagy, we silenced ATG7 and observed complete abolishment of lysosome tubulation and reformation, together with enlargement of lysosomes (Figure 7c). This experiment confirmed that CLN3-mediated lysosomal reformation is autophagy dependent.

Second, key definitive features of ALR inhibition are lacking from CLN3 KO cells. It is known that ALR is not active under fed conditions, but is required for lysosome homeostasis during prolonged starvation-induced autophagy. As a consequence, the disruption of ALR regulatory proteins does

not affect lysosome numbers under fed conditions, but lysosomes become progressively depleted following starvation. CLN3 KO cells do not exhibit this highly characteristic effect, as lysosome numbers do not differ between Fed and fasted states. Further to this, the images in Figure 3C do not support data on lysosome numbers shown in the graph – in the images presented, lysosomes appear increased in CLN3 KO cells compared to WT and CL3-gain of function cells.

Images in Fig. 7b (previously reported as 3c) have now been updated. In the original figures, a lysosomal mask was applied in green (labelled as ‘total lysosomes’), which made it difficult to appreciate differences in size and number of lysosomes among conditions. We are now showing images only with lysosomal tubules outlined in red. Lysosomes number did not differ between the basal and starvation condition. We think that this due to the fact that already in basal condition there is some impairment of lysosomal reformation. Indeed we detected differences in lysosome morphology and size already in basal condition, which are then worsened upon prolonged starvation (Fig.7b, Rebuttal Figure 7).

Rebuttal Figure 7

Dextran-labelled lysosomes in WT and CLN3-KO cells in basal and prolonged starvation conditions

Also, EM analysis in basal and HU-treated cells (which blocks cell cycle), clearly showed signs of ALR failure, as lysosomes were fused in KO cells already in the basal condition (Figure 6a) and lysosomes fusion and aggregation became more pronounced in the HU-treated condition (Figure 6b). Also, quantification of EM data showed that autolysosomes accumulated and that lysosomes size increased, both features being an indication of impaired ALR (Figure 6a).

Third, a second major definitive feature of ALR inhibition is the pervasive accumulation of highly enlarged LAMP1-positive structures (enlarged autolysosomes). While the graph in Figure 3

measuring lysosome area suggests a minimal size increase in CL3 KO cells under fasting conditions; when ALR is inhibited these should be very prominent and significantly larger. The images of LAMP1 staining in CL3 KO cells also do not appear to show any highly enlarged LAMP1 structures that would be consistent with ALR inhibition.

This point was addressed above, especially by EM data, which revealed accumulation of fused autolysosomes (Figure 6a), lysosome fusion and enlargement in HU-treated cells (Figure 6b), and enlarged autolysosomes accumulation in CLN3-KO cells upon prolonged starvation (Figure 6d).

Issues with the specificity of the new CLN3 antibody to examine endogenous CLN3 localization and expression is also questioned. Immunoblot analysis in three independent CLN3 KO HeLa cells still consistently identifies one/multiple bands – whether this represents CLN3 (ie. KO is unsuccessful) or non-specific detection of additional proteins is uncertain as molecular weight markers are not shown (Supplementary Figures 1e and 2a). Either way, this is problematic in terms of antibody specificity and/or validation of the CLN3 KO cells. Furthermore, immunostaining with the CLN3 antibody in CLN3 KO HeLa cells also still detects punctate structures with some overlap with LAMP1 at lysosomes (Supplementary Figure 1d). Given this, what assurances are there that the punctate CLN3 localization in Figure 1a-b and Supplementary Figure 1a-c does not represent detection of this non-specific protein? These CLN3 antibody issues are also possibly impact Supplementary Fig. 2B-E.

Image in Supplementary 1e was overexposed, we replaced it and added molecular weights to all figures. In supplementary figure 1D there was a processing error, same channel intensities are now shown in the two figures (Supplementary Figure 1d). We understand the concerns about the antibody, but we want to outline that this is the first antibody able to recognize the endogenous CLN3 protein. Unfortunately, CLN3 protein is expressed at very low levels. Even if we detect some non-specific signal, the differences between WT and CLN3-KO cells are very clear, and thanks to this antibody we have been able to visualize the protein, examine its degradation rate and stability, immunoprecipitate it and detect it in the lysosomal immunopurified fraction. Therefore, we think that it represents a very important tool in the field, considering that, in spite of intensive research efforts on CLN3, antibodies recognizing the endogenous protein are still unavailable.

Additional points:

1. The localization of CLN3 at the Golgi is convincing, the presence of CLN3 at lysosomes less so and should be conclusively verified by immuno-EM. Additional localization studies are needed to independently verify that recombinant-tagged CLN3 localizes the Golgi and lysosomes, therefore corroborating the localization pattern observed for the endogenous protein. The conclusion that “CLN3 traffics through the Golgi and lysosomal compartments” is not supported by the data shown, which identifies the presence of CLN3 at these sites and no evidence of trafficking effects per se.

We thank the reviewer for the interesting suggestion of using EM. We tried to validate CLN3 localization by EM analysis, but unfortunately the antibody does not work after the EM fixation protocol. CLN3 lysosomal localization is validated in Supplementary Fig.2b, which clearly shows that the antibody only recognizes the specific highly glycosylated band in the lysosomal immunopurification sample. This result also suggests that some non-specific signal is only detected in the

input, whereas the lysosomal signal is very specific. The reviewer also suggested to confirm CLN3 localization in recombinantly-tagged cells and to confirm trafficking of the protein at the two compartments. In Supplementary Figure 1c we confirmed that recombinantly-tagged CLN3 also localizes at Golgi and lysosomes, whereas no localization is observed on endosomes, confirming the results with the CLN3 antibody on endogenous CLN3. In Supplementary Figure 2f we treated pLVX-CLN3-innHA overexpressing cells with a short doxycycline induction protocol (4h Dox induction), to visualize the newly synthesized protein, or with a short dox treatment followed by a long wash-out (4h dox+12h washout), to visualize the mature protein. A schematic of the experiment is reported in Figure 1g. Using this approach we demonstrated that epitope-tagged CLN3 is localized at Golgi and lysosomes (this time the staining was performed with the CLN3 antibody). We also demonstrated that newly synthesized CLN3 (4h dox) is localized at Golgi, and then the mature protein (4h dox+12h washout) reaches the lysosome, with a portion of the mature protein still localizing at Golgi. This confirms that the newly synthesized protein traffics to Golgi, where it matures and gets glycosylated, and subsequently reaches the lysosomes (Figure 1d). Also, in supplementary Fig. 8c we show that epitope-tagged CLN3 localizes at Golgi, lysosomes and lysosomal tubules.

2. Supplementary Figure 2e: CLN3 mRNA levels need to be examined via quantitative RT-PCR to verify effects are due to post-translational degradation. Is CLN3 degradation lysosome- or autophagy-dependent or both? Bafilomycin studies do not differentiate between these two possibilities and is a critical question give the identified role for CLN3 in ALR. The authors should examine CLN3 expression in autophagy-deficient cells such as Atg5 or Beclin KO cells. Are there labels missing from the top of supplementary Figure 2e, as only three of the four rows of +++ and --- are labelled (ie. WT, MG132 and Bafilomycin)?

We observed that CLN3 mRNA levels are slightly and equally induced in both MG132 and Bafilomycin conditions (Supplementary Figure 2e). In Supplementary Figure 2d, each condition is replicated three times, the labels are therefore correct. Also, as suggested by the reviewer, we inhibited autophagy by silencing ATG7 and combined this with bafilomycin treatment. No degradation of CLN3 was observed by silencing ATG7 alone (i.e. without bafilomycin), suggesting that CLN3 degradation is lysosome-dependent and autophagy-independent (Rebuttal Figure 8).

Rebuttal Figure 8
 ARPE19 cells were treated with bafilomycin (12h, 20nM), silenced for ATG7 for 72h, or treated with both siATG7+bafilomycin, and analyzed by immunoblot using the indicated antibodies.

3. Supplementary Figure 3b: IP HA, the HA immunoblot is poor quality and HA-CLN3 is not clearly discernable. Molecular weight markers are needed on all immunoblots in Supplementary Figure 3 to verify precipitated proteins are running at their expected molecular weights. We replaced the HA immunoblot and added molecular weight markers to all blots.

4. Figure 3C shows that gain of CLN3 results in increased formation of lysosome reformation tubules during ALR, while their formation is reduced in CLN3 KO cells. One interpretation of this data that has not been considered is that CLN3 controls ALR initiation via effects on mTOR reactivation during autophagy (PMID: 20526321). This is an important consideration given mTOR activation is reduced in CLN3 KO mice (PMID: 16714284). To strengthen their subsequent mechanistic studies, the authors should exclude that CLN3 affects mTOR reactivation during prolonged starvation-induced autophagy, and thereby does not impact initiation of the ALR process per se. This would include phospho-mTOR immunoblot analysis to examine mTOR reactivation in cell lysates and immunostaining for locally active mTOR (phosphorylated) on LC3/LAMP1 immunostained autolysosomes.

We thank the reviewer for suggesting this experiment. In this revised version of the manuscript, we show that CI-M6PR is of crucial importance in CLN3-mediated ALR process. Specifically, we report that CLN3 is degraded inside lysosomes in CLN3-KO cells (Figure 3), affecting lysosomal content and degradative capacity (Figure 5) and resulting in failure of both mTOR reactivation (Figure 6c) and of ALR (Figures 6, 7). This is also the case of Spinster, whose mutations cause defects in sugar transport and consequently in lysosomal function, resulting in failure of mTOR reactivation and of ALR (PMID: 21518918). Finally, we show that CI-M6PR is required for ALR to happen, as its depletion completely abolished CLN3-mediated lysosomal tubulation and reformation phenotype (see Figure 8c). We also performed LC3/LAMP1 immunostaining on cells upon prolonged starvation, confirming accumulation of enlarged autolysosomes in CLN3-KO cells (Figure 6d).

5. If ALR is inhibited then autophagosomes accumulate during starvation-induced autophagy, because there are not enough lysosomes to fuse with. In contrast, increased ALR and therefore enhanced lysosome production, should increase the maturation of autophagosomes. The authors should examine effects on autophagosomes in CLN3 loss- or gain- of function cells, to ascertain if this is consistent with functional effects on ALR.

This point was addressed above (see reviewer 3, second and third reply to major points, see Fig 6a, 6b and 6d). Accumulation of autophagosome in CLN3-KO cells is also evident in figure 6c (LC3 immunoblot).

6. Are the images in Supplementary Figure 4h labelled correctly, as there appear to be more reformation tubules in CLN3 KO cells without reconstitution of CLN3, compared to those re-expressing CLN3? Furthermore, despite the graph showing lysosome numbers are increased in CLN3 KO cells when CLN3 protein is re-expressed, this is inconsistent with the cell images in Supplementary Figure 4e and 4h which clearly show no differences in lysosome numbers in CLN3 KO cells +/- CLN3 re-expression. Data is internally inconsistent here.

We apologize for the error, the names on the graph were inverted and now they have been adjusted (Figure 6e). Also, we removed lysosomal masks from all tubulation images, as they hampered the interpretation of the results.

7. All experiments use CLN3 KO cells. As Batten disease is frequently caused by CL3 missense mutations, authors should consider examining effects of CL3 mutants of ALR, to improve the disease relevance of their novel pathway.

A CLN3 missense mutation (CLN3 G1001A, p.Arg334His, exon 13), which is causative of NCL3 (PMID: 9311735) was cloned into the same lentiviral Tet-on backbone used for CLN3 overexpression experiments. The anti-CLN3 antibody was able to recognize this mutant, revealing CLN3 mislocalization: most of the protein was dispersed in the cytoplasm, but some protein was still localizing in Golgi or lysosomes. In these cells, no tubulation was observed, indicating that the hyper-tubulation and lysosome reformation phenotype is the result of a specific CLN3 function (Rebuttal Figure 5). This further proves the importance of having generated an antibody that recognizes the CLN3 protein.

Rebuttal Fig. 5

Cells transduced with the Tet-on CLN3^{G1001A} lentiviral vector, induced with doxycycline for 40h and starved (serum+glutamine) for 16h. Scale bar is 20µm.

Reviewer #4 (Remarks to the Author):

This interesting manuscript reports functions for a lysosomal protein, CLN3, in lysosome tubulation and autophagic lysosomal reformation (ALR) through interaction with the BORC complex and PI4K2A. This part of the study is novel, strong and interesting. The authors go on to describe a complementary role for CLN3 in CI-M6PR-mediated delivery of lysosomal enzymes. The manuscript is well-written, thorough and well presented and the study provides insight into the role of CLN3-mediated trafficking pathways and lysosomal function in the pathogenesis of Batten disease.

We thank the reviewer for pointing out the novelties and strengths of our study.

However, I didn't see evidence of a role for CLN3 in "de novo lysosomal biogenesis" as claimed. The defective trafficking of lysosomal enzymes in CLN3 knockout cells is consistent with previously reported findings that maturation and delivery of lysosomal protease are reduced in cells lacking CLN3 due to defective CI-M6PR traffic (PMID 18817525). This paper is of clear relevance and should have been cited. The effect shown here of CLN3 overexpression on pepstatin, which binds cathepsin-D builds on this established role of CLN3 in delivery/maturation of lysosomal enzymes, but I'm not convinced it reflects de novo lysosomal biogenesis.

We thank the reviewer for making these important points. In the revised manuscript we have now cited reference PMID 18817525. We stated that CLN3 is involved in lysosomal biogenesis, because of its role on CI-M6PR sorting and lysosomal enzyme trafficking. Lysosomal biogenesis process is often referred as an orchestration of the structural and functional elements of the lysosome, and involves the coordination of the synthesis, targeting, functional residence, and turnover of the proteins that are responsible for lysosomal function (PMID: 9223373). In this view, CI-M6PR pathway and the delivery of lysosomal hydrolases have also been considered as a lysosomal biogenesis pathway (PMID:19672277). In addition, we have added some critical experiments in the revised version of the paper, which reinforced the relevance of CLN3 in mediating CI-M6PR function, and therefore the delivery/maturation of lysosomal enzymes. Also, CI-M6PR plays an essential role in modulating CLN3-mediated ALR, as its ablation completely abolished lysosome tubulation and reformation process, indicating that lysosomal maturation is a key driver of CLN3-mediated lysosome reformation.

Specific comments:

- It would be nice to see markers of ALR at tubulating lysosomes in CLN3 overexpressing cells. By performing live-imaging of cells transduced with LAMP1-mCherry and transfected with LC3-GFP we detected the presence of tubules in CLN3-overexpressing cells budding from LC3-LAMP1+ lysosomes (Supplementary Figure 8d).

- The interaction between CLN3 and PI4K2A is interesting. This may be beyond the scope of this study but is PI4K2A mis-localised in CLN3 KO cells and is endo-lysosomal PI4P affected? This could provide insight into the mechanism of CLN3-mediated tubulation and fission. The potential importance of PI4P in tubulation is discussed in the context of PI4P being a precursor for PI(4,5)P₂, but PI4P production also drives PS enrichment at the endosome for the recruitment of fission machinery; similar mechanisms may occur at tubulating LE/Lys.

We thank the reviewer for these interesting thoughts on the potential role for PI4P in the PS-dependent fission machinery. However, considering our most recent results we decided to focus on the role of CI-M6PR. We believe that exploring the potential role of PI4K2A would be an interesting project, which is out of the scope of the present study.

- The CI-M6PR assay only looks at a subpopulation of protein internalized from the PM - would be nice to see global distribution. It is suggested that CI-M6PR is diverted away from the Golgi to

the PM but is reduced at the PM - is that because it is so rapidly internalized? Any why is CI-M6PR total protein so reduced? Couldn't this alone explain the reduction in efficiency of lysosomal enzyme delivery?

We thank the reviewer for raising these points, as they contribute important insights to the story. As stated by the reviewer, in Figure 3e (previously 4c) we observed that CLN3-depleted cells have less PM CI-M6PR at the steady-state, as this measurement was performed by incubating cells with the lumCI-M6PR antibody at 4°C. We also observed that total levels of CI-M6PR were significantly reduced in CLN3-KO cells, and their levels went back to normal upon inhibition of lysosomal degradation with bafilomycin (Figure 3a, b). Also, bafilomycin treatment revealed that a portion of CI-M6PR, which in WT cells is detectable at the lysosomal membrane, accumulated within the lysosomal lumen in CLN3-depleted cells, suggesting mistrafficking of the receptor inside the lysosomes in KO cells (Figure 3c). To better dissect the trafficking of the receptor, we followed its sorting from the PM to Golgi, and found that this trafficking is impaired and that less receptor reaches the Golgi apparatus, remaining in peripheral structures which are endosomes (Figure 4a, b, Supplementary Figure 4d, e). This confirms that the trafficking defect happens at the level of endosome-to-Golgi delivery, and that this may cause the receptor redirection to lysosomes. As the reviewer suggested, we believe that depletion of CI-M6PR results in missorting of lysosomal enzymes in the medium, as it happens in other LSD diseases like Inclusion-cell (I-cell) disease and mucopolipidosis III (PMID: 8577054), and this in turn results in a defective lysosomal degradative function.

- Lysosomal enzymes appear reduced in the lysosome, but increased in the medium. Is this due to loss of M6PR retrograde traffic to the Golgi resulting in increased secretory traffic from the TGN? Or increased lysosome exocytosis? I assume the former but it would be nice to show if possible.

As reported in the previous point, we believe that release of lysosomal enzymes into the media is a result of increased secretory traffic from the TGN due to defect in CI-M6PR sorting. Secretion of lysosomal enzymes into the media is a well-established feature of conditions associated with defective M6P pathway (PMID: 8577054). An increase of lysosomal exocytosis would not explain the pathological accumulation of substrates (e.g. alpha-synuclein and mitochondrial proteins), that we observed in CLN3-KO lysosomes (Figure 5, Supplementary Figure 5).

- Fig. 6a) Assuming that GAA/LAMP colocalization is shown in white, by eye it appears increased in the KO image shown and LAMP1/EEA1 appear to both be staining several GAA +ve compartments. Do you have a different image to better reflect the quantitation?

We apologize for the confusion. In figure 5d (previously 6a) the signal represented in white indicates GAA particles. In the CLN3-KO cells, co-localization with endosomes is highlighted by the yellow arrows. We hope this clarification facilitates the interpretation of the data.

- Fig. 6b) The legend says 30 and 60 min but the fig says T0 and 60min. Please clarify what T0 means. In the KO. The Y axis label isn't visible.

In Figure 5e (previously 6b), cells were incubated with saturating concentrations of rhGAA in complete media at 37°C for 45 min. This time point is considered as the starting point (T0). Next, samples were washed and chased for the time points indicated in the figure legend and

quantified in the graph (30 and 60 min), but we just showed images relative to the 60min time point.

- Fig. 6c) This is very clear data but please add details of the dextran incubation to the legend (it's in the methods but easier to see in the legend).

We added this information in the legend.

- Fig. 6d) The PepA experiments need repeating in KO and WT cells to be able to draw any conclusions.

We repeated the experiment in WT and CLN3-KO cells as suggested by the reviewer. Unfortunately, we were unable to clearly detect lysosomal tubules due to the difficulty in detecting lysosomal tubules in WT cells after fixation. We noticed that PepA signal looked more diffuse in CLN3-KO cells but was still able to label lysosomes. This is not a surprise, because PepA labels Cathepsin D, whose levels were not affected in CLN3-KO lysosomes (Rebuttal Figure 9)

Rebuttal Figure 9

Labelling of lysosomes with PepA (green) and LAMP1 (red) in WT and CLN3-KO cells upon PS. Scale bar 10 μ M.

- In the discussion, line 285 reads "CLN3 localizes at the Golgi where it regulates the sorting of CI-M6PR". Your data seems to be more suggesting that CLN3 is predominantly functioning at the endosome to sort CI-M6PR to the Golgi.

In Figure 1 and Supplementary Figure 1 we observed that in WT cells CLN3 localizes at both Golgi and lysosomes, whereas little or no CLN3 was detected at endosomes. The same localization pattern was detected in CLN3-overexpressing cells (Supplementary Figure 1c). These data support a primary CLN3 function at Golgi rather than on endosomes.

This email has been sent through the Springer Nature Tracking System NY-610A-NPG&MTS

Confidentiality Statement:

This e-mail is confidential and subject to copyright. Any unauthorised use or disclosure of its contents is prohibited. If you have received this email in error please notify our Manuscript Tracking System Helpdesk team at <http://platformsupport.nature.com> .

Details of the confidentiality and pre-publicity policy may be found here <http://www.nature.com/authors/policies/confidentiality.html>

REVIEWER COMMENTS

Reviewer #1 (Remarks to the Author):

Overall, the authors have done a reasonable job of addressing the major concerns of the previous review. On the whole, this study presents insight into the function of CLN3 and will likely stimulate further work in this area.

Reviewer #2 (Remarks to the Author):

I had previously requested that the authors achieve a better mechanistic understanding of how Cln3 regulates two major processes that they have identified: i) autophagic lysosome reformation (ALR) and ii) cation-independent mannose 6-phosphate receptor (CI-M6PR) transport. In the case of ALR, they identified Cln3 to act through PI4K2A. Thus, because PI4K2A generates PI4P, how this particular phospholipid promotes ALR needed to be clarified. For CI-M6PR transport, I noted that studies have already found that Cln3 regulates CI-M6PR transport as well as identifying transport factors participating in this regulation (PMID 18817525 and 32034082). Thus, simply showing that Cln3 regulates the retrograde transport of CI-M6PR from endosome to Golgi would not represent a significant advancement.

For the revision, rather than addressing my two concerns directly, the authors have instead sought to improve the significance of their study by taking a different strategy, showing that: i) CI-M6PR is needed for Cln3 to regulate ALR, and ii) Cln3 deficiency results in the misrouting of CI-M6PR which leads to its degradation at the lysosome. Based on these new findings, the authors propose that the regulation of CI-M6PR transport is central to explaining how Cln3 affects lysosome biogenesis and function. While I am amenable to this alternate way of attempting to enhance the significance of their study, I also believe that the central role attributed to CI-M6PR needs to be better supported.

Specifically, the authors had pursued proteomic analysis of purified lysosomes, as well as performing functional studies on these lysosomes, in providing a detailed understanding of how Cln3 deficiency affects lysosomal content and function. A similar set of analyses should be done to examine the effect of CI-M6PR deficiency. If the authors are correct, the results should largely replicate those seen for Cln3 deficiency.

Reviewer #3 (Remarks to the Author):

The authors have undertaken extensive new experiments, which for the most part address my major concerns. There are however some outstanding concerns (one major) from my original evaluation that require further attention before I can recommend this manuscript for publication. Refer below for further comments.

Original major point #1: As requested, the authors have improved their explanation of ALR in the introduction and now also provided appropriate background information on the role of ALR dysfunction in LSDs. However, still lacking from this manuscript is acknowledgement that the roles of BORC and PI4K2A have already been established during ALR (PMID: 33618608, 33629936). While the relevant papers now appear in the reference list; Khundadze, M. et al. Mouse models for hereditary spastic paraplegia uncover a role of PI4K2A in autophagic lysosome reformation. *Autophagy* 17, 3690–3706 (2021), AND Wu, K. et al. BLOC1S1/GCN5L1/BORCS1 is a critical mediator for the initiation of autolysosomal tubulation. *Autophagy* 17, 3707 (2021) - I cannot find in the manuscript where there is acknowledgement and discussion of the known roles of PI4K2A and BLOC1S1 in ALR. These references numbered 55 and 57 do not appear to be discussed anywhere in the main text. I suspect there is significant text missing from the discussion since the reference numbers skip from 54 to 61. Without acknowledgement of the established roles of PI4K2A and BLOCS proteins during ALR, it gives the false impression that this is a novel finding of this work, when it is not.

Original major point #2: For the most part, the authors have provided convincing new data showing that CLN3 is required for ALR, with one major exception in Figure 6e. The number of lysosomes need to be quantified in CLN3 KO cells (+/- CLN3 rescue) under BOTH growth (basal) conditions and during prolonged starvation-induced autophagy. While the authors acknowledge that lysosome numbers may be decreased under growth conditions in CLN3 KO cells – this is insufficient to argue against the need to compare lysosome numbers under basal conditions versus prolonged starvation-induced autophagy. If CLN3 is a bona fide regulator of ALR, then CLN3 KO cells will be unable to sustain lysosome homeostasis in the switch from basal autophagy to prolonged starvation induced autophagy. While effects on lysosome morphology and size worsen in starved versus basal CLN3 KO cells – the critical experiment to do here is to quantify differences in lysosome number as ALR is a lysosome REFORMATION pathway. While the lysosome number may be different between CLN3 KO and control cells under basal conditions, what is critical to show is that the baseline number of lysosomes in CLN3 KO cells is further reduced during prolonged starvation and does not recover, unlike in control cells. While this essential ALR defining experiment that is still missing it is challenging to recommend acceptance of this manuscript. EM imaging to show “lysosome fusion and aggregation” are not definitive of ALR defects – for example, defective lysosome positioning can cause lysosome aggregation.

The legend for supplementary figure 7 is incorrect – panel b is not confocal images as stated, but is instead a Western blot. Panel c shows confocal images, not a Western blot. There is no panel d in the

figure, despite what is written in the legend. What do the individual columns in the graphs in panel c represent – this is not labelled nor stated in the legend. Given this problem, and the issues raised above with the discussion, the manuscript would benefit from a more thorough proof read by the authors.

In their rebuttal for minor point #4, the authors state that they now “report that CLN3 is degraded inside lysosomes in CLN3-KO cells (Figure 3)”. I am assuming CLN3 is a typo and they mean M6PR, which is an intriguing new observation. The new mTOR data (Figure 6C) is very interesting, especially when paired with the lysosomal enzyme sorting data in Figure 5, and likely explains a potential ALR defect.

Reviewer #5 (Remarks to the Author):

The manuscript by Ballabio and colleagues claims a role for the CLN3 protein in lysosomal biogenesis and argues that CLN3 functions as a hub in vesicular trafficking. Batten disease, caused by mutations in CLN3, is a rare and devastating type of neurodegenerative lysosomal storage disorder and understanding CLN3 function is a key step to understand disease pathology. The authors used proteomic analysis to identify several endo-lysosomal trafficking proteins that interact with CLN3, including the cation-independent mannose 6 phosphate receptor (CI-M6PR), which is responsible for targeting lysosomal enzymes to lysosomes.

The authors made several major claims: 1) CLN3 is responsible for CI-M6PR retrograde PM-to-Golgi sorting, and CLN3 depletion causes degradation of the receptor in the lysosome and secretion of lysosomal enzymes into the media. 2) They use an in-house developed antibody to show that there is a significant presence of CLN3 protein in the Golgi, which supports its role in trafficking. 3) They indicate that depletion of lysosomal enzymes because of the perturbed trafficking led to a failure of CLN3 KO cells to undergo autophagic lysosomal reformation (ALR), which requires starvation-induced lysosomal biogenesis and mTOR reactivation.

These observations seem to be well-supported by the data presented by the authors, however, the claim that is stated in the title is not supported i.e. “THE BATTEN DISEASE PROTEIN CLN3 IS A VESICULAR TRAFFICKING HUB COORDINATING LYOSOMAL BIOGENESIS AND FUNCTION”

To clarify, the observations provided by the authors in these cells are all reported in a wide range of lysosomal storage diseases (LSDs) and in cell and animal models with depletion of lysosomal proteins as indicated by the authors themselves and almost all the reviewers. I agree with the authors that CLN3 loss causes the observed cellular phenotypes including a reduction in lysosomal enzyme trafficking (previously reported in CLN3 models), mis-localization of CI-M6PR (also previously reported in CLN3

models) and ALR, which has been reported in other lysosomal disease models including NCL but is novel for NCL3 to my best knowledge. Examples of previous reports are cited by the other reviewers. I believe that the conclusions are over-interpretation of the results as there is no direct evidence that any of these phenotypes are directly caused by lacking CLN3 biochemical function instead of being secondary to lysosomal dysfunction, which is consistent with the fact that these cellular changes are being observed in a wide range of LSDs. This makes me believe that the title used by the author is not appropriate and it misleads the readers to believe that these cellular functions are directly mediated by CLN3 protein, a claim that has no direct data to support.

Despite this, I acknowledge the importance of proper models to study Batten disease and commend the authors for their thorough characterization of their cellular models, including the ARPE cell model, as well as their omics analysis, which can be a valuable resource for the community. I suggest that the authors modify the title to describe their cellular model rather than the function of the CLN3 protein. A title such as “BATTEN DISEASE CELL MODELS SHOW VESICULAR TRAFFICKING AND LYSOSOMAL BIOGENESIS FAILURE” would be more appropriate.

Few major concerns related to the technical part:

- 1) The validation of the antibody for IF is not strong. Clear signal is observed in the CLN3 KO cells in Fig. 1C. These experiments should be performed side by side with wt cells to compare the signal. I agree that the validation of the antibody for immunoblot is well-done, however, the IF validation is more important as this is the approach the authors used to make the Golgi localization claim (see below).
- 2) Another major concern is the co-localization experiments. These should be done with Nocodazole treatment as the cytoplasm seems very crowded with even LAMP1 very close and overlaps with TGN46. It is hard to use these images to make a major claim that CLN3 is a Golgi resident protein. This concern becomes more relevant when looking at the Golgi localization in CLN3-innHA used in Fig. S1c, which is not convincing at all.
- 3) The data for CI-M6PR and CLN3 are very confusing and difficult to use to support the model that the authors propose towards the end of the manuscript. While the authors claim that CLN3 is needed for the PM to Golgi retrograde trafficking of the receptor, CLN3 was never found on the PM nor on the early endosomes based on the provided experiments. Furthermore, the final outcome that the authors report is more degradation of CI-M6PR in the lysosome in CLN3-deficient cells. This leads me to question the direct involvement of CLN3 in CI-M6PR trafficking and again argues that the effects on the receptor are secondary to lysosomal dysfunction. Proving a direct relationship between the two proteins and their functions require more rigorous experiments that include mapping the binding interface and generating binding deficient mutants that will recapitulate the KO phenotypes, which I believe is beyond the scope of this work as long as the claims about the CLN3 protein function are changed based on the suggestion above.

Reviewer #1 (Remarks to the Author):

Overall, the authors have done a reasonable job of addressing the major concerns of the previous review. On the whole, this study presents insight into the function of CLN3 and will likely stimulate further work in this area.

Reviewer #2 (Remarks to the Author):

I had previously requested that the authors achieve a better mechanistic understanding of how Cln3 regulates two major processes that they have identified: i) autophagic lysosome reformation (ALR) and ii) cation-independent mannose 6-phosphate receptor (CI-M6PR) transport. In the case of ALR, they identified Cln3 to act through PI4K2A. Thus, because PI4K2A generates PI4P, how this particular phospholipid promotes ALR needed to be clarified. For CI-M6PR transport, I noted that studies have already found that Cln3 regulates CI-M6PR transport as well as identifying transport factors participating in this regulation (PMID 18817525 and 32034082). Thus, simply showing that Cln3 regulates the retrograde transport of CI-M6PR from endosome to Golgi would not represent a significant advancement.

For the revision, rather than addressing my two concerns directly, the authors have instead sought to improve the significance of their study by taking a different strategy, showing that: i) CI-M6PR is needed for Cln3 to regulate ALR, and ii) Cln3 deficiency results in the misrouting of CI-M6PR which leads to its degradation at the lysosome. Based on these new findings, the authors propose that the regulation of CI-M6PR transport is central to explaining how Cln3 affects lysosome biogenesis and function. While I am amenable to this alternate way of attempting to enhance the significance of their study, I also believe that the central role attributed to CI-M6PR needs to be better supported.

Specifically, the authors had pursued proteomic analysis of purified lysosomes, as well as performing functional studies on these lysosomes, in providing a detailed understanding of how Cln3 deficiency affects lysosomal content and function. A similar set of analyses should be done to examine the effect of CI-M6PR deficiency. If the authors are correct, the results should largely replicate those seen for Cln3 deficiency.

We thank the reviewer for appreciating the enhanced significance of our study. We agree with the reviewer on the importance of demonstrating that the results obtained in CI-M6PR deficient cells replicate those obtained in Cln3 deficient cells. In order to demonstrate this, the reviewer is asking us to perform lysosomal proteomic and functional analyses in CI-M6PR deficient cells. However, such studies have already been performed by multiple groups and there are extensive data in the literature showing that alteration of lysosomal enzyme trafficking results in reduced lysosomal enzyme levels and activities. Indeed mis-sorting of lysosomal enzymes and their secretion into the medium has already been shown for MLII and

MLIII, two diseases associated with defects of the M6P pathway (PMID 29773673, 16200072) with consequent impairment of lysosomal degradative capacity and storage of undegraded substrates. More recently, Pechincha et al. (PMID 36074822) showed that LYSET is a core component of the M6P pathway and that its depletion determines multiple lysosomal phenotypes, including enlargement of the lysosomal compartment, accumulation of undigested material, depletion of several enzymes from the lysosomes and their secretion into the culture medium. In this study, the authors also performed proteomic analyses of lysosomes isolated from LYSET-depleted cells, which showed severe depletion of several lysosomal enzymes. Most importantly, all of the 14 enzymes that we identified as being reduced in CLN3-KO lysosomes were also reduced in LYSET-depleted cells. These results confirm that alteration of the M6P pathway determines a lysosomal phenotype that is very similar to the one observed in CLN3-depleted cells. This information, together with the reference from Pechincha et al. (PMID 36074822), has now been included in the discussion and is reported in Table 1.

Reviewer #3 (Remarks to the Author):

The authors have undertaken extensive new experiments, which for the most part address my major concerns. There are however some outstanding concerns (one major) from my original evaluation that require further attention before I can recommend this manuscript for publication. Refer below for further comments.

Original major point #1: As requested, the authors have improved their explanation of ALR in the introduction and now also provided appropriate background information on the role of ALR dysfunction in LSDs. However, still lacking from this manuscript is acknowledgement that the roles of BORC and PI4K2A have already been established during ALR (PMID: 33618608, 33629936). While the relevant papers now appear in the reference list; Khundadze, M. et al. Mouse models for hereditary spastic paraplegia uncover a role of PI4K2A in autophagic lysosome reformation. *Autophagy* 17, 3690–3706 (2021), AND Wu, K. et al. BLOC1S1/GCN5L1/BORCS1 is a critical mediator for the initiation of autolysosomal tubulation. *Autophagy* 17, 3707 (2021) - I cannot find in the manuscript where there is acknowledgement and discussion of the known roles of PI4K2A and BLOC1S1 in ALR. These references numbered 55 and 57 do not appear to be discussed anywhere in the main text. I suspect there is significant text missing from the discussion since the reference numbers skip from 54 to 61. Without acknowledgement of the established roles of PI4K2A and BLOCS proteins during ALR, it gives the false impression that this is a novel finding of this work, when it is not.

The reviewer is right. Unfortunately, this was due to an error that occurred while editing the text. We apologize for this. We have now added the following paragraph to the discussion:

Other interactors identified in our proteomic analysis may also play a role in the lysosomal reformation process. Lyspersin, a BORC subunit essential for lysosomal positioning and motility⁴⁸, as well as PI4K2A, a member of the family of phosphatidylinositol (PtdIns) 4-kinases responsible for phosphatidylinositol 4-monophosphate (PI4P) production^{49,50}, were also highly enriched among CLN3 interactors. The BORC subunit BLOC1S1 was recently shown to be important in

*the initiation of the lysosomal tubulation process*⁵¹. Also, mutations of *Borcs7*, a subunit of the BORG complex, result in a neurodegenerative phenotype associated with progressive axonal dystrophy and perinatal death in mice⁵². Furthermore, *PI4K2A* was shown to play a role in ALR in the context of hereditary spastic paraplegia (HSP), a group of neurodegenerative diseases associated with severe motor decline^{53,54}.

Original major point #2: For the most part, the authors have provided convincing new data showing that CLN3 is required for ALR, with one major exception in Figure 6e. The number of lysosomes need to be quantified in CLN3 KO cells (+/- CLN3 rescue) under BOTH growth (basal) conditions and during prolonged starvation-induced autophagy. While the authors acknowledge that lysosome numbers may be decreased under growth conditions in CLN3 KO cells – this is insufficient to argue against the need to compare lysosome numbers under basal conditions versus prolonged starvation-induced autophagy. If CLN3 is a bona fide regulator of ALR, then CLN3 KO cells will be unable to sustain lysosome homeostasis in the switch from basal autophagy to prolonged starvation induced autophagy. While effects on lysosome morphology and size worsen in starved versus basal CLN3 KO cells – the critical experiment to do here is to quantify differences in lysosome number as ALR is a lysosome REFORMATION pathway. While the lysosome number may be different between CLN3 KO and control cells under basal conditions, what is critical to show is that the baseline number of lysosomes in CLN3 KO cells is further reduced during prolonged starvation and does not recover, unlike in control cells. While this essential ALR defining experiment that is still missing it is challenging to recommend acceptance of this manuscript. EM imaging to show “lysosome fusion and aggregation” are not definitive of ALR defects – for example, defective lysosome positioning can cause lysosome aggregation.

As stated by the reviewer, specific defects of ALR are characterized by a reduced number of lysosomes during prolonged starvation, which does not increase over time, unlike what is observed in control cells. However, we wish to emphasize that CLN3-KO cells are defective not only for ALR but also for lysosomal degradative function. Rong et al (PMID 21518918) previously showed that during prolonged starvation a portion of the lysosomes is initially consumed until they are restored by ALR upon starvation-induced autophagy. We believe that depletion of lysosomal hydrolases in CLN3-KO cells prevents lysosomal consumption during the initial phase of starvation. To address the reviewer's concern we performed a new experiment in which we quantified lysosome number at 3 different time points during starvation. As shown in the new Figure 7b and in new Supplementary Figure 8d, we observed that in CLN3-KO cells the number of lysosomes remains stable over time, unlike in WT cells. This suggests that lysosomes cannot be consumed in the initial phase of starvation and cannot be reformed during prolonged starvation. Conversely, we observed that in CLN3-overexpressing cells (in which lysosomal degradative capacity is unaffected) the number of lysosomes is: 1) consistently higher compared to WT cells, 2) it decreases after 8 hrs of starvation, and 3) it recovers after 16 hrs to levels that are significantly higher compared to WT cells. A comment on these findings has also been added to the discussion.

The legend for supplementary figure 7 is incorrect – panel b is not confocal images as stated, but is instead a Western blot. Panel c shows confocal images, not a Western blot. There is no panel d in the figure, despite what is written in the legend. What do the individual columns in the graphs in panel c represent – this is not labelled nor stated in the legend. Given this problem, and the issues raised above with the discussion, the manuscript would benefit from a more thorough proof read by the authors.

We apologize for the errors. We have now fixed the figure and figure legend. Individual columns in panel c indicate the following cell lines: WT = black dots, CLN3-KO = red dots and CLN3-KO+CLN3 = blue dots.

In their rebuttal for minor point #4, the authors state that they now “report that CLN3 is degraded inside lysosomes in CLN3-KO cells (Figure 3)”. I am assuming CLN3 is a typo and they mean M6PR, which is an intriguing new observation. The new mTOR data (Figure 6C) is very interesting, especially when paired with the lysosomal enzyme sorting data in Figure 5, and likely explains a potential ALR defect.

We apologize for the error in the rebuttal, the reviewer is correct. The correct sentence is “we report that CI-M6PR is degraded inside lysosomes in CLN3-KO cells (Figure 3)”. We thank the reviewer for pointing out the relevance of this discovery.

Reviewer #5 (Remarks to the Author):

The manuscript by Ballabio and colleagues claims a role for the CLN3 protein in lysosomal biogenesis and argues that CLN3 functions as a hub in vesicular trafficking. Batten disease, caused by mutations in CLN3, is a rare and devastating type of neurodegenerative lysosomal storage disorder and understanding CLN3 function is a key step to understand disease pathology. The authors used proteomic analysis to identify several endo-lysosomal trafficking proteins that interact with CLN3, including the cation-independent mannose 6 phosphate receptor (CI-M6PR), which is responsible for targeting lysosomal enzymes to lysosomes.

The authors made several major claims: 1) CLN3 is responsible for CI-M6PR retrograde PM-to-Golgi sorting, and CLN3 depletion causes degradation of the receptor in the lysosome and secretion of lysosomal enzymes into the media. 2) They use an in-house developed antibody to show that there is a significant presence of CLN3 protein in the Golgi, which supports its role in trafficking. 3) They indicate that depletion of lysosomal enzymes because of the perturbed trafficking led to a failure of CLN3 KO cells to undergo autophagic lysosomal reformation (ALR), which requires starvation-induced lysosomal biogenesis and mTOR reactivation.

These observations seem to be well-supported by the data presented by the authors, however, the claim that is stated in the title is not supported i.e “THE BATTEN DISEASE

PROTEIN CLN3 IS A VESICULAR TRAFFICKING HUB COORDINATING LYSOSOMAL BIOGENESIS AND FUNCTION"

To clarify, the observations provided by the authors in these cells are all reported in a wide range of lysosomal storage diseases (LSDs) and in cell and animal models with depletion of lysosomal proteins as indicated by the authors themselves and almost all the reviewers. I agree with the authors that CLN3 loss causes the observed cellular phenotypes including a reduction in lysosomal enzyme trafficking (previously reported in CLN3 models), mis-localization of CI-M6PR (also previously reported in CLN3 models) and ALR, which has been reported in other lysosomal disease models including NCL but is novel for NCL3 to my best knowledge. Examples of previous reports are cited by the other reviewers.

I believe that the conclusions are over-interpretation of the results as there is no direct evidence that any of these phenotypes are directly caused by lacking CLN3 biochemical function instead of being secondary to lysosomal dysfunction, which is consistent with the fact that these cellular changes are being observed in a wide range of LSDs. This makes me believe that the title used by the author is not appropriate and it misleads the readers to believe that these cellular functions are directly mediated by CLN3 protein, a claim that has no direct data to support.

Despite this, I acknowledge the importance of proper models to study Batten disease and commend the authors for their thorough characterization of their cellular models, including the ARPE cell model, as well as their omics analysis, which can be a valuable resource for the community. I suggest that the authors modify the title to describe their cellular model rather than the function of the CLN3 protein. A title such as "BATTEN DISEASE CELL MODELS SHOW VESICULAR TRAFFICKING AND LYSOSOMAL BIOGENESIS FAILURE" would be more appropriate.

We thank the reviewer for acknowledging the importance of our observations. We agree with the reviewer that, at least in part, the abnormalities observed in CLN3 KO cells may be a secondary consequence of lysosomal dysfunction. However, we believe that whereas impaired ALR in CLN3 KO cells is likely caused by defective lysosomal function, the mis-trafficking of M6PR and consequent mis-sorting of lysosomal enzymes is likely not the consequence, but rather the cause, of lysosomal dysfunction. Considering that proving a direct role of CLN3 in the trafficking of CI-M6PR would be out of the scope of this manuscript, we decided to change the title as follows: "LOSS OF THE BATTEN DISEASE PROTEIN CLN3 LEADS TO MISTRAFFICKING OF CI-M6PR AND DEFECTIVE AUTOPHAGIC-LYSOSOMAL REFORMATION". This title does not indicate a direct role of CLN3 in either trafficking of CI-M6PR or ALR but leaves this possibility open for future studies.

Few major concerns related to the technical part:

- 1) The validation of the antibody for IF is not strong. Clear signal is observed in the CLN3 KO cells in Fig. 1C. These experiments should be performed side by side with wt cells to compare

the signal. I agree that the validation of the antibody for immunoblot is well-done, however, the IF validation is more important as this is the approach the authors used to make the Golgi localization claim (see below).

We wish to clarify that the data obtained in CLN3 KO cells and shown in the "old" Figure 1c were indeed generated side by side with those obtained in WT. We apologize for the unclarity of our data presentation format. As suggested by the reviewer, in the new Figure 1 the data of CLN3 KO and WT cells are shown side by side and reported in Figure 1a. Furthermore, in Rebuttal Figure 1 we show additional examples of images of WT and CLN3-KO cells, also generated in parallel.

Rebuttal Figure 1

ARPE19 WT and CLN3-KO cells stained for CLN3. All scale bars are 40 μ m.

2) Another major concern is the co-localization experiments. These should be done with Nocodazole treatment as the cytoplasm seems very crowded with even LAMP1 very close and overlaps with TGN46. It is hard to use these images to make a major claim that CLN3 is a Golgi resident protein. This concern becomes more relevant when looking at the Golgi localization in CLN3-innHA used in Fig. S1c, which is not convincing at all.

The reviewer is correct. For clarity we have now added single channel images (CLN3-innHA signal in green and TGN46 signal in blue) to Fig. S1c. Furthermore, to better resolve CLN3 Golgi localization, we performed Airyscan confocal super-resolution microscopy, with and

without Nocodazole treatment, as suggested by the reviewer. The results are now shown in the new Figure 1c.

3) The data for CI-M6PR and CLN3 are very confusing and difficult to use to support the model that the authors propose towards the end of the manuscript. While the authors claim that CLN3 is needed for the PM to Golgi retrograde trafficking of the receptor, CLN3 was never found on the PM nor on the early endosomes based on the provided experiments. Furthermore, the final outcome that the authors report is more degradation of CI-M6PR in the lysosome in CLN3-deficient cells. This leads me to question the direct involvement of CLN3 in CI-M6PR trafficking and again argues that the effects on the receptor are secondary to lysosomal dysfunction. Proving a direct relationship between the two proteins and their functions require more rigorous experiments that include mapping the binding interface and generating binding deficient mutants that will recapitulate the KO phenotypes, which I believe is beyond the scope of this work as long as the claims about the CLN3 protein function are changed based on the suggestion above.

As stated above, we do not think that the intra-lysosomal degradation of CI-M6PR is secondary to lysosomal dysfunction, but that it is rather the cause. Pechincha et al. (PMID 36074822) recently showed that loss of the lysosomal protein LYSET led to impairment of the M6P pathway and depletion of several enzymes from the lysosomes, together with their consequent secretion into the medium. We found that all of the 14 enzymes that were reduced in CLN3-KO lysosomes, were also reduced in LYSET-KO lysosomes. This information is now reported in the discussion and in Table 1. We also would like to emphasize that lysosomal dysfunction does not normally cause a general intra-lysosomal depletion of lysosomal enzymes. Several conditions associated with lysosomal dysfunction (e.g. lysosomal storage diseases due to single hydrolase deficiency), or even treatment of WT cells with lysosomal inhibitors such as bafilomycin, are not associated with general mis-targeting of lysosomal enzymes and their general depletion. However, we agree with the reviewer that the interaction between CLN3 and CI-M6PR may be indirect and that demonstration of direct binding will require mapping of the binding interface, which would be out of the scope of this manuscript. Therefore, we changed the title of the paper and removed all the claims stating a direct effect of CLN3 in the regulation of the PM-to-Golgi trafficking of CI-M6PR.

** See Nature Portfolio's author and referees' website at www.nature.com/authors for information about policies, services and author benefits.

This email has been sent through the Springer Nature Tracking System NY-610A-NPG&MTS

Confidentiality Statement:

This e-mail is confidential and subject to copyright. Any unauthorised use or disclosure of its contents is prohibited. If you have received this email in error please notify our Manuscript Tracking System Helpdesk team at <http://platformsupport.nature.com> .

Details of the confidentiality and pre-publicity policy may be found here <http://www.nature.com/authors/policies/confidentiality.html>

Privacy Policy | Update Profile

DISCLAIMER: This e-mail is confidential and should not be used by anyone who is not the original intended recipient. If you have received this e-mail in error please inform the sender and delete it from your mailbox or any other storage mechanism. Springer Nature America, Inc. does not accept liability for any statements made which are clearly the sender's own and not expressly made on behalf of Springer Nature America, Inc. or one of their agents. Please note that neither Springer Nature America, Inc. or any of its agents accept any responsibility for viruses that may be contained in this e-mail or its attachments and it is your responsibility to scan the e-mail and attachments (if any).

REVIEWERS' COMMENTS

Reviewer #2 (Remarks to the Author):

The revision has responded to my concern adequately. I have no further concerns.

Reviewer #3 (Remarks to the Author):

The authors of this study have now satisfactorily addressed all of my concerns both major and minor, and I now recommend this manuscript ready for publication.

Reviewer #5 (Remarks to the Author):

I thank the authors for following my suggestion by focusing on the consequence of CLN3 loss rather than claiming a direct role of CLN3 in the studied processes.

A major concern I have had was the co-localization experiments that led to the claim that CLN3 is a resident Golgi protein. The authors kindly repeated these experiments with Nocodazole treatment and added the results to Fig. 1c. The result they obtained further supports the concern as addition of Nocodazole lowered the colocalization between CLN3 and TGN46, at least from a qualitative look at the provided image. Because this is a major claim in the paper and directly relates to several other conclusions, this point should be addressed in a more rigorous manner. These imaging experiments should be quantified and a correlation metric should be used to provide objective assessment of the colocalization or the lack of it.

Reviewer #5:

I thank the authors for following my suggestion by focusing on the consequence of CLN3 loss rather than claiming a direct role of CLN3 in the studied processes.

A major concern I have had was the co-localization experiments that led to the claim that CLN3 is a resident Golgi protein. The authors kindly repeated these experiments with Nocodazole treatment and added the results to Fig. 1c. The result they obtained further supports the concern as addition of Nocodazole lowered the colocalization between CLN3 and TGN46, at least from a qualitative look at the provided image. Because this is a major claim in the paper and directly relates to several other conclusions, this point should be addressed in a more rigorous manner. These imaging experiments should be quantified and a correlation metric should be used to provide objective assessment of the col-localization or the lack of it.

Regarding the reviewer's statement "A major concern I have had was the co-localization experiments that led to the claim that CLN3 is a resident Golgi protein", we would like to clarify that we have never claimed that CLN3 is a "resident Golgi protein" and we do not believe this to be the case. Indeed, we clearly showed that CLN3 traffics through the Golgi to reach the lysosomal compartment. Multiple lines of evidence support this conclusion (Fig. 1 and Fig.S2). At the steady state (endogenous protein Fig.1a and Suppl Fig.1, or inducible tagged protein after 18 h of induction Suppl. Fig 2f) CLN3 has a Golgi and a lysosome localization. The observation that at early times of induction CLN3 is mainly localized at the Golgi, whereas it is significantly reduced after a wash out of the inducer in the absence of newly synthesized protein (CHX treatment), strongly suggests that the newly synthesized protein reaches the Golgi and transits through this organelle to reach the lysosomes (Suppl Fig. 2f). Interestingly this change of localization is nicely paralleled by a shift in the glycosylation of the protein with partially and fully glycosylated CLN3 forms being present at the steady state (18h), the incompletely glycosylated form being predominant at early times of induction (2, 4h), and fully glycosylated form being the predominant form after wash-out and CHX treatment. (Fig. 1 e). Thus, altogether our data show that CLN3 traffics through the Golgi complex as a cargo protein to be properly glycosylated and sorted to the lysosomes.

We agree with the reviewer that nocodazole treatment would be a valuable experiment to assess the Golgi and sub-Golgi localization of "Golgi resident proteins", however, considering that CLN3 is not a Golgi resident protein, but it traffics through the Golgi to reach the lysosomes, this experiment may yield misleading results. Indeed, the treatment with nocodazole (as the one in Fig. 1C: 1,5 h at 33micromolar) is known to impair the trafficking of newly synthesized proteins to and through the Golgi and may also affect many endosomal trafficking steps (Gruenberg et al. 1989; Bomsel et al. 1990, Mallet et al 1999). Therefore, we decided to remove the nocodazole experiment from the manuscript.

In response to the reviewer's comment, we better clarified the concept that CLN3 is not a Golgi-resident protein by including the following paragraphs:

"Notably, upon short doxycycline treatment, CLN3 exclusively localized to the Golgi apparatus (de novo synthesis), whereas the long doxycycline washout resulted in reduction of the protein at Golgi and appearance of the protein on lysosomes, indicating that the newly synthesized protein transits through the Golgi to reach the lysosomes (Supplementary Fig. 2f)."

This paragraph has been included in the results section entitled "CLN3 traffics through the Golgi apparatus and lysosomes".

"We also observed that CLN3 Golgi localization, which was visible at early times of induction, was significantly reduced upon inhibition of protein biosynthesis (CHX treatment). These results strongly suggest that CLN3 is not a Golgi resident protein, but that the newly synthesized protein transits through the Golgi to reach the lysosomes."

This paragraph has now been included in the discussion.